# ONLINE BANDIT NONLINEAR CONTROL WITH DYNAMIC BATCH LENGTH AND ADAPTIVE LEARNING RATE

## ABSTRACT

This paper is concerned with the online bandit nonlinear control, which aims to learn the best stabilizing controller from a pool of stabilizing and destabilizing controllers of unknown types for a given nonlinear dynamical system. We develop an algorithm, named **D**ynamic **B**atch length and **A**daptive learning **R**ate (DBAR), and study its stability and regret. Unlike the existing Exp3 algorithm requiring an exponentially stabilizing controller, DBAR only needs a significantly weaker notion of controller stability, in which case substantial time may be required to certify the system stability. Dynamic batch length in DBAR effectively addresses this issue and enables the system to attain asymptotic stability, where the algorithm behaves as if there were no destabilizing controllers. Moreover, adaptive learning rate in DBAR only uses the state norm information to achieve a tight regret bound even when none of the stabilizing controllers in the pool are exponentially stabilizing.

## 1 INTRODUCTION

The multi-armed bandit (MAB) problem aims to minimize the total cost of pulling a series of arms while receiving immediate cost feedback for each arm pulled. Given a finite number of arms, the problem balances between exploration and exploitation of arms without knowing the exact cost structure of each arm. On the other hand, the online optimal control problem considers a transition dynamic $x_{t+1} = f(x_t, u_t, w_t)$ and a set of cost functions $c_t(x_t, u_t)$, $t = 0, \dots, T$, where the goal is to minimize the sum of costs over time, while both $f$ and $c_t$ are fully or partially unknown. Basically, MAB is a special type of the online optimal control problem in the sense that MAB is stateless and simply selects an action each time, while the online control problem has a countable or an uncountable number of states and selects a controller, acting as a function from states into actions, each time without knowing the cost functions. Bandit algorithms can thus be leveraged for online control, wherein the average cost incurred with a controller can be interpreted as the bandit feedback of pulling the controller-arm (Lin et al., 2023; Li et al., 2023).

In this paper, we address the online nonstochastic control problem where both a transition dynamic $f$ and cost functions $c_t$ can be *unbounded*, *nonlinear*, and *adversarially* chosen. We only have knowledge about $x_t$ and the bandit feedback $c_t(x_t, u_t)$ at time $t$, with adversarial disturbances $w_t$ injected at each time step as in Gradu et al. (2020) and Cassel & Koren (2020). We operate the system with a single trajectory where the system state cannot be reset. To overcome the difficulties of an unknown nonlinear system, we are given a finite set of $N$ controllers in advance, where we are not aware of whether each controller can stabilize the system but we are allowed to alternate between these controllers within a single trajectory according to a specific logic. We refer to this problem as the *online bandit nonlinear control* problem.

To deal with this online bandit nonlinear control, Li et al. (2023) adopted their Exp3-ISS algorithm, which uses the well-known Exp3 algorithm (Auer et al., 2002) with a mini-batch approach (Arora et al., 2012), while successively removing destabilizing controllers when detected in terms of input-to-state stability (ISS). In this paper, we aim to significantly relax the requirement on the controllers and yet guarantee asymptotic stability of the closed-loop system and sharpen the regret bound by designing our algorithm **DBAR** (**D**ynamic **B**atch length and **A**daptive learning **R**ate).

Table 1: Summary of required controllers and results: $\mathcal{U}$ is the set of destabilizing controllers and $|\mathcal{U}|$ denotes its cardinality. Polynomial factors on $N$ and $|\mathcal{U}|$ are hidden.

| Algorithm | Required Controller | Closed-loop system asymptotic stability | Regret Bound |
|---|---|---|---|
| Chen & Hazan (2021) | Exponential | N/A | $\tilde{O}(T^{2/3}) + \exp(O(|\mathcal{U}|))$ |
| Li et al. (2023) | Exponential | No | $\tilde{O}(T^{2/3}) + \exp(O(|\mathcal{U}|))$ |
| Dynamic Batching | Asymptotic | Yes | $\tilde{O}(T^{2/3}) + o(T^{1/3}) \cdot \exp(O(|\mathcal{U}|))$ |
| Algorithm 1 (DBAR) | Asymptotic | Yes | $\tilde{O}(T^{2/3}) + \tilde{O}(T^{-1/3}) \cdot \exp(O(|\mathcal{U}|))$ |

**Motivation and contribution.** Our main contribution is to allow a broader class of controllers to qualify as a stabilizing controller within *a priori* controller pool. For the motivation, consider a continuous-time gradient flow in the vector space:

$$\dot{x}(t) = -\nabla F(x(t)), \tag{1}$$

where $F : \mathbb{R}^n \to \mathbb{R}$ is a smooth function. A merely convex $F$ can be extremely flat around its minimum, leading to a slowly (asymptotically) converging trajectory unlike exponentially converging behavior achieved for strongly convex $F$ (Khalil, 2015). In fact, assuming that a minimizer $x^*$ of $F$ exists, the decay rate $F(x(t)) - F(x^*)$ is $O(1/(t \log^2 t))$ if $F$ is convex[1] (Siegel & Wojtowytsch, 2023), and $O(e^{-t})$ if $F$ is strongly convex. In the machine learning literature, a loss function $l(g(x), y)$ of a gradient-based method is often given as a convex function in $g$ (*e.g.*, mean-squared error or cross-entropy loss), but not necessarily strongly convex since $g$ is often over-parameterized and there could be a continuum of parameters corresponding to the value of $g$. Analogous to this concept, one can consider $F$ as $f(x_t, \pi(x_t), w_t)$, a dynamic governed by a given controller $\pi$ and its converging behavior as a (asymptotic or exponential) controller stability. Our work merely requires the existence of at least one *asymptotically* stabilizing controller in the pool, which is far weaker than *exponentially* stabilizing notions and represents a more realistic environment one may encounter.

The existing literature on online bandit control of linear dynamics with adversarial disturbance has intrinsically assumed the existence of strongly stable controllers, which are exponentially stabilizing controllers in our context, and achieves $\tilde{O}(T^{2/3})$ regret under general convex cost functions (Cassel & Koren, 2020; Chen & Hazan, 2021; Ghai et al., 2023). In this paper, we will achieve the same $\tilde{O}(T^{2/3})$ regret bound even when none of the stabilizing controllers are exponentially stabilizing.

**Algorithm Design.** The idea of our algorithm is two-fold:

1. We adopt a dynamic batch length instead of a fixed length to certify the stability of the system without requiring exponentially stabilizing controllers and achieve *both* asymptotic system stability and a sublinear regret bound. The batch length is scheduled to be non-decreasing and growing unboundedly over time, but its growth amount eventually saturates. However, the strategy suffers from a resulting multiplicative exponential regret in return.

2. To alleviate the multiplicative exponential regret without requiring the conservative notion of exponentially stabilizing controllers, we adopt a novel adaptive learning rate scheme that relies on the system state norm, instead of a fixed learning rate. While the conventional way to apply the Exp3 Algorithm is to use a non-increasing learning rate, we decrease the learning rate if the state is unstable and subsequently increase the learning rate if the state returns to a stable region. By implementing this approach, we can alleviate the multiplicative exponential term in all cases. In particular, for a specific class of stabilizing controllers beyond exponential notions, we attain a regret bound order $[\tilde{O}(T^{2/3}) + \tilde{O}(T^{-1/3}) \cdot \exp(O(|\mathcal{U}|))] \cdot (|\mathcal{U}| + 1)^\alpha$, where $\alpha = 1/3$ if $|\mathcal{U}|$ is known and $\alpha = 1/2$ if $|\mathcal{U}|$ is unknown.

Table 1 shows a summary of our results with related works. Appendix A provides more details on the intermediate step "Dynamic Batching", which operates under asymptotically stabilizing controller assumptions, and on how we devised DBAR algorithm to avoid the multiplicative exponential term.

**Related works.** *Optimal control* problems have been widely leveraged in a variety of fields with the influential dynamic programming approach (Bellman, 1957). Recent successes of reinforcement

---

[1]Note that $O(1/(t \log^2 t))$ is integrable at infinity. In the context of controllers, we also handle the challenging case where $f(x_t, \pi(x_t), w_t) - \inf_{x \in \mathbb{R}^n} f(x, \pi(x), w_t)$ may not be integrable at infinity. This corresponds to a convex function without minimizers, such as a log-exp-type softmax loss function for classification.

learning (RL) in safety-critical systems, such as aircraft (Razzaghi et al., 2022), robotics (Ibarz et al., 2021), and autonomous driving (Kiran et al., 2021), are also deeply rooted in optimal control methods (Bertsekas, 2019). The common idea to gain system stability of optimal control problems is to falsify the detected destabilizing controller, meaning that one can completely remove those controllers failing to satisfy certain stability criteria from the controller pool (Baldi et al., 2010; Battistelli et al., 2010; 2014; 2018; Stefanovic & Safonov, 2011; Li et al., 2023).

*Online nonstochastic control* considers a dynamical system with adversarial disturbances, which is more challenging than having statistical noise. Early papers assumed full access to cost functions, enabling us to leverage optimal policy structure with cost function gradients (Agarwal et al., 2019; Foster & Simchowitz, 2020; Hazan et al., 2020; Hazan & Singh, 2022). Later, studies were generalized to address the problem without cost gradients information (Gradu et al., 2020; Cassel & Koren, 2020; Ghai et al., 2023; Sun et al., 2023); instead, they estimated the cost gradients, using the history of scalar cost (bandit feedback) along the trajectory. However, the above research restricts the system to linear transition dynamics. Instead, our work considers the candidate controller pool to handle unknown nonlinear systems.

*Multi-armed bandits* with adversarial disturbances were first addressed in the pioneering work by Auer et al. (2002) under bounded costs in their notable Exp3 algorithm. Arora et al. (2012) later improved the algorithm using the same controller within a mini-batch, attaining a regret bound equivalent to the lower bound presented in Dekel et al. (2014). As we have access to the candidate controller pool in our problem setting, we adopt a bandit-related approach.

*Dynamic batching* gained considerable attention for training deep neural networks by increasing the batch size over time and adaptively increasing the learning rate to maintain the ratio between the two (Devarakonda et al., 2017; Bollapragada et al., 2018; Shallue et al., 2019; Ma et al., 2023). Although this has been widely used in the machine learning literature, we adopt this idea to online control, progressively increasing the batch length within a single trajectory to achieve asymptotic stability.

*Adaptive learning rate* in machine learning is generally determined by a set of gradients observed so far (Ruder, 2016). As we do not have access to the gradients in our problem, we focus on the learning rate for bandit algorithms. Several works (van Erven et al., 2011; de Rooij et al., 2014) in hedge setting, an instance of multi-armed bandit problem, suggested using decreasing learning rate as the batch length increases. Building on this idea, Li et al. (2023) proposed to use a non-increasing learning rate over time, while no theoretical guarantee was presented. To the best of our knowledge, this paper is the first work to provide theoretical guarantees for the adaptive learning rate scheme based on the stability of state norm, where the rate is not necessarily non-increasing.

**Outline.** The paper is organized as follows. In Section 2, we formulate the problem and provide necessary definitions and assumptions. In Section 3, we propose our DBAR algorithm. In Section 4, we study the stability of the algorithm, the regret bound, and its applications in switched systems. In Section 5, we present numerical experiments on the DBAR algorithm with an ablation study on batch length and learning rate. Finally, concluding remarks are provided in Section 6.

**Notation.** For a vector $z$, $\|z\|$ denotes the Euclidean norm of the vector. We use $O(\cdot)$ for the big-O notation, $o(\cdot)$ for the small-o notation, and $\tilde{O}(\cdot)$ for the big-O notation hiding logarithmic factors. Let $\mathbb{E}$ denote the expectation operator. For a set $Z$, we use $|Z|$ for the cardinality and $Z^c$ for the complement of the set $Z$. For a real number $e$, we use $\lfloor e \rfloor$ for the floor and $\lceil e \rceil$ for the ceiling of $e$. Let $\mathbb{R}$ denote the set of real numbers and $\mathbb{Z}_+$ denote the set of nonnegative integers. For $e_1, e_2 \in \mathbb{Z}_+$ where $e_2 \leq e_1$, let $i_{e_1:e_2}$ denote the set $\{i_e : e_2 \leq e \leq e_1, e \in \mathbb{Z}_+\}$. For the notations used in the problem formulation and algorithm, see Appendix B.

## 2  PROBLEM FORMULATION

Consider a general discrete-time dynamical system $x_{t+1} = f(x_t, u_t, w_t)$, $t = 0, \ldots, T-1$, where $x_t \in \mathbb{R}^n$ is the system state at time $t$, $u_t \in \mathbb{R}^m$ is the control input at time $t$ to be designed via an algorithm. $u_t$ is determined by selecting a controller from *a priori* finite number of controller pool consisting of $\pi_i : \mathbb{R}^n \to \mathbb{R}^m$, $i = 1, \ldots, N$. $w_t \in \mathcal{W} \subset \mathbb{R}^g$ is the adversarial noise at time $t$, where $\mathcal{W} = \{w \in \mathbb{R}^g : \|w\| \leq w_{\max}\}$ and the bounding constant $w_{\max} > 0$ is assumed to be known. Each time instance $t$ is associated with a cost function $c_t : \mathbb{R}^n \times \mathbb{R}^m \to \mathbb{R}$. The state transition is governed by the dynamic $f : \mathbb{R}^n \times \mathbb{R}^m \times \mathbb{R}^g \to \mathbb{R}$. We have the following assumptions on the dynamic $f$.

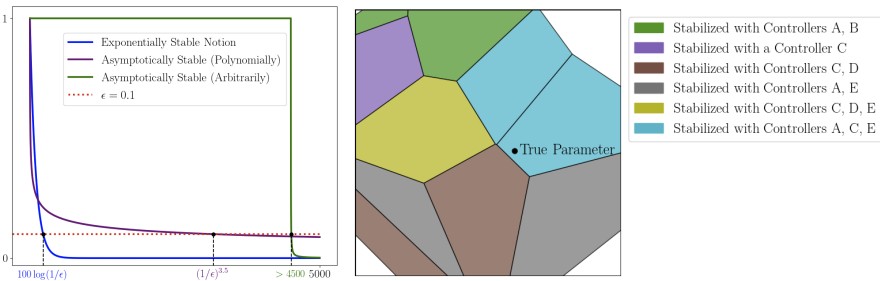

(a) Controller Stability notions      (b) Controller pool for unknown nonlinear systems

Figure 1: Illustration of Assumption 2.5: (a) Our work does not require exponentially stabilizing controllers, which allow the learner to detect the stability in $O(\log(1/\epsilon))$ time. Instead, we only require an asymptotically stabilizing controller of the true system, where the detectable time may be arbitrarily long. (b) One can design stabilizing controllers for each parameter characterizing the nonlinear system. While we know that at least one of them should work, we do not know which one works, since the learner is unaware of the true parameter of the system. Assumption 2.5 is always satisfied if the given pool contains a rich set of controllers, as long as the true system is stabilizable.

**Assumption 2.1** (Dynamic). The transition dynamic $f$ is $L_f$-Lipschitz continuous with $L_f \geq 1$; *i.e.*, $|f(x, u, w) - f(\tilde{x}, \tilde{u}, \tilde{w})| \leq L_f(\|x - \tilde{x}\| + \|u - \tilde{u}\| + \|w - \tilde{w}\|)$ for all $x, \tilde{x} \in \mathbb{R}^n$, $u, \tilde{u} \in \mathbb{R}^m$, $w, \tilde{w} \in \mathcal{W}$. We let $f(0, 0, 0) = f_0$.

We adopt the notion of locally Lipschitz continuous cost functions $c_t$ given in Li et al. (2023), which contains quadratic tracking costs along an arbitrary bounded state trajectory and action sequence.

**Assumption 2.2** (Cost functions). There exist $L_{c1}, L_{c2} > 0$ such that $|c_t(x, u) - c_t(\tilde{x}, \tilde{u})| \leq (L_{c1}(\max\{\|x\|, \|\tilde{x}\|\} + \max\{\|u\|, \|\tilde{u}\|\}) + L_{c2})(\|x - \tilde{x}\| + \|u - \tilde{u}\|)$ for all $x, \tilde{x} \in \mathbb{R}^n, u, \tilde{u} \in \mathbb{R}^m, t \in \mathbb{Z}_+$. There exists $c_{0,\max} \geq 0$ such that $|c_t(0, 0)| \leq c_{0,\max}$ for all $t \in \mathbb{Z}_+$.

Input-to-state (asymptotic) stability (ISS) is a classic notion of stability implying that the controller successfully stabilizes the system under any bounded noises (Sontag, 2008; Khalil, 2015). Incremental (asymptotic) stability extends the input-to-state stability to describe the asymptotic behavior of some trajectory towards a different trajectory (Tran et al., 2016). It is worth noting that Li et al. (2023) also adopted these concepts under an exponential stability assumption; *i.e.*, they require some controllers to satisfy exponential ISS and exponential incremental stability. However, in practice, general asymptotic concepts need to be considered for stabilizing controllers. We will address this *controller stability* issue below.

**Definition 2.3** (Input-to-state stable controller). A controller $\pi$ is (asymptotically) input-to-state stable (ISS) if there exists a non-increasing function $\beta(\cdot) : \mathbb{Z}_+ \to \mathbb{R}$ that satisfies $\beta(0) = 1^2$ with $\lim_{t \to \infty} \beta(t) = 0$ and $\gamma > 0$ such that for any $x_0 \in \mathbb{R}^n$ and $\|w_t\| \leq w_{\max}$ for all $t \geq 0$, the sequence $\{x_t\}_{t \geq 0}$ determined by $x_{t+1} = f(x_t, \pi(x_t), w_t)$ satisfies $\|x_t\| \leq \beta(t)\|x_0\| + \gamma w_{\max}$.

**Definition 2.4** (Incrementally stable controller). A controller $\pi$ is (asymptotically) incrementally stable if there exists a non-increasing function $\beta(\cdot) : \mathbb{Z}_+ \to \mathbb{R}$ that satisfies $\beta(0) = 1$ with $\lim_{t \to \infty} \beta(t) = 0$ such that for any $x_0, \tilde{x}_0 \in \mathbb{R}^n$ and $\|w_t\| \leq w_{\max}$ for all $t \geq 0$, it holds that $\|x_t - \tilde{x}_t\| \leq \beta(t)\|x_0 - \tilde{x}_0\|$ for any two sequences determined by $x_{t+1} = f(x_t, \pi(x_t), w_t)$ and $\tilde{x}_{t+1} = f(\tilde{x}_t, \pi(\tilde{x}_t), w_t)$.

**Assumption 2.5** (Controller pool). Consider the candidate controller index set $\mathcal{P}_0 = \{1, \ldots, N\}$, in which there exists a controller satisfying Definitions 2.3 and 2.4. There exists $\pi_{0,\max} \geq 0$ such that $\|\pi_i(0)\| \leq \pi_{0,\max}$ for all $i \in \mathcal{P}_0$. All candidate controllers are $L_\pi$-Lipschitz continuous; *i.e.*, $\|\pi_i(x) - \pi_i(\tilde{x})\| \leq L_\pi\|x - \tilde{x}\|$ for all $x, \tilde{x} \in \mathbb{R}^n$ and $i \in \mathcal{P}_0$.

In Figure 2, we illustrate a concept of the controller pool for the unknown system, and how general the requirement of asymptotically stable notion is. For future use, we define the relevant sets regarding controller stability below.

---

[2]This assumption in Definitions 2.3 and 2.4 is to guarantee $\beta(t)^2 \leq \beta(t)$ for all $t$, which can be overcome by a large $\gamma$. If we relax Assumption 2.2 on $c_t$ to be Lipschitz continuous, we can remove the assumption $\beta(0) = 1$.

**Definition 2.6** (Stabilizing and destabilizing controller). Let $\mathcal{S}$ denote an index set of stabilizing controllers that satisfy both of Definitions 2.3 and 2.4. We also let $\mathcal{U}$ denote an index set of destabilizing controllers that do not satisfy Definition 2.3. Thus, we have $|\mathcal{S}| \geq 1$ and $\mathcal{S} \subseteq \mathcal{U}^c$.

**Remark 2.7.** Definition 2.4 is a stronger notion than Definition 2.3 due to the triangle inequality. However, for a special case of linear systems with additive noise; *i.e.*, $f(x_t, \pi(x_t), w_t) = Ax_t + h(w_t)$, where $A \in \mathbb{R}^{n \times n}$ and $h : \mathbb{R}^g \to \mathbb{R}^n$, a controller $\pi$ satisfying Definition 2.3 also satisfies Definition 2.4. In such a case, Assumption 2.5 boils down to requiring at least one ISS controller in the pool.

Now, we define different notions of closed-loop *system stability* with bounded adversarial disturbances $w_t$, where $\|w_t\| \leq w_{\max}$ holds. Asymptotic stability and finite-gain stability both shed light on the connection between the disturbance input and the state output, where none of them implies the other (Hill & Moylan, 1980). Hence, it is desirable to achieve both system stability notions.

**Definition 2.8** (Asymptotic stability). A system is asymptotically stable if the sum of state norms satisfies $\lim_{T \to \infty} \frac{1}{T} \sum_{t=0}^{T} \|x_t\| \leq \gamma w_{\max}$.

**Definition 2.9** (Finite-gain stability). A system is finite-gain $\mathcal{L}_1$ stable if there exist constants $A_1, A_2 > 0$ such that for all $T \in \mathbb{Z}_+$, it holds that $\sum_{t=0}^{T} \|x_t\| \leq A_1 \cdot w_{\max} T + A_2$.

Recall that $x_t$ and $u_t$ denote the state and action sequence for the system according to the algorithm. We also let $x_t^*$ and $u_t^*$ denote the optimal state and action sequence generated by the best stabilizing controller $i^*$ that satisfies both of Definitions 2.3 and 2.4; *i.e.*, $i^* = \arg\min_{i \in \mathcal{S}} \mathbb{E}[\sum_{t=0}^{T} c_t(x_t, \pi_i(x_t))]$ subject to the dynamic $f$. Then, the regret of the algorithm is defined as follows.

**Definition 2.10** (Regret). The regret of the algorithm implementing the policy $\pi_{i_t}$ at time $t = 0, \ldots, T-1$ is defined as $Regret_T = \mathbb{E}_{i_{T-1:0}} \sum_{t=0}^{T}[c_t(x_t, u_t) - c_t(x_t^*, u_t^*)]$.

## 3 ALGORITHM DESCRIPTION

Denote the number of batches in the algorithm by $B$. Denote by $t_b$ the start time for each batch $b = 0, 1, \ldots, B-1$. We implement the same policy within the mini-batch.

**Assumption 3.1** (Dynamic batch length). We design our batch length $(\tau_b)_{b \geq 0}$ as follows:

1. $\tau_b$ is non-decreasing in $b$ and $\lim_{b \to \infty} \tau_b = +\infty$.

2. $\max_{b \geq 0} \frac{\tau_{b+1}}{\tau_b} = \frac{\tau_1}{\tau_0}$ and $\lim_{b \to \infty} \frac{\tau_{b+1}}{\tau_b} = 1$.

For example, $\tau_0 = \lfloor z_1(z_2)^{z_3} \rfloor > 0$ and $\tau_b = \lceil z_1(\nu b + z_2)^{z_3} \rceil$ for every $b \geq 1$ with the constants $z_1, z_2, z_3, \nu > 0$ satisfy Assumption 3.1. For future use, we refer to this type of formulation as polynomial batches with $(z_1, z_2, z_3, \nu)$.

**Remark 3.2.** As our dynamic batch length eventually grows unboundedly over time, excessively strict controller stability criteria may result in most of the candidate controllers violating these criteria. Thus, it is crucial to adopt (asymptotic) ISS and incremental stability as our criteria, instead of exponential notions in Li et al. (2023) and the literature on linear dynamics (Cassel & Koren, 2020; Chen & Hazan, 2021; Ghai et al., 2023). Figure 4 in Appendix A strongly supports the necessity of a growing batch length regardless of the noise assumption. On the other hand, our batch length requires $\lim_{b \to \infty} \frac{\tau_{b+1}}{\tau_b} = 1$, which means the ratio of two consecutive batch lengths should approach 1 as time goes by (*e.g.*, geometric sequences are not acceptable). In other words, the batch length is designed to *increase* over time but eventually *saturates*, which is used to ensure both asymptotic system stability and a sublinear regret. We formally present both properties in Theorems 4.1 and 4.6.

We propose our DBAR algorithm in Algorithm 1 (see Appendix B for the notations). Lines 3-9 generate the state trajectory based on the selected controller $\pi_{K_b}$ for the current batch $b$, and falsify the controller if it is found to violate Definition 2.3; *i.e.*, $K_b \in \mathcal{U}$. Here, let $U$ denote the number of times that the Break statement in Line 7 is activated. In the rest of the paper, when we say the Break statement is activated, it means that Line 7 of Algorithm 1 has been activated. As the controllers in $\mathcal{U}^c$ do not suffer from the Break statement, they always remain in the controller pool. Accordingly, we have $U \leq |\mathcal{U}|$.

Lines 11-20 keep track of the state norm of $x_{b+1}$ by determining $\alpha_{b+1}$ and $s_{b+1}$ that indicates the magnitude of the next batch's initial state norm compared to $\|x_0\|$. Note that we keep adjusting the

---

**Algorithm 1** DBAR

---

**Input:** $T$. $\eta_0 > 0$. $(\tau_b)_{b \geq 0}$. $\beta(\cdot)$. $\gamma$. $W_0(k) = 0$ for all $k \in \mathcal{P}_0$. $t_0 = 0, s_0 = 0$.
A uniform distribution $p_0$; *i.e.*, $p_0(k) = \frac{1}{N}$ for all $k \in \mathcal{P}_0$. $x_0 \neq 0$. $\alpha_0 > \beta(0) = 1$. $\delta \geq \frac{\gamma w_{\max}}{1 - \beta(\tau_0)}$.

---

1: **for** Batch $b = 0, 1, 2, \ldots,$ **do**
2:     Sample $K_b$ from a distribution $p_b$. Terminate the algorithm if $\mathcal{P}_b$ is empty.
    // Phase 1: Falsify a detected destabilizing controller
3:     **for** $t = t_b, \ldots, \min(t_b + \tau_b - 1, T)$ **do**
4:       Implement $\pi_{K_b}$, observe $x_{t+1}$.
5:       **if** $\|x_{t+1}\| > \beta(t + 1 - t_b)\|x_{t_b}\| + \gamma w_{\max}$ **then**
6:         Set $\mathcal{P}_{b+1} = \mathcal{P}_b - \{K_b\}$.
7:         **Break**
8:       **end if**
9:     **end for**
10:     Let $t_{b+1} = t + 1$.
    // Record the magnitude of the state norm for Phase 2
11:     **if** $\|x_{t_{b+1}}\| \geq \alpha_b\|x_0\| + \delta$ **then**
12:       Pick $s \geq 1$ that satisfies
      $(\alpha_b)^s\|x_0\| \leq \|x_{t_{b+1}}\| - \delta < (\alpha_b)^{s+1}\|x_0\|$.
13:       **if** $s - s_b > 1$ **then**
14:         Let $\alpha_{b+1}$ be any $\alpha > \alpha_b$ such that
        $\alpha^{s_b+1}\|x_0\| \leq \|x_{t_{b+1}}\| - \delta < \alpha^{s_b+2}\|x_0\|$
        and let $s_{b+1} = s_b + 1$.

15:       **else**
16:         Let $s_{b+1} = s$ and let $\alpha_{b+1} = \alpha_b$.
17:       **end if**
18:     **else**
19:       Let $s_{b+1} = 0$ and let $\alpha_{b+1} = \alpha_b$.
20:     **end if**
    // Phase 2: Set or reset weight for each controller
21:     Let $w_b(K_b) = \sum_{t=t_b}^{t_{b+1}-1} c_t(x_t, u_t)$
    and $w_b'(k) = \frac{w_b(K_b)}{p_b(k)} \mathcal{I}_{(K_b=k)}$ for $k \in \mathcal{P}_b$.
22:     **if** $s_{b+1} \neq s_b$ **then**
23:       Let $W_{b+1}(k) = 0$ for all $k \in \mathcal{P}_b$.
24:     **else**
25:       Let $W_{b+1}(k) = W_b(k) + w_b'(k)$ for $k \in \mathcal{P}_b$.
26:     **end if**
27:     Let $\eta_{b+1} = \eta_0/(\alpha_{b+1})^{2s_{b+1}}$.
28:     For all $k \in \mathcal{P}_{b+1}$, let
    $p_{b+1}(k) = \frac{\exp(-\eta_{b+1} W_{b+1}(k))}{\sum_{i \in \mathcal{P}_{b+1}} \exp(-\eta_{b+1} W_{b+1}(i))}$

29: **end for**

---

value of $\alpha_{b+1}$ to avoid $s_{b+1} > s_b + 1$ (Line 14), and the adjusted $\alpha_{b+1}$ is guaranteed to be bounded by some constant (see Lemma C.5 in the Appendix). It is later discussed formally in Lemma 4.7 that these observations cause $s_b \neq 0$ to occur at most $O(U)$ times throughout the algorithm.

Lines 21-26 determine the weight $W_{b+1}(k)$ for each controller $k$. In Line 21, we use the sum of costs at the current batch $b$ to add up to the weight in Line 25. In Lines 22-26, we reset the weight if $s_{b+1} \neq s_b$. This resetting weight idea to forget the costs in the past is also proposed in van Erven et al. (2011). In the scenario that the Lipschitz constant $L_f$ is very large, it may help to forget the time-varying costs $c_0, \ldots, c_{t-1}$ and restart gathering the information from the outset. Line 22 reflects this case where the next batch's state norm significantly deviates from the current state norm.

Lines 27-29 calculate the adaptive learning rate $\eta_{b+1} = \eta_0/(\alpha_{b+1})^{s_{b+1}}$ for the next batch $b + 1$ used to apply the Exp3 algorithm to our problem. Since $(\alpha_{b+1})^{s_{b+1}}$ increases when the state norm $\|x_{t_{b+1}}\|$ is large, and $s_{b+1}$ resets to zero for sufficiently small state norm, the corresponding learning rate decreases in unstable states and increases back to the initial value when the state norm returns to a stable region. Thus, the learning rate fluctuates depending on the state norm. However, it is essential to note that the *effective* learning rate, determined by the ratio $\frac{\eta_b}{\tau_b}$, indeed decreases as the batch length increases even if $s_{b+1} = s_b$. The only plausible situation in which the effective rate may increase is $s_{b+1} < s_b$ with $(\alpha_{b+1})^2 > \frac{\tau_{b+1}}{\tau_b}$. Apart from this scenario, the effective learning rate experiences a polynomial decay with polynomial batches defined in Assumption 3.1, which does not cause any contradiction with the polynomially decreasing learning rate concept proposed in Aubert et al. (2023).

Our adaptive learning rate stabilizes the cost of current batch, alleviating the multiplicative exponential term in the regret bound (see Table 1). Moreover, since we run the algorithm along a single trajectory with the selection of the policy only relying on the state norm as a context, we obtain a linear-time algorithm by harnessing a form of contextual bandit without requiring strict assumptions.

# 4 MAIN RESULTS

## 4.1 STABILITY

In this section, we will present the stability results of Algorithm 1, which deeply hinge on Lemma 4.3 (see the proof details in Lemma C.1).

**Theorem 4.1** (Asymptotic stability)**.** *In Algorithm 1, suppose that $\frac{\tau_1}{\tau_0}\beta(\tau_0) < 1$. Then, it holds that*

$$\lim_{T \to \infty} \frac{1}{T} \sum_{t=0}^{T} \|x_t\| \leq \gamma w_{max}.$$

**Theorem 4.2** (Finite-gain stability). *In Algorithm 1, suppose that $\frac{\tau_1}{\tau_0}\beta(\tau_0) < 1$. Assume that $\lim_{t\to\infty} H(t) < \infty$. Then, Algorithm 1 achieves finite-gain $\mathcal{L}_1$ stability; i.e., there exist constants $A_1, A_2 > 0$ such that for all $T \in \mathbb{Z}_+$,*

$$\sum_{t=0}^{T} \|x_t\| \le A_1 \cdot w_{max} T + A_2.$$

**Lemma 4.3.** *Define $H(t) := \sum_{i=0}^{t-1} \beta(i)$, which determines the scope of stabilizing controllers throughout the entire horizon. Under Assumption 3.1, we have $\lim_{t\to\infty} \frac{H(t)}{t} = 0$.*

*Proof sketch of Theorems 4.1 and 4.2:* By Lemma 4.3, we have $\lim_{t\to\infty} \frac{H(t)}{t} = 0$. Using this result with the non-decreasing property of both $\tau_b$ and $H(\tau_b)$, we obtain that $\sum_{b=0}^{B-1} H(\tau_b) = o(T)$ according to Assumption 3.1 for the dynamic batch length. This assumption further indicates that falsifying destabilizing controllers in Lines 5-8 results in the existence of a constant $M > 0$ such that the following inequality holds for all $T \ge 0$:

$$\sum_{t=0}^{T} \|x_t\| \le M + \gamma w_{\max} \cdot (O(\sum_{b=0}^{B-1} H(\tau_b)) + T). \tag{2}$$

Thus, $\sum_{b=0}^{B-1} H(\tau_b) = o(T)$ along with (2) proves both Theorems 4.1 and 4.2. More details about the proof are provided in Appendix C. □

**Remark 4.4.** With a fixed batch length $\tau$ as presented in Li et al. (2023), the resulting closed-loop system cannot achieve asymptotic stability since $\lim_{T\to\infty} \frac{1}{T}\sum_{t=0}^{T} \|x_t\| = \gamma w_{\max}(1 + O(\frac{1}{\tau})) > \gamma w_{\max}$. Thus, it is intuitively desirable to design as $\lim_{b\to\infty} \tau_b = \infty$ to achieve an asymptotic system stability, validating our dynamic batch length strategy in Algorithm 1. This idea also results in having $\lim_{T\to\infty} B/T = 0$ (see Lemma C.9 in the Appendix). It is crucial to note that we have achieved asymptotic stability even when $\lim_{t\to\infty} H(t) = \infty$. In addition, finite-gain stability can be achieved for every $\beta(\cdot)$ that satisfies $H(\cdot) < \infty$, which incorporates exponentially stabilizing controllers.

## 4.2 REGRET

In this section, we will present the regret bound of Algorithm 1, where the regret defined in Definition 2.10 is equivalent to $\mathbb{E}_{K_{B-1:0}} \sum_{t=0}^{T}[c_t(x_t, u_t) - c_t(x_t^*, u_t^*)]$, considering that the policy at each time $t$ is determined by the policy at the corresponding batch.

**Theorem 4.5** (Regret Bound). *In Algorithm 1, suppose that $\frac{\tau_1}{\tau_0}(\beta(\tau_0))^2 < \frac{1}{2\sqrt{2}}$. Then, we have*

$$Regret_T = O(|\mathcal{U}|) + O(\sum_{b=0}^{B-1} H(\tau_b)) + \frac{\tilde{O}(|\mathcal{U}|+1)}{\eta_0} + \frac{\eta_0 N}{2}[\exp(O(|\mathcal{U}|))O(\tau_{B-1}H(\tau_{B-1})) + O(\sum_{b=0}^{B-1}(\tau_b)^2)].$$

**Theorem 4.6** (Regret bound with known $|\mathcal{U}|$). *Consider Algorithm 1 with polynomial batches defined in Assumption 3.1 with proper parameters satisfying $(\beta(\tau_0))^2 < \frac{1}{2\sqrt{2}}$. Then, with $\eta_0 = O(\frac{(|\mathcal{U}|+1)^{2/3}}{T^{2/3}N^{1/3}})$ and $T \ge \max\{\frac{|\mathcal{U}|^{3/2}}{(N(|\mathcal{U}|+1))^{1/2}}, N(|\mathcal{U}|+1)\}$, we achieve a **sublinear** regret bound. Moreover[3], when $H(t) \le O(\sum_{i=1}^{t} \frac{1}{i})$ for all $t \ge 1$, we have*

$$Regret_T = [\tilde{O}(T^{2/3}) + \tilde{O}(T^{-1/3})\exp(O(|\mathcal{U}|))]N^{1/3}(|\mathcal{U}|+1)^{1/3}.$$

The regret bound deeply relies on Lemma 4.7. For the lemma, define $\mathcal{L} := \{0 \le b \le B-1, b \in \mathbb{Z}_+ : s_{b+1} \ne s_b\}$ and Also, define $\mathcal{V} := \{0 \le b \le B-1, b \in \mathbb{Z}_+ : s_b \ne 0\}$. In other words, $|\mathcal{L}|$ is the number of transitions of $s_b$ across the batches, and $|\mathcal{V}|$ is the number of batches whose $s_b$ is nonzero. It turns out that both quantities are bounded in terms of the number of the Break statement activation. The proof details can be found in Lemma D.3.

**Lemma 4.7.** *In Algorithm 1, suppose that $\beta(\tau_0) < 1$ and let $U$ denote the number of times that the Break statement is activated. Then, it holds that $|\mathcal{L}| = O(U)$ and $|\mathcal{V}| = O(U)$.*

---

[3]Among stabilizing controllers achieving $\tilde{O}(T^{2/3})$ regret bound, we also cover the case where $H(t)$ can be of the order of a harmonic series that is not summable at infinity.

*Proof sketch of Theorems 4.5 and 4.6:* By adopting the analysis performed in previous works (Cesa-Bianchi & Lugosi, 2006; van Erven et al., 2011; de Rooij et al., 2014), we divide the expected total cost into the mix loss $-\frac{1}{\eta_0} \log(\mathbb{E}_{k \sim p_b} \exp(-\eta_b w_b'(k)))$ and the mixability gap $\mathbb{E}_{k \sim p_b}[w_b'(k)] + \frac{1}{\eta_0} \log(\mathbb{E}_{k \sim p_b} \exp(-\eta_b w_b'(k)))$. The big difference between the previous analysis and our approach is that we use different learning rates for the one in the denominator ($\eta_0$) and the other inside the exponential term ($\eta_b$) as we use an adaptive learning rate. The additional term introduced by using different rates is in $|\mathcal{L}|$ and $|\mathcal{V}|$, which are bounded in terms of $U$ by Lemma 4.7.

After bounding the expected total cost with cumulative mix loss and mixability gap, we need to study $\mathbb{E}_{K_{B-1:0}} \sum_{b=0}^{B-1} \sum_{t=t_b}^{t_{b+1}-1} \left[ \frac{c_t(x_t^K(i^*), u_t^K(i^*))}{(\alpha_b)^{2s_b}} - c_t(x_t^*, u_t^*) \right]$, where $x_t^K(i)$ and $u_t^K(i)$ for $t = t_b, \ldots, t_{b+1}-1$ denote the state and action sequence generated by selecting the controllers before batch $b$ according to Algorithm 1, while selecting the controller $i$ at batch $b$. This does not produce any exponential term since the costs are regularized with the factor $(\alpha_b)^{2s_b}$. The additional term introduced by regularization is also bounded by the order of $U$ due to Lemma 4.7. The proof details are provided in Appendix D. □

**Remark 4.8** (Lower bound). The regret bound $\tilde{O}(T^{2/3} N^{1/3} (|\mathcal{U}| + 1)^{1/3})$ provided in Theorem 4.6 is similar to the *lower bound* presented in Dekel et al. (2014), except that there is an extra term $(|\mathcal{U}| + 1)^{1/3}$, reflecting the unbounded costs for the bandits. Moreover, a stability-agnostic nature of the given controllers implies that any algorithm will normally encounter destabilizing controllers and it is unavoidable to face the exponential term $\exp(O(|\mathcal{U}|))$ in regret. To be more specific, our work has an exponential term in the number of destabilizing controllers ($|\mathcal{U}|$), while the work Chen & Hazan (2021) provides the *lower bound* involving an exponential term in $L > k d_u$ (see Section 2.1 and Theorem 3), where $d_u$ is the dimension of the action and $k$ is the controllability index. Here, a large controllability index implies that the system is complex to control as more stages of control actions are needed to stabilize the system. Thus, together with a dimension of the controller action $d_u$, a large $k d_u$ in their work is analogous to a large $|\mathcal{U}|$ in our setting. Thus, due to the *lower bound*, the exponentially increasing term can be tackled by reducing it by the inverse power term on $T$ at best. Theorem 4.6 aligns with this idea since the resulting regret bound involves the term $\tilde{O}(T^{-1/3}) \cdot \exp(O(|\mathcal{U}|))$ by factoring in every potential exponential term to be multiplied with the initial learning rate $\eta_0 = O(T^{-2/3})$, which inherently serves as a mitigating factor. Note that instead of dramatically reducing the regret bound, our main contribution is on significantly relaxing the stability assumptions for required controllers (see Table 1 and Appendix A).

**Remark 4.9** (Nonlinear control). Our approach is useful to extend the stability and regret analysis beyond linear dynamics, but if $|\mathcal{U}|$ is too large, it would be difficult to reach good enough performance as the regret bound depends on $\exp(O(|\mathcal{U}|))$. This occurs because we have focused on a discrete set of controllers instead of a connected set as in linear dynamics. Note that in the linear dynamics case, it is guaranteed that the set of stabilizing controllers is connected. However, adopting a discrete set was inevitable to handle unknown nonlinear systems since the set of stabilizing controllers may not be connected. To address this limitation, we believe that this issue can be mitigated by the formulation where the problem of interest is $|\mathcal{U}|$ number of connected sets, where $|\mathcal{U}|$ is not too large and each set is disjoint from the others. The agent can apply techniques of continuous parameterization (*e.g.* gradient descent) within a set and also transition between separate sets by leveraging our technique. This mixture of algorithms for discrete and connected sets will be an interesting future work.

Now, a question arises as to what happens if $|\mathcal{U}|$ is *not known* in advance. With Algorithm 1, one can leverage $|\mathcal{U}| + 1 \leq N$ to upper-bound the regret in Theorem 4.6 and achieve $\tilde{O}(T^{2/3} N^{2/3})$ at best (without considering exponential terms) by determining $\eta_0$ and $(\tau_b)_{b \geq 0}$ as if there were only one stabilizing controller. It turns out that we can reduce the bound to $\tilde{O}(T^{2/3} N^{1/3} (|\mathcal{U}| + 1)^{1/2})$ by adaptively changing the value of $\eta_b$ as in Algorithm 2, where we increase the value of $\mu_b$ if the Break statement in Algorithm 1 is activated and keep it unchanged otherwise.

**Theorem 4.10** (Regret bound with unknown $|\mathcal{U}|$). *Consider Algorithm 2 with polynomial batches defined in Assumption 3.1 with proper parameters satisfying $\frac{\tau_1}{\tau_0}(\beta(\tau_0))^2 < \frac{1}{2\sqrt{2}}$. Then, with $y = \frac{1}{2}$, $\eta_0 = O(\frac{1}{T^{2/3} N^{1/3}})$, and $T \geq \max\{\frac{|\mathcal{U}|^{3/2}}{N^{1/2}(|\mathcal{U}|+1)^{3/4}}, N\}$, we achieve a **sublinear** regret bound. Moreover, when $H(t) \leq O(\sum_{i=1}^{t} \frac{1}{i})$ for all $t \geq 1$, we have*

$$Regret_T = \left[ \tilde{O}(T^{2/3}) + \tilde{O}(T^{-1/3}) \exp(O(|\mathcal{U}|)) \right] N^{1/3} (|\mathcal{U}| + 1)^{1/2}.$$

---

**Algorithm 2** DBAR-unknown $|\mathcal{U}|$

---

**Input:** Add two more inputs $\mu_0 = 0$. $y > 0$.

// Modification 1: Add the following IF-ELSE Statement right after Line 9 in Algorithm 1.

**if** $\mathcal{P}_{b+1} = \mathcal{P}_b$ **then** $\mu_{b+1} = \mu_b$. **else** $\mu_{b+1} = \mu_b + 1$. **end if**

// Modification 2: Incorporate $\mu_{b+1}$ to set $\eta_{b+1}$ in Line 27 in Algorithm 1.

$\eta_{b+1} = \frac{\eta_0 (\mu_{b+1}+1)^y}{(\alpha_{b+1})^{2s_{b+1}}}$.

---

*Proof sketch:* Define $\eta_{0,r} := \eta_0 \sqrt{r+1}$. It turns out that for every $r = 0, \ldots, U$, $\tilde{O}(\frac{1}{\eta_{0,r}})$ appears in the regret instead of the integrated term $\tilde{O}(\frac{|\mathcal{U}|+1}{\eta_0})$ in Theorem 4.5. The constant $|\mathcal{U}|+1$ is distributed among each $\tilde{O}(\frac{1}{\eta_{0,r}})$ term. Under the constraints given by the disintegration rule using Lemma 4.7 for each $r$, one can establish an upper bound of $\tilde{O}(\frac{(|\mathcal{U}|+1)^{1/2}}{\eta_0})$ on the sum of $\tilde{O}(\frac{1}{\eta_{0,r}})$ terms over $r = 0, \ldots, U$ by attaining the coefficients of these terms with complementary slackness in Karush-Kuhn-Tucker (KKT) conditions. The details are available in Appendix E. $\square$

Our DBAR algorithm can also be applied to scenarios such as those switched systems (Tousi et al., 2008; Zhao et al., 2022) in which the transition dynamics and the associated controller pool change according to either the detection of a destabilizing controller or pre-determined time instants (Battistelli et al., 2011), as well as the ballooning problem (Ghalme et al., 2021) where the controller pool may expand. We proposed Algorithm 3, the switching version of DBAR, in Appendix F.

## 5 NUMERICAL EXPERIMENTS

To demonstrate the main results of this paper, we provide illustrative examples on both linear and nonlinear dynamics with adversarial disturbances.

*Example 1:* Consider the following linear dynamical system with $x_t \in \mathbb{R}^2$ and $u_t \in \mathbb{R}^2$:

$$x_{t+1} = \begin{bmatrix} 2 & 1.2 \\ 1.1 & 2.5 \end{bmatrix} x_t + \begin{bmatrix} 1 & 0.3 \\ 0.4 & 0.9 \end{bmatrix} u_t + w_t, \quad t = 0, 1, \ldots, \tag{3}$$

where $x_0 = [100, 200]'$ and $w_t = [\sin(\frac{t}{5\pi}), \sin(\frac{t}{11\pi})]'$. We consider a linear policy $u_t = K x_t = \begin{bmatrix} k_1 & k_2 \\ k_3 & k_4 \end{bmatrix} x_t$ and a controller pool $K' = \{K \in \mathbb{R}^{2 \times 2} : k_1, k_3, k_4 \in \{-3, -2, -1\}, k_2 \in \{-1, 0, 1\}\}$ that has $|\mathcal{U}| = 53$ out of 81 candidate controllers. The goal is to keep the state near the origin, where the cost function is quadratic at each time, namely $c_t(x_t, u_t) = \|x_t\|^2$.

Falsifying destabilizing controllers moderately stabilizes the state norm (Li et al., 2023). Compared to their work, Figures 2(a) and 2(b) show that both integral components of our algorithm DBAR, dynamic batch length and adaptive learning rate, further lowers the regret and stabilizes the system, where approximately 2/3 of controllers in $K'$ are destabilizing the system. In this case, Figures 2(c) and 2(d) both demonstrate that the two components of our algorithm mutually reinforce each other, where each component stabilizes the state norm with or without time delay. This supports the observations in Appendix A. In Appendix G.1, we also provide the experiment details and simulation results with noise terms generated by uniform random walk, where $w_t - w_{t-1}$ has a uniform distribution for $t \geq 1$, as well as the results with truncated Gaussian noise for sanity check.

*Example 2:* Consider the following nonlinear noise-injected ball-beam system (Hauser et al., 1992):

$$\ddot{x} = B(x\dot{\theta}^2 - 9.81 \sin \theta) + 3w_x, \quad \ddot{\theta} = u_x, \quad B = 0.7143, \tag{4}$$

where $x$ is the ball position, $\theta$ is the beam angle, $u_x$ is the action, and $w_x(t) = \sin(\frac{t}{7\pi})$. To provide the simulations for high-dimensional systems, we consider the leader-follower system (Morbidi et al., 2011), where the leader is represented by a ball-beam system, and the followers leverage the leader's state to stabilize themselves. Specifically, if the leader is controlled by destabilizing controllers, the followers may also fail to stabilize. Consider the followers' system:

$$\dot{z} = A[x, \; \dot{x}, \; -9.81B\theta, \; -9.81B\dot{\theta}]' + \tilde{A}z + u_z + 3w_z, \tag{5}$$

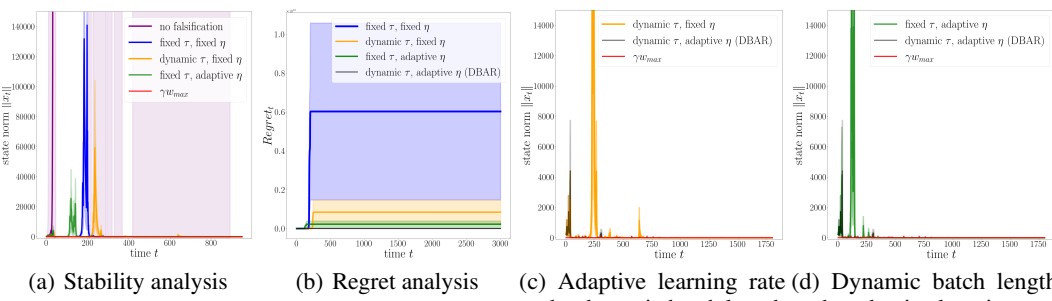

(a) Stability analysis  (b) Regret analysis  (c) Adaptive learning rate under dynamic batch length  (d) Dynamic batch length under adaptive learning rate

Figure 2: The stability and the regret in the linear system under sinusoidal noise. *Fixed $\tau$, fixed $\eta$* represents the algorithm in Li et al. (2023). Ablation study of the algorithm is presented.

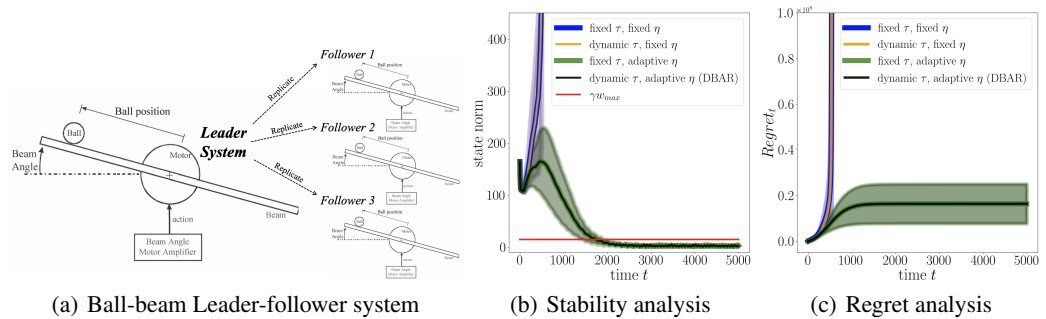

(a) Ball-beam Leader-follower system  (b) Stability analysis  (c) Regret analysis

Figure 3: The stability and the regret in the leader-follower system under sinusoidal noise, where the leader is represented by a ball-beam system. We selected $\beta(t) = \min\{10/t^{1.05}, 1\}$ (see Definition 2.3) and used squared sum of state and action norms as the cost.

where $[x, \dot{x}, \theta, \dot{\theta}] \in \mathbb{R}^4$ are the states of the leader given in (4), $z \in \mathbb{R}^{96}$ are the states of the followers, $u_z \in \mathbb{R}^{96}$ is the action of the followers, $w_z = [\sin\left(\frac{t}{5\pi}\right), \sin\left(\frac{t}{11\pi}\right), \sin\left(\frac{t}{5\pi}\right), \sin\left(\frac{t}{11\pi}\right), \dots] \in \mathbb{R}^{96}$, and $A, \tilde{A}$ are relevant random matrices. Note that the number of states in the entire system is 100.

For the action $u_x$, we now adopt a broader notion of stabilizing controllers and choose the policy class to be the nested saturating control (Teel, 1992), without considering exponentially stabilizing notions. For the action $u_z$, we consider a linear policy in $z$; however, the policy is inherently nonlinear with respect to the entire state, as the leader's system itself is nonlinear. In Figures 3(b) and 3(c), we observe that dynamic batching does not necessarily stabilize the state norm by itself. However, if an adaptive learning rate is additionally applied, DBAR effectively stabilizes the explosion of the nonlinear system and enjoys the improved regret, even when we use a polynomially stabilizing criterion $O(1/t^{1.05})$ to define the stabilizing controllers (see Definition 2.3). We also provide the simulation results with the other polynomially decreasing $\beta(\cdot)$ series at a different rate. More experiment details are available in Appendix G.2.

## 6 CONCLUSION

In an online bandit nonlinear control problem, an agent makes decisions with the bandit feedback information, while suffering from nonlinear dynamics and adversarial disturbances. To address such challenges, this paper develops a novel Exp3-type algorithm with theoretical guarantees. The proposed algorithm uses a dynamic batch length to achieve asymptotic stability of the system without requiring an exponential assumption on stabilizing controllers in the pool. Our adaptive learning rate scheme observes the stability of state norm to overcome the inherent multiplicative exponential term in the regret, thereby improving the overall regret. Future directions include extending these results to problems with explicit safety constraints while selecting the best stabilizing controller among a continuum of candidate controllers.

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

# A  NECESSITY OF DBAR UNDER WEAKER STABILITY NOTION OF REQUIRED CONTROLLERS

To illustrate how significant the weaker controller stability notion is compared to the exponential notions, let us further present a one-dimensional system, where the current system state is 1. The goal is to achieve a state near 0, and we would like to detect this stability by observing whether one arrives at a state less than $1 - \epsilon$, where $\epsilon$ is an arbitrarily small positive number. Exponentially stabilizing controllers guarantee to detect the stability in $O(\log(1/\epsilon))$ time. However, with an asymptotically stabilizing controller, if the controller is designed to keep the system state unchanged for an arbitrarily long time $T$ and then collapse the state towards 0 afterward, one cannot detect the stability before time $T$ regardless of how small $\epsilon$ is. In such a case, even though the controller ultimately achieves the goal, it may take a lot of time to learn whether a closed-loop system would be stable or not.

Note that dynamic batch length is an important part of our work. If an exponentially stabilizing controller is applied to a system, one can quickly certify the stability. However, if we only have the asymptotically stabilizing controllers as in our problem setting, it may take a long time to observe any abnormal behavior in the closed-loop system. Such an issue cannot be handled by a fixed batch length and in that sense dynamic batch length is a necessary part of our work. In Table 1, we have stated the intermediate step "Dynamic Batching" to achieve closed-loop system asymptotic stability, which was not achievable by the previous works.

Figure 4 also demonstrates the necessity of a dynamic batch length regardless of the noise assumption. The blue and orange lines represent the state norms generated by a fixed batch length and a dynamic batch length, respectively. With both relatively easier statistical noise and more challenging adversarial noise, the blue line shows a larger state norm than the orange line. Moreover, the blue line occasionally has higher values than the red line, which is our asymptotic stability bound $\gamma w_{\max} = 1.5$, while the orange line remains below the red line after a certain time.

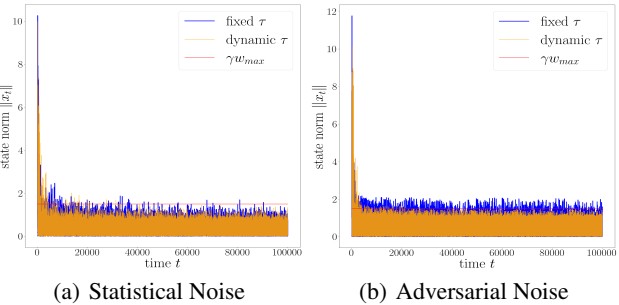

(a) Statistical Noise          (b) Adversarial Noise

Figure 4: The state norm with a fixed batch length compared to that with a dynamic batch length. $x_{t+1} = x_t + 0.15 u_t + w_t$ with $u_t = K x_t$ where $K \in [-3.0, -2.9, -2.8, \ldots, 4.9, 5.0]$. We use $\tau_0 = 10, \gamma = 3$, and set $w_{\max} = 0.5$. The noise $w_t$ is (a) i.i.d. sampled from Uniform$[-0.2, 0.5]$, and (b) $0.15 + 0.35 \sin(\frac{t}{3\pi})$.

However, it turns out that the resulting regret by dynamic batching contains the multiplicative term $o(T^{1/3}) \cdot \exp(O(|\mathcal{U}|))$, which is because a dynamic batch length induces $H(\tau_{B-1})$ to be necessarily multiplied with $\exp(O(|\mathcal{U}|))$. (see Corollary D.10). Thus, we came up with a careful switching strategy, an adaptive learning rate, to address this issue. The multiplicative term can be resolved with splitting technique by introducing an adaptive learning rate, achieving both closed-loop system asymptotic stability (by dynamic batch length) and the improved regret (by adaptive learning rate), even though we have greatly relaxed the assumption on controller stability (exponential to asymptotic). We developed this approach by factoring in every potential exponential term to be multiplied with the initial learning rate $\eta_0 = O(T^{-2/3})$, which has a negative exponent on $T$, thus inherently serving as a mitigating factor (see Theorem 4.6 and the term $\frac{\eta_0 N}{2} \sum_{b=0}^{B-1} \mathbb{E}_{K_{b-1:0}} (w_b^K(i^b))^2$ in Lemma D.5). Due to Lemma 4.7, one can explain that the remaining terms produced by the splitting can be bounded by $O(|\mathcal{U}|)$. More details can be found in Appendix D.

## B  GLOSSARY

Before formally presenting the proofs, we provide a glossary to help readers understand the notations of our algorithm DBAR (see Algorithm 1).

Table 2: Glossary

| Notation | Meaning |
|---|---|
| $x_t$ | state at time $t$ in the algorithm |
| $x_t^*$ | optimal state at time $t$ |
| $u_t$ | action at time $t$ in the algorithm |
| $u_t^*$ | optimal action at time $t$ |
| $c_t(x_t, u_t)$ | cost at time $t$ |
| $w_{\max}$ | the maximum norm of the noise |
| $T$ | the length of time in the algorithm |
| $B$ | the number of batches in the algorithm |
| $t_b$ | the start time for each batch $b$ |
| $\tau_b$ | the batch length at batch $b$ |
| $\eta_b$ | learning rate at batch $b$ |
| $K_b$ | the controller selected at batch $b$ |
| $N$ | the number of controllers in the candidate pool |
| $W_b(k)$ | the weight of controller $k$ at batch $b$ |
| $p_b(k)$ | the probability of selecting controller $k$ at batch $b$ |
| $P_b$ | a set of available controllers at batch $b$ |
| $\alpha_b, s_b$ | $(\alpha_b)^{s_b}$ indicates the magnitude of the state norm at $t_b$ compared to $\|x_0\|$ |
| $\beta(t), \gamma$ | applying a stabilizing controller incurs $\|x_t\| \leq \beta(t)\|x_0\| + \gamma w_{\max}$ |
| $L_f$ | Lipschitz constant for the dynamic $f$ |
| $L_\pi$ | Lipschitz constant for any controller $\pi$ |
| $U$ | the number of times the Break statement is activated |
| $b_1, \ldots, b_U$ | the next batch after the Break statement is activated |

## C  STABILITY PROOF

Let $b_1, \ldots, b_U$ denote the next batch after the Break statement is activated; *i.e.*, $\|x_{t_{b_u}}\| > \beta(t_{b_u} - t_{b_u-1})\|x_{t_{b_u-1}}\| + \gamma w_{\max}$ for every $u = 1, \ldots, U$. For future use, let $b_0 = 0$ and $b_{U+1} = B$. Accordingly, $t_{b_0} = t_0 = 0$ and $t_{b_{U+1}} = t_B = T + 1$.

**Lemma C.1** (Restatement of Lemma 4.3). *Define $H(t) := \sum_{i=0}^{t-1} \beta(i)$. Under Assumption 3.1, we have*

$$\lim_{t \to \infty} \frac{H(t)}{t} = 0.$$

*Proof.* Recall that we designed $\beta(\cdot)$ to be non-increasing and nonnegative. Then, we have $\beta(i) \leq \int_{i-1}^{i} \beta(x)dx$ for every integer $i \geq 1$. Using the inequality, one can write

$$0 \leq H(t) = \beta(0) + \sum_{i=1}^{t-1} \beta(i) \leq \beta(0) + \int_0^{t-1} \beta(x)dx. \tag{6}$$

If $\lim_{t \to \infty} H(t) < \infty$, clearly $\lim_{t \to \infty} \frac{H(t)}{t} = 0$ holds. If $\lim_{t \to \infty} H(t) = \infty$, we leverage L'Hôpital's rule with $\beta(t) \to 0$ as $t \to \infty$ to derive

$$\lim_{t \to \infty} \frac{H(t)}{t} \leq \lim_{t \to \infty} \frac{\beta(0) + \int_0^{t-1} \beta(x)dx}{t} = \lim_{t \to \infty} \frac{\beta(t-1)}{1} = 0,$$

where the first inequality follows from (6). $\square$

**Lemma C.2.** *For $0 \le j \le k$, we have*

$$\frac{H(\tau_k)}{H(\tau_j)} \le \frac{\tau_k}{\tau_j}.$$

*Proof.* For $0 \le j \le k$,

$$\frac{H(\tau_k)}{H(\tau_j)} \le \frac{H(\tau_j) + \sum_{i=\tau_j}^{\tau_k - 1} \beta(i)}{H(\tau_j)} \le 1 + \frac{(\tau_k - \tau_j) \cdot \beta(\tau_j)}{\tau_j \cdot \beta(\tau_j)} = \frac{\tau_k}{\tau_j},$$

where the last inequality is due to the non-increasing property of $\beta(\cdot)$. The equality holds when $\beta(0) = \cdots = \beta(\tau_k - 1)$. □

**Lemma C.3** (Sum of state norms in a single batch). *In Algorithm 1, for each batch $b = 0, 1, \ldots, B-1$, the following inequality holds:*

$$\sum_{t=t_b}^{t_{b+1}-1} \|x_t\| \le H(\tau_b)\|x_{t_b}\| + \gamma w_{max}(\tau_b - 1)$$

*Proof.* For $t = t_b$, we have $\|x_t\| \le \beta(0)\|x_{t_b}\|$ since $\beta(0) = 1$. For $t_b < t \le t_{b+1} - 1$, we have

$$\|x_t\| \le \beta(t - t_b)\|x_{t_b}\| + \gamma w_{\max}. \tag{7}$$

Summing up all inequalities gives

$$\sum_{t=t_b}^{t_{b+1}-1} \|x_t\| \le H(t_{b+1} - t_b)\|x_{t_b}\| + \gamma w_{\max}(t_{b+1} - t_b - 1).$$

Since Line 5 of Algorithm 1 is not satisfied, $\tau_b = t_{b+1} - t_b$. This completes the proof. □

**Lemma C.4** (Weighted sum of state norms between the two consecutive Break statements). *In Algorithm 1, suppose that $\frac{\tau_1}{\tau_0}\beta(\tau_0) < 1$. For every next batch index after the Break statement $u = 0, \ldots, U$, the following inequality holds:*

$$\sum_{b=b_u}^{b_{u+1}-1} H(\tau_b)\|x_{t_b}\| \le \frac{1}{1 - \frac{\tau_{b_u+1}}{\tau_{b_u}}\beta(\tau_{b_u})} H(\tau_{b_u})\|x_{t_{b_u}}\| + \frac{\gamma w_{max}}{1 - \beta(\tau_{b_u+1})} \sum_{b=b_u+1}^{b_{u+1}-1} H(\tau_b).$$

*Proof.* Since we designed $(\tau_b)_{b \ge 0}$ to have a non-decreasing $\tau_b$ and non-increasing $\frac{\tau_{b+1}}{\tau_b}$, notice that we have $\beta(\tau_b) \le \frac{\tau_{b+1}}{\tau_b}\beta(\tau_b) \le \frac{\tau_b}{\tau_{b-1}}\beta(\tau_{b-1}) \le \frac{\tau_1}{\tau_0}\beta(\tau_0) < 1$ for every $b \ge 1$ since $\beta(\cdot)$ is non-increasing.

If $b_{u+1} = b_u + 1$, the inequality clearly holds since $\frac{1}{1 - \frac{\tau_{b_u+1}}{\tau_{b_u}}\beta(\tau_{b_u})} > 0$. Otherwise, consider the following inequality for $b_u < b \le b_{u+1} - 1$:

$$H(\tau_b)\|x_{t_b}\| \le H(\tau_b)\beta(\tau_{b-1})\|x_{t_{b-1}}\| + H(\tau_b)\gamma w_{\max}$$
$$= \frac{H(\tau_b)}{H(\tau_{b-1})}\beta(\tau_{b-1})H(\tau_{b-1})\|x_{t_{b-1}}\| + H(\tau_b)\gamma w_{\max},$$

where the inequality holds since Line 5 of Algorithm 1 is not satisfied. Recursively applying this inequality, one arrives at

$$H(\tau_b)\|x_{t_b}\| \le \Pi_{a=b_u}^{b-1}\left[\frac{H(\tau_{a+1})}{H(\tau_a)}\beta(\tau_a)\right] \cdot H(\tau_{b_u})\|x_{t_{b_u}}\| + H(\tau_b)\gamma w_{\max}\left(1 + \sum_{b'=b_u+1}^{b-1} \Pi_{a=b'}^{b-1}\beta(\tau_a)\right)$$

$$\le \Pi_{a=b_u}^{b-1}\left[\frac{H(\tau_{a+1})}{H(\tau_a)}\beta(\tau_a)\right] \cdot H(\tau_{b_u})\|x_{t_{b_u}}\| + H(\tau_b)\gamma w_{\max}\left(1 + \sum_{b'=b_u+1}^{b-1} [\beta(\tau_{b_u+1})]^{b-b'}\right)$$

$$\le \Pi_{a=b_u}^{b-1}\left[\frac{H(\tau_{a+1})}{H(\tau_a)}\beta(\tau_a)\right] \cdot H(\tau_{b_u})\|x_{t_{b_u}}\| + H(\tau_b)\frac{\gamma w_{\max}}{1 - \beta(\tau_{b_u+1})} \tag{8}$$

$$\leq \Pi_{a=b_u}^{b-1} \left[ \frac{\tau_{a+1}}{\tau_a} \beta(\tau_a) \right] \cdot H(\tau_{b_u}) \|x_{t_{b_u}}\| + H(\tau_b) \frac{\gamma w_{\max}}{1 - \beta(\tau_{b_u+1})}$$

$$\leq \left[ \frac{\tau_{b_u+1}}{\tau_{b_u}} \beta(\tau_{b_u}) \right]^{b-b_u} \cdot H(\tau_{b_u}) \|x_{t_{b_u}}\| + H(\tau_b) \frac{\gamma w_{\max}}{1 - \beta(\tau_{b_u+1})},$$

where the second inequality comes from the non-increasing property of $\beta(\cdot)$, the third inequality is by $\beta(\tau_{b_u+1}) < 1$, the fourth inequality is due to Lemma C.2, and the last inequality comes from the non-increasing property of $\frac{\tau_{b+1}}{\tau_b} \beta(\tau_b)$. Since $\frac{\tau_{b_u+1}}{\tau_{b_u}} \beta(\tau_{b_u}) < 1$, summing up the above inequalities for $b_u < b \leq b_{u+1} - 1$ completes the proof. $\square$

**Lemma C.5** (Next state norm after the Break statement). *Define* $M_1 := L_f(1 + L_\pi)\gamma w_{max} + L_f(\pi_{0,max} + w_{max}) + f_0$. *Then, for every* $u = 1, \ldots, U$, *we have*

$$\|x_{t_{b_u}}\| \leq L_f(1 + L_\pi)\beta(0) \|x_{t_{b_u}-1}\| + M_1.$$

*Proof.* Suppose we picked a controller $\pi_t$ at time step $t$. Then, by Assumption 2.5, we have

$$\|u_t\| = \|\pi_t(x_t) - \pi_t(0) + \pi_t(0)\| \leq \|\pi_t(x_t) - \pi_t(0)\| + \|\pi_t(0)\| \leq L_\pi \|x_t\| + \pi_{0,\max}. \quad (9)$$

Combining the above inequality with Assumption 2.1, one can write

$$\begin{aligned}
\|x_{t+1}\| &= \|f(x_t, u_t, w_t) - f(0,0,0) + f(0,0,0)\| \\
&\leq \|f(x_t, u_t, w_t) - f(0,0,0)\| + \|f(0,0,0)\| \leq L_f(\|x_t\| + \|u_t\| + \|w_t\|) + f_0 \\
&\leq L_f(\|x_t\| + L_\pi \|x_t\| + \pi_{0,\max} + w_{\max}) + f_0 \\
&= L_f(1 + L_\pi)\|x_t\| + L_f(\pi_{0,\max} + w_{\max}) + f_0.
\end{aligned}$$

Thus, for every $u = 1, \ldots, U$, we obtain that

$$\begin{aligned}
\|x_{t_{b_u}}\| &\leq L_f(1 + L_\pi)\|x_{t_{b_u}-1}\| + L_f(\pi_{0,\max} + w_{\max}) + f_0 \\
&\leq L_f(1 + L_\pi)(\beta(t_{b_u} - t_{b_u-1} - 1)\|x_{t_{b_u}-1}\| + \gamma w_{\max}) + L_f(\pi_{0,\max} + w_{\max}) + f_0 \\
&= L_f(1 + L_\pi)\beta(t_{b_u} - t_{b_u-1} - 1)\|x_{t_{b_u}-1}\| + M_1 \\
&\leq L_f(1 + L_\pi)\beta(0)\|x_{t_{b_u}-1}\| + M_1,
\end{aligned}$$

where the second inequality holds since Line 5 of Algorithm 1 is not satisfied during $t_{b_u-1} \leq t \leq t_{b_u} - 1$ and the equality holds for the last inequality when $t_{b_u} = t_{b_u-1} + 1$. This completes the proof. $\square$

**Lemma C.6** (Weighted sum of state norms along the Break statements). *In Algorithm 1, suppose that* $\frac{\tau_1}{\tau_0} \beta(\tau_0) < 1$. *Define* $M_2 := L_f(1 + L_\pi)\beta(0)\frac{\gamma w_{max}}{1-\beta(\tau_1)} + M_1$. *Then, there exists a constant* $C \geq 1$ *such that*

$$\sum_{u=0}^{U} H(\tau_{b_u}) \|x_{t_{b_u}}\| \leq \frac{[L_f(1+L_\pi)\beta(0)C]^{U+1} - 1}{L_f(1+L_\pi)\beta(0)C - 1} H(\tau_0)\|x_0\| + \frac{([L_f(1+L_\pi)\beta(0)C]^U - 1)M_2}{[L_f(1+L_\pi)\beta(0)C - 1]^2} H(\tau_{b_U})$$

*Proof.* Since we designed $\frac{\tau_{b+1}}{\tau_b}$ to converge, there exists $R > 0$ such that $\frac{\tau_{b+1}}{\tau_b} \leq R$ for all $b \geq 0$. Moreover, since $\lim_{b \to \infty} \frac{\tau_{b+1}}{\tau_b} = 1$ and $\beta(\tau_0) < 1$, there exists $b^* > 0$ such that

$$b \geq b^* \implies \frac{\tau_{b+1}}{\tau_b} < \frac{1}{\beta(\tau_0)}. \quad (10)$$

Accordingly, for any two batches $b' > b \geq 0$, we have

$$\frac{\tau_{b'}}{\tau_b} [\beta(\tau_b)]^{b'-b-1} \leq [\beta(\tau_0)]^{b'-b-1} \Pi_{a=b}^{b'-1} \frac{\tau_{a+1}}{\tau_a} \leq \frac{R^{b^*}}{\beta(\tau_0)}, \quad (11)$$

considering that $b = 0$ and $b' = b^*$ yields the largest possible upper bound due to (10). Now, define $C := \frac{R^{b^*}}{\beta(\tau_0)}$. Notice that we have $C \geq 1$ since the left-hand side of (11) is greater than equal to 1 when $b' = b + 1$. Then, for every $u = 1, \ldots, U$, one can write

$$H(\tau_{b_u})\|x_{t_{b_u}}\| \leq L_f(1 + L_\pi)\beta(0)\frac{H(\tau_{b_u})}{H(\tau_{b_u-1})} H(\tau_{b_u-1})\|x_{t_{b_u}-1}\| + H(\tau_{b_u})M_1$$

$$\leq L_f(1+L_\pi)\beta(0)\frac{H(\tau_{b_u})}{H(\tau_{b_u-1})}\Pi_{a=b_{u-1}}^{b_u-2}\left[\frac{H(\tau_{a+1})}{H(\tau_a)}\beta(\tau_a)\right]\cdot H(\tau_{b_{u-1}})\|x_{t_{b_{u-1}}}\|$$

$$+ L_f(1+L_\pi)\beta(0)H(\tau_{b_u})\frac{\gamma w_{\max}}{1-\beta(\tau_{b_{u-1}+1})} + H(\tau_{b_u})M_1$$

$$\leq L_f(1+L_\pi)\beta(0)\frac{H(\tau_{b_u})}{H(\tau_{b_{u-1}})}[\beta(\tau_{b_{u-1}})]^{b_u-b_{u-1}-1}\cdot H(\tau_{b_{u-1}})\|x_{t_{b_{u-1}}}\| + H(\tau_{b_u})M_2$$

$$\leq L_f(1+L_\pi)\beta(0)\frac{\tau_{b_u}}{\tau_{b_{u-1}}}[\beta(\tau_{b_{u-1}})]^{b_u-b_{u-1}-1}\cdot H(\tau_{b_{u-1}})\|x_{t_{b_{u-1}}}\| + H(\tau_{b_u})M_2$$

$$\leq L_f(1+L_\pi)\beta(0)C\cdot H(\tau_{b_{u-1}})\|x_{t_{b_{u-1}}}\| + H(\tau_{b_u})M_2$$

where the first inequality is due to Lemma C.5, the second inequality is by (8) in Lemma C.4, the fourth inequality is due to Lemma C.2, and the last inequality is by (11). Recursively applying this inequality, one arrives at

$$H(\tau_{b_u})\|x_{t_{b_u}}\| \leq [L_f(1+L_\pi)\beta(0)C]^u H(\tau_0)\|x_0\| + M_2\cdot\sum_{i=1}^{u}[L_f(1+L_\pi)\beta(0)C]^{u-i}H(\tau_{b_i})$$

$$\leq [L_f(1+L_\pi)\beta(0)C]^u H(\tau_0)\|x_0\| + M_2 H(\tau_{b_U})\cdot\frac{[L_f(1+L_\pi)\beta(0)C]^u - 1}{L_f(1+L_\pi)\beta(0)C - 1}$$

$$< [L_f(1+L_\pi)\beta(0)C]^u\cdot\left[H(\tau_0)\|x_0\| + \frac{M_2 H(\tau_{b_U})}{L_f(1+L_\pi)\beta(0)C - 1}\right],$$

where the second inequality comes from the non-decreasing property of $H(\cdot)$ and the equality holds when $H(\tau_{b_1}) = \cdots = H(\tau_{b_U})$. Notice that for $b' > b \geq 0$, the case $H(\tau_{b'}) = H(\tau_b)$ arises when $\tau_{b'} = \tau_b$ or $\beta(\tau_b + 1) = \cdots = \beta(\tau_{b'}) = 0$. Since $L_f(1+L_\pi)\beta(0)C > 1$, summing up the above inequality for $u = 1, \ldots, U$ completes the proof. $\qquad\square$

**Lemma C.7** (Sum of state norms). *In Algorithm 1, suppose that $\frac{\tau_1}{\tau_0}\beta(\tau_0) < 1$. Then, we have*

$$\sum_{t=0}^{T}\|x_t\| \leq O([L_f(1+L_\pi)\beta(0)C]^U(\|x_0\| + H(\tau_{b_U}))) + \gamma w_{max}\cdot(O(\sum_{b=0}^{B-1}H(\tau_b)) + T)$$

*Proof.* Applying Lemma C.3, C.4, and C.6 in turn, we have

$$\sum_{t=0}^{T}\|x_t\| = \sum_{u=0}^{U}\sum_{b=b_u}^{b_{u+1}-1}\sum_{t=t_b}^{t_{b+1}-1}\|x_t\|$$

$$\leq \sum_{u=0}^{U}\sum_{b=b_u}^{b_{u+1}-1}\left[H(\tau_b)\|x_{t_b}\| + \gamma w_{\max}(\tau_b - 1)\right]$$

$$\leq \sum_{u=0}^{U}\left[\frac{1}{1-\frac{\tau_{b_u+1}}{\tau_{b_u}}\beta(\tau_{b_u})}H(\tau_{b_u})\|x_{t_{b_u}}\| + \frac{\gamma w_{\max}}{1-\beta(\tau_{b_u+1})}\sum_{b=b_u+1}^{b_{u+1}-1}H(\tau_b) + \gamma w_{\max}(t_{b_{u+1}} - t_{b_u} - 1)\right]$$

$$\leq \frac{1}{1-\frac{\tau_1}{\tau_0}\beta(\tau_0)}\sum_{u=0}^{U}H(\tau_{b_u})\|x_{t_{b_u}}\| + \frac{\gamma w_{\max}}{1-\beta(\tau_1)}(\sum_{b=0}^{B-1}H(\tau_b) - \sum_{u=0}^{U}H(\tau_{b_u})) + \gamma w_{\max}(T - U)$$

$$\leq \frac{1}{1-\frac{\tau_1}{\tau_0}\beta(\tau_0)}\sum_{u=0}^{U}H(\tau_{b_u})\|x_{t_{b_u}}\| + \frac{\gamma w_{\max}}{1-\beta(\tau_1)}\sum_{b=0}^{B-1}H(\tau_b) + \gamma w_{\max}T$$

$$\leq \frac{H(\tau_0)\|x_0\|}{1-\frac{\tau_1}{\tau_0}\beta(\tau_0)}\frac{[L_f(1+L_\pi)\beta(0)C]^{U+1} - 1}{L_f(1+L_\pi)\beta(0)C - 1} + \frac{H(\tau_{b_U})}{1-\frac{\tau_1}{\tau_0}\beta(\tau_0)}\frac{([L_f(1+L_\pi)\beta(0)C]^U - 1)M_2}{[L_f(1+L_\pi)\beta(0)C - 1]^2}$$

$$+ \frac{\gamma w_{\max}}{1-\beta(\tau_1)}\sum_{b=0}^{B-1}H(\tau_b) + \gamma w_{\max}T$$

$$= O([L_f(1+L_\pi)\beta(0)C]^U(\|x_0\| + H(\tau_{b_U}))) + \gamma w_{\max}\cdot(O(\sum_{b=0}^{B-1}H(\tau_b)) + T)$$

where the equality holds for the fourth inequality when Line 5 of Algorithm 1 is not satisfied for the entire horizon. $\square$

**Theorem C.8** (Restatement of Theorem 4.1, Asymptotic stability)**.** *In Algorithm 1, suppose that* $\frac{\tau_1}{\tau_0}\beta(\tau_0) < 1$. *Then, it holds that*

$$\lim_{T\to\infty} \frac{1}{T} \sum_{t=0}^{T} \|x_t\| \le \gamma w_{max}.$$

*Proof.* We mainly use Lemma C.1 to prove the asymptotic stability. First, we have

$$H(\tau_{b_U}) \le H(\tau_{B-1}) = o(\tau_{B-1}) = o(T), \tag{12}$$

where the first equality is due to Lemma C.1 and $\tau_{B-1} = T$ when there is only one batch over the entire horizon. Now, consider the following relationship between the number of batch $B$ and the time horizon $T$:

$$\sum_{b=0}^{B-1} \tau_b \ge T \ge \sum_{b=0}^{B-U-1} \tau_b + U, \tag{13}$$

where the second inequality is due to the non-decreasing property of $\tau_b$. Now, if $\sum_{b=0}^{B-1} H(\tau_b) < \infty$, clearly $\sum_{b=0}^{B-1} H(\tau_b) = o(T)$. Otherwise, define $H(\tau_B) = H(\tau_{B-1})$. Then, we have

$$\lim_{T\to\infty} \frac{\sum_{b=0}^{B-1} H(\tau_b)}{T} \le \lim_{B\to\infty} \frac{\sum_{b=0}^{B-1} H(\tau_b)}{\sum_{b=0}^{B-U-1} \tau_b + U} \le \lim_{B\to\infty} \frac{\int_0^B H(\tau_b)db}{\tau_0 + \int_0^{B-U-1} \tau_b db + U}$$

$$= \lim_{B\to\infty} \frac{H(\tau_{B-1})}{\tau_{B-U-1}} = \lim_{B\to\infty} \frac{H(\tau_{B-1})}{\tau_{B-1}} \Pi_{b=B-U-1}^{B-2} \frac{\tau_{b+1}}{\tau_b}$$

$$= 0 \cdot 1^U = 0 \tag{14}$$

where the second inequality leverages the non-decreasing property of both $\tau_b$ and $H(\tau_b)$, the remaining equalities leverage L'Hôpital's rule, Lemma C.1, and $\lim_{b\to\infty} \frac{\tau_{b+1}}{\tau_b} = 1$. Thus, with Lemma C.7, we have

$$\sum_{t=0}^{T} \|x_t\| \le O([L_f(1+L_\pi)\beta(0)C]^U(\|x_0\| + o(T))) + \gamma w_{\max} \cdot (T + o(T)).$$

This completes the proof. $\square$

**Lemma C.9.** *In Algorithm 1, we have*

$$\lim_{T\to\infty} \frac{B}{T} = 0$$

*Proof.* Recall the relationship stated in (13) between $T$ and $B$. Using the second inequality, we have

$$0 \le \lim_{T\to\infty} \frac{B}{T} \le \lim_{T\to\infty} \frac{B}{\sum_{b=0}^{B-U-1} \tau_b + U} \le \lim_{T\to\infty} \frac{B}{\tau_0 + \int_0^{B-U-1} \tau_b db + U}$$

$$= \lim_{T\to\infty} \frac{1}{\tau_{B-U-1}} = 0,$$

where the third inequality uses the non-decreasing property of $\tau_b$, after which we use L'Hôpital's rule. This completes the proof. $\square$

**Theorem C.10** (Restatement of Theorem 4.2, Finite-gain stability)**.** *In Algorithm 1, suppose that* $\frac{\tau_1}{\tau_0}\beta(\tau_0) < 1$. *Assume that* $\lim_{t\to\infty} H(t) < \infty$. *Then, Algorithm 1 achieves finite-gain* $\mathcal{L}_1$ *stability; i.e., there exist constants* $A_1, A_2 > 0$ *such that for all* $T \in \mathbb{Z}_+$,

$$\sum_{t=0}^{T} \|x_t\| \le A_1 \cdot w_{max} T + A_2.$$

*Proof.* Since $\lim_{t \to \infty} H(t) < \infty$, there exists a constant $q_1$ that upper-bounds $H(t)$; *i.e.*, $H(t) \le q_1$ for all $t \ge 0$. Likewise, by Lemma C.9, there exists a constant $q_2$ that upper-bounds $\frac{B}{T}$. Thus, with Lemma C.7, one can write

$$\sum_{t=0}^{T} \|x_t\| \le O([L_f(1 + L_\pi)\beta(0)C]^U(\|x_0\| + q_1)) + \gamma w_{\max} \cdot (O(Bq_1) + T)$$

$$= O([L_f(1 + L_\pi)\beta(0)C]^U(\|x_0\| + q_1)) + \gamma(1 + \frac{B}{T}O(q_1)) \cdot w_{\max}T$$

$$\le O([L_f(1 + L_\pi)\beta(0)C]^U(\|x_0\| + q_1)) + \gamma(1 + O(q_1 q_2)) \cdot w_{\max}T.$$

This completes the proof. $\qquad\square$

## D REGRET PROOF FOR ALGORITHM 1

**Lemma D.1.** *In Algorithm 1, we have*

$$\mathbb{E}_{K_{B-1:0}}[w_b(K_b)] = \mathbb{E}_{K_{B-1:0}}[\mathbb{E}_{k \sim p_b}[w_b'(k)]],$$

*Proof.* Given $K_{b-1}, \ldots, K_0$, we have

$$\mathbb{E}_{k \sim p_b}[w_b'(k)] = \sum_{k \in \mathcal{P}_b} p_b(k) \frac{w_b(K_b)}{p_b(k)} \mathcal{I}_{(K_b = k)} = w_b(K_b), \tag{15}$$

which implies that $w_b'(k)$ sampled from $p_b$ is an unbiased estimator of $w_b(K_b)$.

Thus, for all $b = 0, 1, \ldots, B - 1$, one can write

$$\mathbb{E}_{K_{B-1:0}}[w_b(K_b)] = \mathbb{E}_{K_{b:0}}[w_b(K_b)] = \mathbb{E}_{K_{b-1:0}}\mathbb{E}_{K_b}[w_b(K_b) \mid K_{b-1:0}]$$

$$= \mathbb{E}_{K_{b-1:0}}\mathbb{E}_{K_b}[\mathbb{E}_{k \sim p_b}[w_b'(k)] \mid K_{b-1:0}]$$

$$= \mathbb{E}_{K_{b:0}}[\mathbb{E}_{k \sim p_b}[w_b'(k)]] = \mathbb{E}_{K_{B-1:0}}[\mathbb{E}_{k \sim p_b}[w_b'(k)]],$$

where the first equality is because $K_{B-1}, \ldots, K_{b+1}$ does not affect the value of $w_b(K_b)$ and the remaining equalities are by law of total expectation and (15). $\qquad\square$

Now, we let $w_b^K(i)$ denote the cost incurred at batch $b$ if one selects the controllers for batch $0, \ldots, b - 1$ according to Algorithm 1, and the controller for batch $b$ to be $i$.

**Lemma D.2.** *In Algorithm 1, for any $i \in \mathcal{P}_b$, we have*

$$\mathbb{E}_{K_{B-1:0}}[w_b'(i)] = \mathbb{E}_{K_{B-1:0}}[w_b^K(i)]$$

*and for some controller $i^b \in \mathcal{P}_b$, we have*

$$\mathbb{E}_{K_{B-1:0}}\left[\frac{\eta_0}{2} \frac{(w_b(K_b))^2}{p_b(K_b)}\right] \le \frac{\eta_0 N}{2} \mathbb{E}_{K_{b-1:0}}(w_b^K(i^b))^2.$$

*Proof.* For all $b = 0, 1, \ldots, B - 1$ and for all $i \in \mathcal{P}_b$, we have

$$\mathbb{E}_{K_{B-1:0}}[w_b'(i)] = \mathbb{E}_{K_{b:0}}[w_b'(i)] = \mathbb{E}_{K_{b-1:0}}[\mathbb{E}_{K_b}[w_b'(i) \mid K_{b-1:0}]]$$

$$= \mathbb{E}_{K_{b-1:0}}\Big[\sum_{K_b \in \mathcal{P}_b} p_b(K_b) \frac{w_b(K_b)}{p_b(i)} \mathcal{I}_{(K_b = i)}\Big]$$

$$= \mathbb{E}_{K_{b-1:0}}[w_b^K(i)] = \mathbb{E}_{K_{B-1:0}}[w_b^K(i)]$$

where the first equality is because $K_{B-1}, \ldots, K_{b+1}$ does not affect the value of $w_b'(i)$ and the last equality is because $K_{B-1}, \ldots, K_b$ does not affect the value of $w_b^K(i)$. Next, we can also obtain that

$$\mathbb{E}_{K_{B-1:0}}\left[\frac{\eta_0}{2} \frac{(w_b(K_b))^2}{p_b(K_b)}\right] = \mathbb{E}_{K_{b:0}}\left[\frac{\eta_0}{2} \frac{(w_b(K_b))^2}{p_b(K_b)}\right] = \mathbb{E}_{K_{b-1:0}}\mathbb{E}_{K_b}\left[\frac{\eta_0}{2} \frac{(w_b(K_b))^2}{p_b(K_b)} \mid K_{b-1:0}\right]$$

$$= \mathbb{E}_{K_{b-1:0}} \sum_{K_b \in \mathcal{P}_b} \left[ \frac{\eta_0}{2} p_b(K_b) \frac{(w_b(K_b))^2}{p_b(K_b)} \right] = \mathbb{E}_{K_{b-1:0}} \sum_{K_b \in \mathcal{P}_b} \left[ \frac{\eta_0}{2} (w_b(K_b))^2 \right]$$

$$\leq \frac{\eta_0 N}{2} \mathbb{E}_{K_{b-1:0}} (w_b^K(i^b))^2,$$

for the controller $i^b = \arg\max_{i \in \mathcal{P}_b} (w_b^K(i))^2$. This completes the proof. $\qquad\square$

In Algorithm 1, define $\mathcal{L} := \{0 \leq b \leq B-1, \ b \in \mathbb{Z}_+ : s_{b+1} \neq s_b\}$ and let $b^1, \ldots, b^{|\mathcal{L}|}$ denote the batch where Line 22 of Algorithm 1 is satisfied; *i.e.*, $s_{b^l+1} \neq s_{b^l}$ for $l = 1, \ldots, |\mathcal{L}|$. For convenience, we let $b^0 = 0$, $b^{|\mathcal{L}|+1} = B-1$, and $s_B = s_{B-1}$. Also, define $\mathcal{V} := \{0 \leq b \leq B-1, \ b \in \mathbb{Z}_+ : s_b \neq 0\}$.

**Lemma D.3** (Restatement of Lemma 4.7). *In Algorithm 1, suppose that $\beta(\tau_0) < 1$ and let $U$ denote the number of times that the Break statement is activated. Then, it holds that $|\mathcal{L}| = O(U)$ and $|\mathcal{V}| = O(U)$.*

*Proof.* For every batch $b = 0, \ldots, B-1$, we have

$$\|x_{t_b}\| < (\alpha_b)^{s_b+1} \|x_0\| + \delta \tag{16}$$

by Lines 11-20. If the Break statement is not activated, since we designed $\delta \geq \frac{\gamma w_{\max}}{1 - \beta(\tau_0)}$, it yields that

$$\|x_{t_{b+1}}\| \leq \beta(\tau_b) \|x_{t_b}\| + \gamma w_{\max} \leq \beta(\tau_0)(\alpha_b)^{s_b+1} \|x_0\| + \beta(\tau_0)\delta + \gamma w_{\max}$$
$$\leq \beta(\tau_0)(\alpha_b)^{s_b+1} \|x_0\| + \delta < (\alpha_b)^{s_b+1} \|x_0\| + \delta,$$

where the second and the last inequalities are due to $\beta(\tau_b) \leq \beta(\tau_0) < 1$ and the third inequality is by the formulation of $\delta$. Then, $s_{b+1} > s_b$ cannot occur when the Break statement is not activated. Also, Line 14 avoids $s_{b+1} > s_b + 1$. As a result, starting from $s_0 = 0$, the event $s_{b+1} = s_b + 1$ can occur at most $U$ times. Accordingly, the event $s_{b+1} < s_b$ also can occur at most $U$ times, leading to $|\mathcal{L}| \leq 2U$.

Now, we observe the number of batches $\tilde{b}$ needed to stabilize the state norm; *i.e.*, $\min\{\tilde{b} > 0 : s_{b+\tilde{b}} < s_b\}$ when the Break statement is not activated. Starting from batch $b$ and the corresponding $s_b$, provided that the Break statement is not activated, one can write

$$\|x_{t_{b+\tilde{b}}}\| \leq \beta(\tau_{b+\tilde{b}-1}) \|x_{t_{b+\tilde{b}-1}}\| + \gamma w_{\max} \leq \beta(\tau_0) \|x_{t_{b+\tilde{b}-1}}\| + \gamma w_{\max}$$

$$\leq (\beta(\tau_0))^{\tilde{b}} \|x_{t_b}\| + \gamma w_{\max} \sum_{a=0}^{\tilde{b}-1} (\beta(\tau_0))^a \leq (\beta(\tau_0))^{\tilde{b}} \|x_{t_b}\| + \frac{\gamma w_{\max}}{1 - \beta(\tau_0)}$$

$$\leq (\beta(\tau_0))^{\tilde{b}} \|x_{t_b}\| + \delta < (\beta(\tau_0))^{\tilde{b}} [(\alpha_b)^{s_b+1} \|x_0\| + \delta] + \delta, \tag{17}$$

where the first and third inequalities are due to not satisfying Line 5 iteratively when the Break statement is not activated, the second and fourth inequalities are by $\beta(\tau_b) \leq \beta(\tau_0) < 1$, and the last two inequalities are by the design of $\delta$ and (16). It is desirable to find the minimum value of $\tilde{b}$ that makes the right-hand side of (17) smaller than $(\alpha_b)^{s_b} \|x_0\| + \delta$:

$$(\beta(\tau_0))^{\tilde{b}} [(\alpha_b)^{s_b+1} \|x_0\| + \delta] + \delta \leq (\alpha_b)^{s_b} \|x_0\| + \delta \iff \frac{1}{(\beta(\tau_0))^{\tilde{b}}} \geq \alpha_b + \frac{\delta}{(\alpha_b)^{s_b} \|x_0\|}, \tag{18}$$

where the right-hand side of (18) can be upper-bounded by $\alpha_b + \frac{\delta}{\|x_0\|}$ since $\alpha_b > 1$. Thus, if $s_b \neq 0$,

$$\min\{\tilde{b} > 0 : s_{b+\tilde{b}} < s_b\} \leq \left\lceil \frac{\log(\alpha_b + \frac{\delta}{\|x_0\|})}{-\log \beta(\tau_0)} \right\rceil, \tag{19}$$

when the Break statement is not activated. In other words, starting from a batch $b$ where $s_b > 0$, within the number of batches on the right-hand side of (19), either the Break statement is activated or the value of $s_b$ decreases.

More specifically, consider two sets of batches: $\mathcal{B}_1 = \{0 \leq b \leq B-1, b \in \mathbb{Z}_+ :$ the Break statement activated$\}$ and $\mathcal{B}_2 = \{0 \leq b \leq B-1, b \in \mathbb{Z}_+ : s_{b+1} < s_b\}$. Let $\mathcal{B} = \mathcal{B}_1 \cup \mathcal{B}_2$

be the set ordered by batch numbers. Then, the batch interval between two consecutive batches in $\mathcal{B}$ is upper-bounded by (19). Thus, considering that $|\mathcal{L}| \leq 2U$, we have

$$|\mathcal{V}| \leq (2U - 1) \left\lceil \frac{\log(\alpha_b + \frac{\delta}{\|x_0\|})}{-\log \beta(\tau_0)} \right\rceil,$$

which completes the proof. $\qquad\square$

**Lemma D.4** (cumulative mix loss). *In Algorithm 1, for any controller $i^l \in \mathcal{U}^c$ for $l = 0, \ldots, |\mathcal{L}|$, the cumulative mix loss is upper-bounded as follows:*

$$\mathbb{E}_{K_{B-1:0}} \sum_{b=0}^{B-1} -\frac{1}{\eta_0} \log(\mathbb{E}_{k \sim p_b} \exp(-\eta_b w_b'(k))) \leq \frac{\tilde{O}(U+1)}{\eta_0} + \mathbb{E}_{K_{B-1:0}} \sum_{l=0}^{|\mathcal{L}|} \sum_{b=b^l}^{b^{l+1}-1} \frac{w_b^K(i^l)}{(\alpha_b)^{2s_b}}$$

*Proof.* Given $l = 0, \ldots, |\mathcal{L}|$, we can analyze a single mix loss for $b = b^l + 1, \ldots, b^{l+1} - 1$ as follows:

$$-\frac{1}{\eta_0} \log(\mathbb{E}_{k \sim p_b} \exp(-\eta_b w_b'(k))) = -\frac{1}{\eta_0} \log(\sum_{k \in \mathcal{P}_b} p_b(k) \exp(-\eta_b w_b'(k)))$$

$$= -\frac{1}{\eta_0} \log(\frac{\sum_{k \in \mathcal{P}_b} \exp(-\eta_b W_b(k)) \exp(-\eta_b w_b'(k))}{\sum_{i \in \mathcal{P}_b} \exp(-\eta_b W_b(i))})$$

$$= -\frac{1}{\eta_0} \log(\frac{\sum_{k \in \mathcal{P}_b} \exp(-\eta_b W_{b+1}(k))}{\sum_{i \in \mathcal{P}_b} \exp(-\eta_b W_b(i))}), \qquad (20)$$

while a mix loss for $b = b^l$ is as follows:

$$-\frac{1}{\eta_0} \log(\mathbb{E}_{k \sim p_b} \exp(-\eta_{b^l} w_{b^l}'(k))) = -\frac{1}{\eta_0} \log(\sum_{k \in \mathcal{P}_{b^l}} p_{b^l}(k) \exp(-\eta_{b^l} w_{b^l}'(k)))$$

$$= -\frac{1}{\eta_0} \log(\frac{1}{|\mathcal{P}_{b^l}|} \sum_{k \in \mathcal{P}_{b^l}} \exp(-\eta_{b^l} w_{b^l}'(k)))$$

$$\leq \frac{\log N}{\eta_0} - \frac{1}{\eta_0} \log(\sum_{k \in \mathcal{P}_{b^l}} \exp(-\eta_{b^l} w_{b^l}'(k))) \qquad (21)$$

$$= \frac{\log N}{\eta_0} - \frac{1}{\eta_0} \log(\sum_{k \in \mathcal{P}_{b^l}} \exp(-\eta_{b^l} W_{b^l+1}(k))), \qquad (22)$$

where the last equality only holds when $b^{l+1} > b^l + 1$. Now, notice that the batches $b = b^l, \ldots, b^{l+1} - 1$ share the same learning rate; *i.e.*, $\eta_{b^l} = \cdots = \eta_{b^{l+1}-1}$ since the same $s_b$ yields the same $\alpha_b$, and thus the same $\eta_b$. Thus, in the case where $b^{l+1} > b^l + 1$, we have

$$\sum_{b=b^l}^{b^{l+1}-1} -\frac{1}{\eta_0} \log(\mathbb{E}_{k \sim p_b} \exp(-\eta_b w_b'(k))) \leq \frac{\log N}{\eta_0} - \frac{1}{\eta_0} \log(\Pi_{b=b^l+1}^{b^{l+1}-1} \frac{\sum_{k \in \mathcal{P}_{b-1}} \exp(-\eta_{b^l} W_b(k))}{\sum_{k \in \mathcal{P}_b} \exp(-\eta_{b^l} W_b(k))})$$

$$- \frac{1}{\eta_0} \log(\sum_{k \in \mathcal{P}_{b^{l+1}-1}} \exp(-\eta_{b^l} W_{b^{l+1}}(k)))$$

$$\leq \frac{\log N}{\eta_0} - \frac{1}{\eta_0} \log(\sum_{k \in \mathcal{P}_{b^{l+1}-1}} \exp(-\eta_{b^l} W_{b^{l+1}}(k))), \qquad (23)$$

where the first inequality is by (20) and (22) and the second inequality comes from $\mathcal{P}_b \subseteq \mathcal{P}_{b-1}$. Considering both cases (21) and (23), for any controller $i^0, \ldots, i^{|\mathcal{L}|} \in \mathcal{U}^c$, one can write

$$\sum_{b=0}^{B-1} -\frac{1}{\eta_0} \log(\mathbb{E}_{k \sim p_b} \exp(-\eta_b w_b'(k))) \leq \sum_{l=0}^{|\mathcal{L}|} \left[ \frac{\log N}{\eta_0} - \frac{1}{\eta_0} \log(\sum_{k \in \mathcal{P}_{b^{l+1}-1}} \exp(-\eta_{b^l} \sum_{b=b^l}^{b^{l+1}-1} w_b'(k))) \right]$$

$$\leq \frac{(|\mathcal{L}|+1)\log N}{\eta_0} - \sum_{l=0}^{|\mathcal{L}|} \frac{1}{\eta_0} \log(\exp(-\eta_{b^l} \sum_{b=b^l}^{b^{l+1}-1} w_b'(i^l)))$$

$$= \frac{\tilde{O}(U+1)}{\eta_0} + \sum_{l=0}^{|\mathcal{L}|} \frac{\sum_{b=b^l}^{b^{l+1}-1} w_b'(i^l)}{(\alpha_{b^l})^{2s_{b^l}}}, \tag{24}$$

where the first inequality considers $W_{b^l+1}(k) = \sum_{b=b^l}^{b^{l+1}-1} w_b'(k)$ in (23), the second inequality is because any controller $i^l$ is an element of $\mathcal{P}_{b^{l+1}-1}$, and the last equality comes from the definition of $\eta_{b^l} = \eta_0/(\alpha_{b^l})^{2s_{b^l}}$ and $|\mathcal{L}| = O(U)$ by Lemma D.3. Finally, by Lemma D.2, taking the expectation of (24) with respect to $K_{B-1:0}$ completes the proof. $\qquad\square$

Now, we consider the cumulative mixability gap.

**Lemma D.5** (cumulative mixability gap). *In Algorithm 1, there exists a set of controllers $i^b \in \mathcal{P}_b$ for $b = 0, \ldots, B-1$ such that the cumulative mixability gap is upper-bounded as follows:*

$$\mathbb{E}_{K_{B-1:0}} \sum_{b=0}^{B-1} \mathbb{E}_{k\sim p_b}[w_b'(k)] + \frac{1}{\eta_0} \log(\mathbb{E}_{k\sim p_b} \exp(-\eta_b w_b'(k))) \leq \frac{O(U)}{2\eta_0} + \frac{\eta_0 N}{2} \sum_{b=0}^{B-1} \mathbb{E}_{K_{b-1:0}} (w_b^K(i^b))^2$$

*Proof.* Given the set $\mathcal{V}$, we can analyze a single mixability gap for $b \notin \mathcal{V}$ and $b \in \mathcal{V}$, respectively. Since $s_b = 0$ for $b \notin \mathcal{V}$, given $K_{b-1}, \ldots, K_0$, we have

$$\mathbb{E}_{k\sim p_b}[w_b'(k)] + \frac{1}{\eta_0} \log(\mathbb{E}_{k\sim p_b} \exp(-\eta_b w_b'(k))) = \mathbb{E}_{k\sim p_b}[w_b'(k)] + \frac{1}{\eta_0} \log(\mathbb{E}_{k\sim p_b} \exp(-\eta_0 w_b'(k)))$$

$$\leq \mathbb{E}_{k\sim p_b}[w_b'(k)] + \frac{1}{\eta_0}(\mathbb{E}_{k\sim p_b} \exp(-\eta_0 w_b'(k)) - 1)$$

$$\leq \mathbb{E}_{k\sim p_b}[w_b'(k)] + \frac{1}{\eta_0}(\mathbb{E}_{k\sim p_b} \frac{\eta_0^2(w_b'(k))^2}{2} - \eta_0 w_b'(k))$$

$$= \frac{\eta_0}{2} \mathbb{E}_{k\sim p_b}[(w_b'(k))^2]$$

$$= \frac{\eta_0}{2} \sum_{k\in\mathcal{P}_b} p_b(k) \frac{(w_b(K_b))^2}{(p_b(k))^2} \mathcal{I}_{(K_b=k)} = \frac{\eta_0}{2} \frac{(w_b(K_b))^2}{p_b(K_b)}, \tag{25}$$

where the first inequality uses $\log(x) \leq x - 1$ for all $x \in \mathbb{R}$ and the second inequality uses $e^x \leq 1 + x + \frac{x^2}{2}$ for all $x \in \mathbb{R}$. Now, for $b \in \mathcal{V}$, given $K_{b-1}, \ldots, K_0$, we obtain that

$$\mathbb{E}_{k\sim p_b}[w_b'(k)] + \frac{1}{\eta_0} \log(\mathbb{E}_{k\sim p_b} \exp(-\eta_b w_b'(k))) \leq \mathbb{E}_{k\sim p_b}[w_b'(k)] + \frac{1}{\eta_0}(\mathbb{E}_{k\sim p_b} \exp(-\eta_b w_b'(k)) - 1)$$

$$\leq \mathbb{E}_{k\sim p_b}[w_b'(k)]$$

$$\leq \mathbb{E}_{k\sim p_b}[w_b'(k)] + \frac{1}{\eta_0}(\mathbb{E}_{k\sim p_b} \frac{\eta_0^2(w_b'(k))^2}{2} - \eta_0 w_b'(k) + \frac{1}{2})$$

$$= \frac{\eta_0}{2} \mathbb{E}_{k\sim p_b}[(w_b'(k))^2] + \frac{1}{2\eta_0} = \frac{\eta_0}{2} \frac{(w_b(K_b))^2}{p_b(K_b)} + \frac{1}{2\eta_0}, \tag{26}$$

where the second inequality uses $e^x \leq 1$ for all $x \leq 0$ and the third inequality uses $\frac{x^2}{2} + x + \frac{1}{2} \geq 0$ for all $x \in \mathbb{R}$. Since $|\mathcal{V}| = O(U)$ by Lemma D.3, we have inequality (26) holding at most $O(U)$ times and (25) holding in the remaining batches among $b = 0, \ldots, B-1$. Finally, by Lemma D.2, taking expectation of (25) and (26) with respect to $K_{B-1:0}$ completes the proof. $\qquad\square$

We let $x_t$ and $u_t$ denote the state and action sequence in the algorithm depending on the context. We let $x_t^K(i)$ and $u_t^K(i)$ for $t = t_b, \ldots, t_{b+1} - 1$ denote the state and action sequence generated by selecting the controllers before batch $b$ according to Algorithm 1, while selecting the controller $i$ at batch $b$. Accordingly, we have $w_b^K(i) = \sum_{t=t_b}^{t_{b+1}-1} c_t(x_t^K(i), u_t^K(i))$. We also let $x_t^*$ and $u_t^*$ denote

the optimal state and action sequence generated by the best stabilizing controller $i^*$ that satisfies both of Definitions 2.3 and 2.4; *i.e.*, $i^* = \arg\min_{i \in \mathcal{S}} \sum_{t=0}^{T} c_t(x_t, \pi_{i^*}(x_t))$ subject to the transition dynamics.

**Lemma D.6.** *In Algorithm 1, suppose that $\frac{\tau_1}{\tau_0}(\beta(\tau_0))^2 < \frac{1}{2\sqrt{2}}$. For any controller $i^b \in \mathcal{P}_b$ for $b = 0, \ldots, B-1$, we have*

$$\sum_{b=0}^{B-1} \mathbb{E}_{K_{b-1:0}}(w_b^K(i^b))^2 = \exp(O(U))O(\tau_{B-1}H(\tau_{B-1})) + O(\sum_{b=0}^{B-1}(\tau_b)^2).$$

*Proof.* By Assumption 2.2, for all $x \in \mathbb{R}^n$ and $u \in \mathbb{R}^m$, we have

$$
\begin{aligned}
|c_t(x,u)| &= |c_t(x,u) - c_t(0,0) + c_t(0,0)| \leq |c_t(x,u) - c_t(0,0)| + |c_t(0,0)| \\
&\leq (L_{c_1}(\|x\| + \|u\|) + L_{c2})(\|x\| + \|u\|) + c_{0,\max} \\
&= L_{c_1}(\|x\| + \|u\|)^2 + L_{c2}(\|x\| + \|u\|) + c_{0,\max} \\
&\leq 2L_{c_1}(\|x\|^2 + \|u\|^2) + L_{c2}(\|x\| + \|u\|) + c_{0,\max},
\end{aligned}
\tag{27}
$$

where the last inequality is due to Cauchy–Schwarz inequality. Thus, we can upper-bound $(w_b^K(i^b))^2$ for any controller $i^b \in \mathcal{P}_b$ for $b = 0, \ldots, B-1$ as follows:

$$
(w_b^K(i^b))^2 = \left[\sum_{t=t_b}^{t_{b+1}-1} c_t(x_t^K(i^b), u_t^K(i^b))\right]^2 \leq \sum_{t=t_b}^{t_{b+1}-1} c_t(x_t^K(i^b), u_t^K(i^b))^2(t_{b+1} - t_b)
$$

$$
\leq (t_{b+1} - t_b) \sum_{t=t_b}^{t_{b+1}-1} (2L_{c_1}(\|x_t^K(i^b)\|^2 + \|u_t^K(i^b)\|^2) + L_{c2}(\|x_t^K(i^b)\| + \|u_t^K(i^b)\|) + c_{0,\max})^2
$$

$$
\leq 5(t_{b+1} - t_b) \sum_{t=t_b}^{t_{b+1}-1} (4L_{c_1}^2(\|x_t^K(i^b)\|^4 + \|u_t^K(i^b)\|^4) + L_{c2}^2(\|x_t^K(i^b)\|^2 + \|u_t^K(i^b)\|^2) + c_{0,\max}^2)
$$

$$\tag{28}$$

where the first and the third inequalities are due to Cauchy–Schwarz inequality.

From (7), for $t_b < t \leq t_{b+1} - 1$, we have

$$\|x_t^K(i^b)\|^2 \leq 2[\beta(t - t_b)]^2\|x_{t_b}^K(i^b)\|^2 + 2\gamma^2 w_{\max}^2 \tag{29}$$

$$\|x_t^K(i^b)\|^4 \leq 8[\beta(t - t_b)]^4\|x_{t_b}^K(i^b)\|^4 + 8\gamma^4 w_{\max}^4, \tag{30}$$

where the inequalities are by Cauchy-Schwarz inequality. Accordingly, we obtain that

$$\sum_{t=t_b}^{t_{b+1}-1} \|x_t^K(i^b)\|^2 \leq 2H(t_{b+1} - t_b)\|x_{t_b}^K(i^b)\|^2 + 2\gamma^2 w_{\max}^2(t_{b+1} - t_b - 1) \tag{31}$$

$$\sum_{t=t_b}^{t_{b+1}-1} \|x_t^K(i^b)\|^4 \leq 8H(t_{b+1} - t_b)\|x_{t_b}^K(i^b)\|^4 + 8\gamma^4 w_{\max}^4(t_{b+1} - t_b - 1), \tag{32}$$

where we use $\beta(\cdot) \leq 1$ to derive $\sum_{t=0}^{t_{b+1}-t_b-1}[\beta(t)]^p \leq \sum_{t=0}^{t_{b+1}-t_b-1}[\beta(t)] = H(t_{b+1} - t_b)$ for $p \geq 1$.

From (9), for $t_b \leq t \leq t_{b+1} - 1$, we have

$$\|u_t^K(i^b)\|^2 \leq 2L_\pi^2\|x_t^K(i^b)\|^2 + 2\pi_{0,\max}^2 \tag{33}$$

$$\|u_t^K(i^b)\|^4 \leq 8L_\pi^4\|x_t^K(i^b)\|^4 + 8\pi_{0,\max}^4, \tag{34}$$

where the inequalities are by Cauchy-Schwarz inequality. Now, we substitute (31), (32), (33), (34), and $t_{b+1} - t_b \leq \tau_b$ into the right-hand side of (28) to upper-bound $(w_b^K(i^b))^2$ as follows:

$$
(w_b^K(i^b))^2 \leq 5\tau_b[32L_{c1}^2(1 + 8L_\pi^4)H(\tau_b)\|x_{t_b}^K(i^b)\|^4 + 2L_{c2}^2(1 + 2L_\pi^2)H(\tau_b)\|x_{t_b}^K(i^b)\|^2] +
$$

$$
5\tau_b^2[32L_{c1}^2((1 + 8L_\pi^4)\gamma^4 w_{\max}^4 + \pi_{0,\max}^4) + 2L_{c2}^2((1 + 2L_\pi^2)\gamma^2 w_{\max}^2 + \pi_{0,\max}^2) + c_{0,\max}^2]
$$

$$= M_3 \tau_b H(\tau_b) \|x_{t_b}^K(i^b)\|^4 + M_4 \tau_b H(\tau_b) \|x_{t_b}^K(i^b)\|^2 + M_5 \tau_b^2$$

$$= M_3 \tau_b H(\tau_b) \|x_{t_b}\|^4 + M_4 \tau_b H(\tau_b) \|x_{t_b}\|^2 + M_5 \tau_b^2, \tag{35}$$

where $M_3, M_4, M_5$ are constants determined by $L_{c1}, L_{c2}, L_\pi, \gamma, w_{\max}, \pi_{0,\max}$, and $c_{0,\max}$. The last equality comes from $x_{t_b}^K(i^b) = x_{t_b}$ for any $i^b \in \mathcal{P}_b$.

Meanwhile, one can upper-bound both $\sum_{b=0}^{B-1} \tau_b H(\tau_b) \|x_{t_b}\|^4$ and $\sum_{b=0}^{B-1} \tau_b H(\tau_b) \|x_{t_b}\|^2$ by successively applying Lemma C.3, C.4, and C.6 in the same fashion as presented in the proof of Lemma C.7. Since $\frac{\tau_1^2}{\tau_0^2} 8(\beta(\tau_0))^4 < 1$, by (29) and (30), there exists $C_1, C_2 \geq 1$ such that

$$\sum_{b=0}^{B-1} \tau_b H(\tau_b) \|x_{t_b}\|^4 = O([8L_f^4(1+L_\pi)^4 \beta(0)^4 C_1]^U (\|x_0\|^4 + \tau_{b_U} H(\tau_{b_U}))) + 8\gamma^4 w_{\max}^4 \cdot O(\sum_{b=0}^{B-1} \tau_b H(\tau_b))$$

$$\sum_{b=0}^{B-1} \tau_b H(\tau_b) \|x_{t_b}\|^2 = O([2L_f^2(1+L_\pi)^2 \beta(0)^2 C_2]^U (\|x_0\|^2 + \tau_{b_U} H(\tau_{b_U}))) + 2\gamma^2 w_{\max}^2 \cdot O(\sum_{b=0}^{B-1} \tau_b H(\tau_b)).$$

Substituting the equalities into the summation of (35) for $b = 0, \ldots, B-1$ yields

$$\sum_{b=0}^{B-1} (w_b^K(i^b))^2 = \exp(O(U)) O(\tau_{b_U} H(\tau_{b_U})) + O(\sum_{b=0}^{B-1} \tau_b H(\tau_b)) + O(\sum_{b=0}^{B-1} (\tau_b)^2). \tag{36}$$

Notice that taking expectation of $(w_b^K(i^b))^2$ with respect to $K_{b-1:0}$ does not affect the inequality. Finally, $\tau_{b_U} \leq \tau_{B-1}$ and $H(\tau_b) = o(\tau_b)$ completes the proof. $\square$

**Lemma D.7.** *In Algorithm 1, for the best stabilizing controller $i^* \in \mathcal{S}$, we have*

$$\mathbb{E}_{K_{B-1:0}} \sum_{b=0}^{B-1} \sum_{t=t_b}^{t_{b+1}-1} \left[ \frac{c_t(x_t^K(i^*), u_t^K(i^*))}{(\alpha_b)^{2s_b}} - c_t(x_t^*, u_t^*) \right] \leq O(U) + O(\sum_{b=0}^{B-1} H(\tau_b)).$$

*Proof.* Since $x_t^*$ is generated by a stabilizing controller, we have

$$\|x_t^*\| \leq \beta(t)\|x_0\| + \gamma w_{\max} \leq \beta(0)\|x_0\| + \gamma w_{\max}$$

$$\|x_t^*\|^2 \leq 2\beta(t)^2 \|x_0\|^2 + 2\gamma^2 w_{\max}^2 \leq 2\beta(0)^2 \|x_0\|^2 + 2\gamma^2 w_{\max}^2,$$

where the inequalities are by Cauchy-Schwarz inequality and the non-increasing property of $\beta(\cdot)$. Then, by (9), (27), and (33), we have

$$c_t(x_t^*, u_t^*) \leq 2L_{c_1}(\|x_t^*\|^2 + \|u_t^*\|^2) + L_{c2}(\|x_t^*\| + \|u_t^*\|) + c_{0,\max}$$

$$\leq 2L_{c_1}((1 + 2L_\pi^2)\|x_t^*\|^2 + 2\pi_{0,\max}^2) + L_{c2}((1 + L_\pi)\|x_t^*\| + \pi_{0,\max}) + c_{0,\max}$$

$$\leq 4L_{c1}(1 + 2L_\pi^2)(\beta(0))^2 \|x_0\|^2 + L_{c2}(1 + L_\pi)\beta(0)\|x_0\| + 4L_{c1}(1 + 2L_\pi^2)\gamma^2 w_{\max}^2$$

$$+ L_{c2}(1 + L_\pi)\gamma w_{\max} + 4L_{c1}\pi_{0,\max}^2 + L_{c2}\pi_{0,\max} + c_{0,\max} := M_6. \tag{37}$$

In Algorithm 1, one can write

$$\left\| \frac{x_{t_b}}{(\alpha_b)^{s_b}} \right\| \leq \frac{(\alpha_b)^{s_b+1}\|x_0\| + \delta}{(\alpha_b)^{s_b}} \leq \alpha_b \|x_0\| + \delta \tag{38}$$

$$\left\| \frac{x_t^*}{(\alpha_b)^{s_b}} \right\| \leq \frac{\beta(t)\|x_0\| + \gamma w_{\max}}{(\alpha_b)^{s_b}} \leq \beta(0)\|x_0\| + \gamma w_{\max}, \tag{39}$$

where the equalities hold for the last inequalities of (38) and (39) when $s_b = 0$.

By Assumption 2.2, for the best stabilizing controller $i^* \in \mathcal{S}$ and for $t_b \leq t < t_{b+1}$, we have

$$\frac{1}{(\alpha_b)^{2s_b}} |c_t(x_t^K(i^*), u_t^K(i^*)) - c_t(x_t^*, u_t^*)|$$

$$\leq \frac{1}{(\alpha_b)^{2s_b}}(L_{c1}(\max\{\|x_t^K(i^*)\|, \|x_t^*\|\} + \max\{\|u_t^K(i^*)\|, \|u_t^*\|\}) + L_{c2})(\|x_t^K(i^*) - x_t^*\| + \|u_t^K(i^*) - u_t^*\|)$$

$$\leq \frac{1}{(\alpha_b)^{2s_b}}(L_{c1}((1+L_\pi)\max\{\|x_t^K(i^*)\|, \|x_t^*\|\} + \pi_{0,\max}) + L_{c2})(1+L_\pi)\|x_t^K(i^*) - x_t^*\|$$

$$= (1+L_\pi)(L_{c1}(1+L_\pi)\max\{\left\|\frac{x_t^K(i^*)}{(\alpha_b)^{s_b}}\right\|, \left\|\frac{x_t^*}{(\alpha_b)^{s_b}}\right\|\} + \frac{L_{c1}\pi_{0,\max} + L_{c2}}{(\alpha_b)^{s_b}})\left\|\frac{x_t^K(i^*) - x_t^*}{(\alpha_b)^{s_b}}\right\|$$

$$\leq (1+L_\pi)(\beta(t-t_b)L_{c1}(1+L_\pi)\max\{\left\|\frac{x_{t_b}^K(i^*)}{(\alpha_b)^{s_b}}\right\|, \left\|\frac{x_{t_b}^*}{(\alpha_b)^{s_b}}\right\|\}$$

$$+ \frac{L_{c1}(1+L_\pi)\gamma w_{\max} + L_{c1}\pi_{0,\max} + L_{c2}}{(\alpha_b)^{s_b}}) \cdot \beta(t-t_b)\left\|\frac{x_{t_b}^K(i^*) - x_{t_b}^*}{(\alpha_b)^{s_b}}\right\|$$

$$\leq L_{c1}(1+L_\pi)^2\beta(t-t_b)^2\left(\left\|\frac{x_{t_b}}{(\alpha_b)^{s_b}}\right\| + \left\|\frac{x_{t_b}^*}{(\alpha_b)^{s_b}}\right\|\right)^2$$

$$+ \frac{L_{c1}(1+L_\pi)\gamma w_{\max} + L_{c1}\pi_{0,\max} + L_{c2}}{(\alpha_b)^{s_b}}(1+L_\pi)\beta(t-t_b)\left(\left\|\frac{x_{t_b}}{(\alpha_b)^{s_b}}\right\| + \left\|\frac{x_{t_b}^*}{(\alpha_b)^{s_b}}\right\|\right)$$

$$\leq L_{c1}(1+L_\pi)^2\beta(t-t_b)^2((\alpha_b + \beta(0))\|x_0\| + \delta + \gamma w_{\max})^2$$

$$+ (L_{c1}(1+L_\pi)\gamma w_{\max} + L_{c1}\pi_{0,\max} + L_{c2})(1+L_\pi)\beta(t-t_b)((\alpha_b + \beta(0))\|x_0\| + \delta + \gamma w_{\max})$$

$$\leq M_7\beta(t-t_b)^2 + M_8\beta(t-t_b), \tag{40}$$

where $M_7$ and $M_8$ are constants determined by $L_{c1}, L_{c2}, L_\pi, \pi_{0,\max}, \beta(0), \delta, \gamma, w_{\max}$ and $\max_{b \in \{0,1,...,B-1\}} \alpha_b$. Notice that $\alpha_b$ in Line 14 of Algorithm 1 is upper-bounded by some constant by Lemma C.5. The second inequality is by (9), the third inequality is due to Definition 2.4 and by leveraging the same stabilizing controller $i^*$ from $t_b$ for both trajectories $x_t^K(i^*)$ and $x_t^*$, the fourth inequality uses $x_{t_b}^K(i^*) = x_{t_b}$, and the fifth inequality is by (38) and (39). By combining (37) and (40), we have

$$\left|\frac{c_t(x_t^K(i^*), u_t^K(i^*))}{(\alpha_b)^{2s_b}} - c_t(x_t^*, u_t^*)\right| = \left|\frac{c_t(x_t^K(i^*), u_t^K(i^*))}{(\alpha_b)^{2s_b}} - \frac{c_t(x_t^*, u_t^*)}{(\alpha_b)^{2s_b}} - \frac{(\alpha_b)^{2s_b} - 1}{(\alpha_b)^{2s_b}}c_t(x_t^*, u_t^*)\right|$$

$$\leq \frac{1}{(\alpha_b)^{2s_b}}|c_t(x_t^K(i^*), u_t^K(i^*)) - c_t(x_t^*, u_t^*)| + \frac{(\alpha_b)^{2s_b} - 1}{(\alpha_b)^{2s_b}}c_t(x_t^*, u_t^*)$$

$$\leq \begin{cases} M_7\beta(t-t_b)^2 + M_8\beta(t-t_b), & \text{if } s_b = 0, \\ M_7\beta(t-t_b)^2 + M_8\beta(t-t_b) + M_6, & \text{if } s_b \neq 0. \end{cases}$$

Thus, one can conclude that

$$\sum_{b=0}^{B-1}\sum_{t=t_b}^{t_{b+1}-1}\left[\frac{c_t(x_t^K(i^*), u_t^K(i^*))}{(\alpha_b)^{2s_b}} - c_t(x_t^*, u_t^*)\right] \leq M_6|\mathcal{V}| + \sum_{b=0}^{B-1}(M_7 + M_8)H(t_{b+1} - t_b)$$

$$= O(U) + O(\sum_{b=0}^{B-1}H(\tau_b)), \tag{41}$$

where the first inequality uses $\beta(\cdot) \leq 1$ to derive $\sum_{t=t_b}^{t_{b+1}-1}[\beta(t-t_b)]^2 \leq \sum_{t=t_b}^{t_{b+1}-1}[\beta(t-t_b)] = H(t_{b+1} - t_b)$ and the last equality uses $t_{b+1} - t_b \leq \tau_b$ and Lemma D.3. Taking expectation of (41) with respect to $K_{B-1:0}$ completes the proof. $\square$

**Theorem D.8** (Restatement of Theorem 4.5, Regret Bound). *In Algorithm 1, suppose that* $\frac{\tau_1}{\tau_0}(\beta(\tau_0))^2 < \frac{1}{2\sqrt{2}}$. *Then, the regret bound is as follows:*

$$\mathbb{E}_{K_{B-1:0}}\sum_{t=0}^T[c_t(x_t, u_t) - c_t(x_t^*, u_t^*)]$$

$$= O(|\mathcal{U}|) + O(\sum_{b=0}^{B-1}H(\tau_b)) + \frac{\tilde{O}(|\mathcal{U}| + 1)}{\eta_0} + \frac{\eta_0 N}{2}[\exp(O(|\mathcal{U}|))O(\tau_{B-1}H(\tau_{B-1})) + O(\sum_{b=0}^{B-1}(\tau_b)^2)].$$

*Proof.* By Lemma D.1, we have

$$\mathbb{E}_{K_{B-1:0}} \sum_{t=0}^{T} c_t(x_t, u_t) = \mathbb{E}_{K_{B-1:0}} \sum_{b=0}^{B-1} \sum_{t=t_b}^{t_{b+1}-1} c_t(x_t, u_t) = \mathbb{E}_{K_{B-1:0}} \sum_{b=0}^{B-1} [w_b(K_b)]$$

$$= \mathbb{E}_{K_{B-1:0}} \sum_{b=0}^{B-1} [\mathbb{E}_{k \sim p_b}[w_b'(k)]]$$

$$\leq \frac{\tilde{O}(U+1)}{\eta_0} + \frac{\eta_0 N}{2} \sum_{b=0}^{B-1} \mathbb{E}_{K_{b-1:0}} (w_b^K(i^b))^2 + \mathbb{E}_{K_{B-1:0}} \sum_{l=0}^{|\mathcal{L}|} \sum_{b=b^l}^{b^{l+1}-1} \frac{w_b^K(i^*)}{(\alpha_b)^{2s_b}}$$

$$\leq \frac{\tilde{O}(U+1)}{\eta_0} + \frac{\eta_0 N}{2}[\exp(O(U))O(\tau_{B-1}H(\tau_{B-1})) + O(\sum_{b=0}^{B-1}(\tau_b)^2)]$$

$$+ O(U) + O(\sum_{b=0}^{B-1} H(\tau_b)) + \mathbb{E}_{K_{B-1:0}} \sum_{t=0}^{T} c_t(x^*, u^*),$$

where the first inequality is due to Lemma D.4 and D.5, and the last inequality is due to Lemma D.6 and D.7. Using $U \leq |\mathcal{U}|$ completes the proof. $\square$

**Theorem D.9** (Restatement of Theorem 4.6, Regret bound with known $|\mathcal{U}|$). *In Algorithm 1, let $\tau_0 = \lfloor (\frac{z}{N(|\mathcal{U}|+1)})^{1/2} \rfloor$ and $\tau_b = \lceil (\frac{(\nu b + z)}{N(|\mathcal{U}|+1)})^{1/2} \rceil$ for every $b \geq 1$ with the constants $z, \nu > 0$ that satisfies $\tau_0 > 0$ and $\frac{\tau_1}{\tau_0}(\beta(\tau_0))^2 < \frac{1}{2\sqrt{2}}$. Also, let $\eta_0 = O(\frac{(|\mathcal{U}|+1)^{2/3}}{T^{2/3}N^{1/3}})$. When $T \geq \max\{\frac{|\mathcal{U}|^{3/2}}{(N(|\mathcal{U}|+1))^{1/2}}, N(|\mathcal{U}|+1)\}$, we have*

$$\mathbb{E}_{K_{B-1:0}} \sum_{t=0}^{T} [c_t(x_t, u_t) - c_t(x_t^*, u_t^*)] = \tilde{O}(T^{2/3}N^{1/3}(|\mathcal{U}|+1)^{1/3})) + o(1)\exp(O(|\mathcal{U}|)) + o(T),$$

*which implies that we achieve a sublinear regret bound. Moreover, when $H(t) \leq O(\sum_{i=1}^{t} \frac{1}{i})$ for all $t \geq 1$, we have*

$$\mathbb{E}_{K_{B-1:0}} \sum_{t=0}^{T} [c_t(x_t, u_t) - c_t(x_t^*, u_t^*)] = [\tilde{O}(T^{2/3}) + \tilde{O}(T^{-1/3})\exp(O(|\mathcal{U}|))] N^{1/3}(|\mathcal{U}|+1)^{1/3}.$$

*Proof.* By the formulation of $(\tau_b)_{b \geq 0}$, we have

$$\sum_{b=0}^{B-1} \frac{(\nu b + z)^{1/2}}{(N(|\mathcal{U}|+1))^{1/2}} - 1 \leq \sum_{b=0}^{B-1} \tau_b = T \leq \sum_{b=0}^{B-1} \frac{(\nu b + z)^{1/2}}{(N(|\mathcal{U}|+1))^{1/2}} + (B-1),$$

where we can further use non-decreasing property of $(\cdot)^{1/2}$ to arrive at

$$\frac{z^{1/2} + \frac{2}{3\nu}[(\nu(B-1) + z)^{3/2} - z^{3/2}]}{(N(|\mathcal{U}|+1))^{1/2}} - 1 = \frac{z^{1/2} + \int_0^{B-1}(\nu b + z)^{1/2}db}{(N(|\mathcal{U}|+1))^{1/2}} - 1 \leq T$$

$$\leq \frac{\int_0^{B}(\nu b + z)^{1/2}db}{(N(|\mathcal{U}|+1))^{1/2}} + (B-1) = \frac{\frac{2}{3\nu}[(\nu B + z)^{3/2} - z^{3/2}]}{(N(|\mathcal{U}|+1))^{1/2}} + (B-1), \tag{42}$$

thus we have $B = O(T^{2/3}N^{1/3}(|\mathcal{U}|+1)^{1/3})$ from the first inequality and $T = O(B^{3/2}N^{-1/2}(|\mathcal{U}|+1)^{-1/2})$ from the second inequality and $T \geq N(|\mathcal{U}|+1)$. Similarly, we can find the order of $\sum_{b=0}^{B-1}(\tau_b)^2$ as follows:

$$\sum_{b=0}^{B-1}(\tau_b)^2 \leq \sum_{b=0}^{B-1}\left[\frac{(\nu b + z)^{1/2}}{(N(|\mathcal{U}|+1))^{1/2}} + 1\right]^2 \leq \int_0^{B}\left[\frac{(\nu b + z)}{(N(|\mathcal{U}|+1))} + \frac{2(\nu b + z)^{1/2}}{(N(|\mathcal{U}|+1))^{1/2}} + 1\right]db$$

$$= O(\frac{B^2}{N(|\mathcal{U}|+1)}) = O(T^{4/3}N^{-1/3}(|\mathcal{U}|+1)^{-1/3}), \tag{43}$$

where the last equality is by $B = O(T^{2/3}N^{1/3}(|\mathcal{U}|+1)^{1/3})$. We also have

$$\tau_{B-1} = \lceil (\frac{(\nu(B-1)+z)}{N(|\mathcal{U}|+1)})^{1/2} \rceil = O(B^{1/2}N^{-1/2}(|\mathcal{U}|+1)^{-1/2}) = O(T^{1/3}N^{-1/3}(|\mathcal{U}|+1)^{-1/3}). \tag{44}$$

Thus, we have

$$O(\tau_{B-1}H(\tau_{B-1})) = o((\tau_{B-1})^2) = o(T^{2/3}N^{-2/3}(|\mathcal{U}|+1)^{-2/3}) = \frac{o(1)}{\eta_0 N}, \tag{45}$$

where the first equality is due to Lemma C.1. With $T \geq \frac{|\mathcal{U}|^{3/2}}{(N(|\mathcal{U}|+1))^{1/2}}$, we have

$$\eta_0 N \exp(O(|\mathcal{U}|))O(\tau_{B-1}H(\tau_{B-1})) = o(1)\exp(O(|\mathcal{U}|)) \tag{46}$$

$$O(|\mathcal{U}|) = O(T^{2/3}N^{1/3}(|\mathcal{U}|+1)^{1/3}). \tag{47}$$

With (43), (45), (46), and (47), we can apply Theorem D.8 to derive

$$\mathbb{E}_{K_{B-1:0}} \sum_{t=0}^{T}[c_t(x_t,u_t)-c_t(x_t^*,u_t^*)] = \tilde{O}(T^{2/3}N^{1/3}(|\mathcal{U}|+1)^{1/3})+o(1)\exp(O(|\mathcal{U}|))+O(\sum_{b=0}^{B-1}H(\tau_b)).$$

Applying (14) to $O(\sum_{b=0}^{B-1}H(\tau_b))$ achieves a sublinear regret bound.

Moreover, when $\lim_{t\to\infty}H(t)<\infty$, there exists a constant $q_1$ that upper-bounds $H(t)$; i.e., $H(t) \leq q_1$ for all $t \geq 0$. Then, we have

$$\sum_{b=0}^{B-1}H(\tau_b) \leq q_1 B = O(B) = O(T^{2/3}N^{1/3}(|\mathcal{U}|+1)^{1/3}). \tag{48}$$

Also, (45) and (46) can be modified to

$$\tau_{B-1}H(\tau_{B-1}) \leq q_1\tau_{B-1} = O(T^{1/3}N^{-1/3}(|\mathcal{U}|+1)^{-1/3}),$$

$$\eta_0 N \exp(O(|\mathcal{U}|)))O(\tau_{B-1}H(\tau_{B-1})) = O(T^{-1/3}N^{1/3}(|\mathcal{U}|+1)^{1/3}) \cdot \exp(O(|\mathcal{U}|)). \tag{49}$$

Similarly, when $H(t) = O(\sum_{i=1}^{t}\frac{1}{i})$ for all $t \geq 1$, we have

$$\sum_{b=0}^{B-1}H(\tau_b) \leq BH(\tau_{B-1}) = O(B\log\tau_{B-1}) = \tilde{O}(T^{2/3}N^{1/3}(|\mathcal{U}|+1)^{1/3}), \tag{50}$$

$$\eta_0 N \exp(O(|\mathcal{U}|)))O(\tau_{B-1}H(\tau_{B-1})) = \tilde{O}(T^{-1/3}N^{1/3}(|\mathcal{U}|+1)^{1/3}) \cdot \exp(O(|\mathcal{U}|)). \tag{51}$$

Using (48), (49), (50), and (51) completes the proof. $\qquad\square$

Small modification provides the regret bound for the intermediate step "Dynamic Batching" mentioned in Appendix A.

**Corollary D.10.** *Consider "Dynamic Batching" strategy without adaptive learning rate, i.e. $s_b = 0$ for all $b = 0, \ldots, B-1$ in Algorithm 1. Let $\tau_0, \ldots, \tau_{B-1}$ and $\eta_0$ be the same quantity with Theorem D.9. When $T \geq \max\{\frac{|\mathcal{U}|^{3/2}}{(N(|\mathcal{U}|+1))^{1/2}}, N(|\mathcal{U}|+1)\}$, the term $o(1)\exp(O(|\mathcal{U}|))$ in the regret bound of Theorem D.9 is replaced by $o(T^{1/3})\exp(O(|\mathcal{U}|))$.*

*Proof.* Since $s_b = 0$ for all $b$, we need to modify Lemma D.7. Equation (40) is modified to

$$|c_t(x_t^K(i^*),u_t^K(i^*)) - c_t(x_t^*,u_t^*)| \leq L_{c1}(1+L_\pi)^2\beta(t-t_b)^2(\|x_{t_b}\|+\|x_{t_b}^*\|)^2 + L_{c1}(1+L_\pi)\gamma w_{\max}$$
$$+ L_{c1}\pi_{0,\max} + L_{c2}(1+L_\pi)\beta(t-t_b)(\|x_{t_b}\|+\|x_{t_b}^*\|),$$

which incurs

$$\sum_{t=t_b}^{t_{b+1}-1} |c_t(x_t^K(i^*), u_t^K(i^*)) - c_t(x_t^*, u_t^*)| \leq O(H(\tau_b)\|x_{t_b}\|^2) + O(H(\tau_b)\|x_{t_b}\|) + O(H(\tau_b)).$$

Thus, it follows that

$$\sum_{b=0}^{B-1} \sum_{t=t_b}^{t_{b+1}-1} |c_t(x_t^K(i^*), u_t^K(i^*)) - c_t(x_t^*, u_t^*)| \leq \exp(O(|\mathcal{U}|))(\|x_0\|^2 + \|x_0\|) + \exp(O(|\mathcal{U}|))H(\tau_{B-1})$$

$$= \exp(O(|\mathcal{U}|)) \cdot o(T^{1/3}),$$

where the last equality is by the choice of $\tau_{B-1} = O(B^{1/2}) = O(T^{1/3})$ and applying Lemma 4.3. This shows that $o(1)\exp(O(|\mathcal{U}|))$ in Theorem D.9 should be replaced by $o(T^{1/3})\exp(O(|\mathcal{U}|))$ in the algorithm without adaptive learning rate. $\qquad\square$

## E  REGRET PROOF FOR ALGORITHM 2

**Theorem E.1** (Restatement of Theorem 4.10, Regret bound with unknown $|\mathcal{U}|$). *In Algorithm 2, let $\tau_0 = \lfloor(\frac{z}{N})^{1/2}\rfloor$ and $\tau_b = \lceil(\frac{(\nu b+z)}{N})^{1/2}\rceil$ for every $b \geq 1$ with the constants $z, \nu > 0$ that satisfies $\tau_0 > 0$ and $\frac{\tau_1}{\tau_0}(\beta(\tau_0))^2 < \frac{1}{2\sqrt{2}}$. Also, let $\eta_0 = O(\frac{1}{T^{2/3}N^{1/3}})$ and $y = \frac{1}{2}$. When $T \geq \max\{\frac{|\mathcal{U}|^{3/2}}{N^{1/2}(|\mathcal{U}|+1)^{3/4}}, N\}$, we have*

$$\mathbb{E}_{K_{B-1:0}} \sum_{t=0}^{T} [c_t(x_t, u_t) - c_t(x_t^*, u_t^*)] = \tilde{O}(T^{2/3}N^{1/3}(|\mathcal{U}|+1)^{1/2}) + o(1)\exp(O(|\mathcal{U}|))(|\mathcal{U}|+1)^{1/2} + o(T),$$

*which implies that we achieve a sublinear regret bound. Moreover, when $H(t) \leq O(\sum_{i=1}^{t}\frac{1}{i})$ for all $t \geq 1$, we have*

$$\mathbb{E}_{K_{B-1:0}} \sum_{t=0}^{T} [c_t(x_t, u_t) - c_t(x_t^*, u_t^*)] = [\tilde{O}(T^{2/3}) + \tilde{O}(T^{-1/3})\exp(O(|\mathcal{U}|))]N^{1/3}(|\mathcal{U}|+1)^{1/2}$$

*Proof.* By the formulation of $(\tau_b)_{b\geq 0}$, as in (42), we can derive

$$B = O(T^{2/3}N^{1/3}) \quad \text{and} \quad T = O(B^{3/2}N^{-1/2})$$

when $T \geq N$. We can also obtain

$$\sum_{b=0}^{B-1} (\tau_b)^2 = O(T^{4/3}N^{-1/3}) \quad \text{and} \quad O(\tau_{B-1}H(\tau_{B-1})) = o(T^{2/3}N^{-2/3})$$

similar to (43) and (45). Now, define $\eta_{0,r} := \eta_0(r+1)^y = \eta_0\sqrt{r+1}$. Let $\mathcal{B}_r$ denote the set of batches where $\mu_b = r$; *i.e.*, $\mathcal{B}_r = \{0 \leq b \leq B-1, b \in \mathbb{Z}_+ : \mu_b = r\}$. Then, one can write

$$\frac{N}{2}\sum_{r=0}^{U}\sum_{b\in\mathcal{B}_r}\eta_{0,r}\mathbb{E}_{K_{b-1:0}}(w_b^K(i^b))^2 \leq \frac{\eta_0 N}{2}\sum_{r=0}^{U}\sum_{b\in\mathcal{B}_r}\sqrt{U+1}\mathbb{E}_{K_{b-1:0}}(w_b^K(i^b))^2$$

$$= \sqrt{U+1} \cdot O(T^{-2/3}N^{2/3})[\exp(O(U))O(\tau_{B-1}H(\tau_{B-1})) + O(\sum_{b=0}^{B-1}(\tau_b)^2)]$$

$$\leq \sqrt{|\mathcal{U}|+1} \cdot [o(1)\exp(O(|\mathcal{U}|)) + O(T^{2/3}N^{1/3})], \tag{52}$$

where the first equality holds by Lemma D.6 and the second inequality holds by $U \leq |\mathcal{U}|$.

Recall the definition and the cardinality of $\mathcal{L} = \{0 \leq b \leq B-1, b \in \mathbb{Z}_+ : s_{b+1} \neq s_b\}$ and $\mathcal{V} = \{0 \leq b \leq B-1, b \in \mathbb{Z}_+ : s_b \neq 0\}$ in Lemma D.3. We focus on the mix loss and the mixability gap with the denominator $\eta_{0,r}$; *i.e.*, $-\frac{1}{\eta_{0,r}}\log(\mathbb{E}_{k\sim p_b}\exp(-\eta_b w_b'(k)))$ and $\mathbb{E}_{k\sim p_b}[w_b'(k)]+$

$\frac{1}{\eta_{0,r}} \log(\mathbb{E}_{k \sim p_b} \exp(-\eta_b w_b'(k)))$. Considering that $\frac{\eta_b}{\eta_{0,r}}$ still remains to be $\frac{1}{(\alpha_b)^{2s_b}}$ as in Algorithm 1, Lemma D.4 can be modified to

$$\mathbb{E}_{K_{B-1:0}} \sum_{r=0}^{U} \sum_{b \in \mathcal{B}_r} -\frac{1}{\eta_{0,r}} \log(\mathbb{E}_{k \sim p_b} \exp(-\eta_b w_b'(k))) \leq \sum_{r=0}^{U} \frac{\rho_r^l \log N}{\eta_{0,r}} + \mathbb{E}_{K_{B-1:0}} \sum_{l=0}^{|\mathcal{L}|} \sum_{b=b^l}^{b^{l+1}-1} \frac{w_b^K(i^l)}{(\alpha_b)^{2s_b}}, \tag{53}$$

where $\rho_r^l$ denotes the number of batches in $\mathcal{B}_r \cap \mathcal{L}$. Similarly, considering that $\eta_{0,r}$ now depends on the value of $r$, Lemma D.5 can be modified to

$$\mathbb{E}_{K_{B-1:0}} \sum_{r=0}^{U} \sum_{b \in \mathcal{B}_r} \mathbb{E}_{k \sim p_b}[w_b'(k)] + \frac{1}{\eta_{0,r}} \log(\mathbb{E}_{k \sim p_b} \exp(-\eta_b w_b'(k)))$$

$$\leq \sum_{r=0}^{U} \frac{\rho_r^v}{2\eta_{0,r}} + \frac{N}{2} \sum_{r=0}^{U} \sum_{b \in \mathcal{B}_r} \eta_{0,r} \mathbb{E}_{K_{b-1:0}}(w_b^K(i^b))^2, \tag{54}$$

where $\rho_r^v$ denotes the number of batches in $\mathcal{B}_r \cap \mathcal{V}$. Now, our goal is to upper-bound $\sum_{r=0}^{U} \frac{\rho_r^l}{\eta_{0,r}} = \frac{1}{\eta_0} \sum_{r=0}^{U} \frac{\rho_r^l \log N}{\sqrt{r+1}}$ in (53) and $\sum_{r=0}^{U} \frac{\rho_r^v}{\eta_{0,r}} = \frac{1}{\eta_0} \sum_{r=0}^{U} \frac{\rho_r^v}{\sqrt{r+1}}$ in (54). It is straightforward to infer that $\rho_0^l + \rho_1^l + \cdots + \rho_U^l \leq 2U+1$ by Lemma D.3 and (24), which also leads to $\rho_0^l + \rho_1^l + \cdots + \rho_r^l \leq 2r+1$ for $r = 0, \ldots, U$. Similarly, we can infer that $\rho_0^v = 0$ and $\rho_1^v + \cdots + \rho_U^v \leq (2U-1) \lceil \frac{\log(\alpha_b + \frac{\delta}{\|x_0\|})}{-\log \beta(\tau_0)} \rceil$ by Lemma D.3 and (26), which also leads to $\rho_1^v + \cdots + \rho_r^v \leq (2r-1) \lceil \frac{\log(\alpha_b + \frac{\delta}{\|x_0\|})}{-\log \beta(\tau_0)} \rceil$ for $r = 1, \ldots, U$. Define $M_9 := \lceil \frac{\log(\alpha_b + \frac{\delta}{\|x_0\|})}{-\log \beta(\tau_0)} \rceil$ and consider the following maximization problems to get the upper bound.

$$l^* = \max_{\rho_0^l, \ldots, \rho_U^l} \sum_{r=0}^{U} \frac{\rho_r^l}{\sqrt{r+1}} \qquad\qquad v^* = \max_{\rho_1^v, \ldots, \rho_U^v} \sum_{r=1}^{U} \frac{\rho_r^v}{\sqrt{r+1}}$$

$$\text{s.t.} \quad \rho_0^l \leq 1 \qquad\qquad\qquad\qquad \text{s.t.} \quad \rho_1^v \leq M_9$$

$$\rho_0^l + \rho_1^l \leq 3 \qquad\qquad\qquad\qquad\quad \rho_1^v + \rho_2^v \leq 3M_9$$

$$\cdots \qquad\qquad\qquad\qquad\qquad\qquad \cdots$$

$$\rho_0^l + \rho_1^l + \cdots + \rho_U^l \leq 2U+1, \qquad\quad \rho_1^v + \rho_2^v + \cdots + \rho_U^v \leq (2U-1)M_9.$$

We can easily achieve an optimal point of each linear programming (LP) problem by the well-known Karush-Kuhn-Tucker (KKT) conditions. There exist positive constants $\lambda_0, \ldots, \lambda_U, \kappa_1, \ldots, \kappa_U$ such that

$$[1 \quad \frac{1}{\sqrt{2}} \quad \cdots \quad \frac{1}{\sqrt{U+1}}] = [\sum_{r=0}^{U} \lambda_r \quad \sum_{r=1}^{U} \lambda_r \quad \cdots \quad \lambda_U] \tag{55}$$

$$[\frac{1}{\sqrt{2}} \quad \frac{1}{\sqrt{3}} \quad \cdots \quad \frac{1}{\sqrt{U+1}}] = [\sum_{r=1}^{U} \kappa_r \quad \sum_{r=2}^{U} \kappa_r \quad \cdots \quad \kappa_U], \tag{56}$$

which yields $\lambda_U = \kappa_U = \frac{1}{\sqrt{U+1}}$, $\lambda_r = \kappa_r = \frac{1}{\sqrt{r+1}} - \frac{1}{\sqrt{r+2}} > 0$ for $r = 1, \ldots, U-1$, and $\lambda_0 = 1 - \frac{1}{\sqrt{2}}$. Since every dual variable is positive, complementary slackness tells that there is no slack for every inequality at the optimal solution. Thus, the optimal solutions are

$$\rho_0^l = 1, \quad \rho_r^l = 2, \quad r = 1, \ldots, U.$$
$$\rho_1^v = M_9, \quad \rho_r^v = 2M_9, \quad r = 2, \ldots, U,$$

where the corresponding optimal objective values are

$$l^* = 1 + \sum_{r=1}^{U} \frac{2}{\sqrt{r+1}} \leq 1 + \sqrt{2} + 2 \int_1^U \frac{1}{\sqrt{r+1}} dr = O(\sqrt{U+1})$$

$$v^* = \frac{M_9}{\sqrt{2}} + \sum_{r=2}^{U} \frac{2M_9}{\sqrt{r+1}} \leq \frac{M_9}{\sqrt{2}} + \frac{2M_9}{\sqrt{3}} + 2M_9 \int_2^U \frac{1}{\sqrt{r+1}} dr = O(\sqrt{U+1}),$$

where we leverage the non-increasing property of $\frac{1}{\sqrt{r+1}}$ for the inequalities. Thus, we have both $\frac{1}{\eta_0} \sum_{r=0}^{U} \frac{\rho_r^l \log N}{\sqrt{r+1}} = \tilde{O}(T^{2/3} N^{1/3} (U+1)^{1/2})$ and $\frac{1}{\eta_0} \sum_{r=0}^{U} \frac{\rho_r^v}{\sqrt{r+1}} = O(T^{2/3} N^{1/3} (U+1)^{1/2})$. Combining (52), (53), and (54) with Lemma D.7 and $U \leq |\mathcal{U}|$, one can write

$$\mathbb{E}_{K_{B-1:0}} \sum_{t=0}^{T} [c_t(x_t, u_t) - c_t(x_t^*, u_t^*)]$$

$$= \tilde{O}(T^{2/3} N^{1/3} (|\mathcal{U}|+1)^{1/2}) + o(1) \exp(O(|\mathcal{U}|))(|\mathcal{U}|+1)^{1/2} + O(|\mathcal{U}|) + O(\sum_{b=0}^{B-1} H(\tau_b))$$

$$= \tilde{O}(T^{2/3} N^{1/3} (|\mathcal{U}|+1)^{1/2}) + o(1) \exp(O(|\mathcal{U}|))(|\mathcal{U}|+1)^{1/2} + O(\sum_{b=0}^{B-1} H(\tau_b)),$$

where the second equality holds when $T \geq \frac{|\mathcal{U}|^{3/2}}{N^{1/2}(|\mathcal{U}|+1)^{3/4}}$. Using (14) shows a sublinear regret bound. When $H(t) \leq O(\sum_{i=1}^{t} \frac{1}{i})$ for all $t \geq 1$, (48) and (49) are modified to

$$\sum_{b=0}^{B-1} H(\tau_b) \leq O(BH(\tau_{B-1})) = \tilde{O}(T^{2/3} N^{1/3}),$$

$$\tau_{B-1} H(\tau_{B-1}) \leq \tau_{B-1} O(\log(\tau_{B-1})) = \tilde{O}(T^{1/3} N^{-1/3}),$$

$$\eta_0 N \exp(O(|\mathcal{U}|))) O(\tau_{B-1} H(\tau_{B-1})) = \tilde{O}(T^{-1/3} N^{1/3}) \cdot \exp(O(|\mathcal{U}|)).$$

Applying this equality to re-derive (52) completes the proof. $\qquad\square$

## F  APPLICATIONS: SWITCHED SYSTEMS

So far, we have used the best stabilizing controller $i^* \in \mathcal{S}$ for all time steps $t = 0, \ldots, T$ as the baseline of regret. However, the proofs of the theorems stated above imply one can even use any set of controllers $\{i^0, i^1, \ldots\} \subseteq \mathcal{S}$ as a baseline, where the controller is switched from $i^l$ to $i^{l+1}$ whenever the cumulative weight $W(\cdot)$ resets. This motivates the application of our DBAR algorithm to scenarios such as the switched systems (Tousi et al., 2008; Zhao et al., 2022) for which the transition dynamics and the associated controller pool may undergo changes, as well as the ballooning problem (Ghalme et al., 2021) where the controller pool may expand up to some finite set. We propose Algorithm 3, the switching version of DBAR, which resets the weight whenever the system is faced with a finite number of $O(U)$ switches. Here, we consider the regret with switching costs where the unit cost $d \geq 1$ is additionally incurred when the controller is switched; i.e., $d \sum_{t=1}^{T} \mathcal{I}_{(i_t \neq i_{t-1})}$ done in Altschuler & Talwar (2018) and Arora et al. (2019).

For an event $A$, $\mathcal{I}_{(A)}$ denotes an indicator function, where $\mathcal{I}_{(A)} = 1$ if an event $A$ occurs and $\mathcal{I}_{(A)} = 0$ otherwise. $Pr(A)$ denotes the probability of an event $A$. Let $x_t'$ and $u_t'$ denote the state and action sequence generated by our set of best stabilizing controllers $\{i_0', \ldots, i_{|\mathcal{L}|}'\} \subseteq \mathcal{S}$. We consider a regret with switching cost where the unit switching cost is $d \geq 1$; i.e., $\mathbb{E}_{K_{B-1:0}} \big[ \sum_{t=0}^{T} [c_t(x_t, u_t) - c_t(x_t', u_t')] + d \sum_{b=1}^{B-1} \mathcal{I}_{(K_b \neq K_{b-1})} - d \sum_{l=1}^{|\mathcal{L}|} \mathcal{I}_{(i_l' \neq i_{l-1}')} \big]$.

Algorithm 3 can easily be generalized to the situation where we have $O(U)$ number of system switches or controller pool switches. In fact, we can simply add $i_{|\mathcal{L}|+1}', \ldots, i_{|\mathcal{L}|+O(U)}' \in \mathcal{S}$ to the set of best stabilizing controllers $\{i_0', \ldots, i_{|\mathcal{L}|}'\} \subseteq \mathcal{S}$, where $|\mathcal{L}| = O(U)$ by Lemma D.3. Thus, it suffices to derive the regret bound of Algorithm 3, even in the context of general switched systems or ballooning problem. We first provide a useful lemma to construct a regret bound.

**Lemma F.1.** *In Algorithm 3, let $\tau_0 = \lfloor (\frac{z}{N(|\mathcal{U}|+1)})^{1/2} \rfloor$ and $\tau_b = \lceil (\frac{(\nu b+z)}{N(|\mathcal{U}|+1)})^{1/2} \rceil$ for every $b \geq 1$ with the constants $z, \nu > 0$ that satisfies $\tau_0 > 0$ and $\frac{\tau_1}{\tau_0}(\beta(\tau_0))^2 < \frac{1}{2\sqrt{2}}$. When*

---

**Algorithm 3** DBAR-switching

---

// Modification: Use this IF-ELSE Statement to select the current policy in Line 2 in Algorithm 1.

    **if** $b > 0$ and $s_b = s_{b-1}$ and $\mathcal{P}_b = \mathcal{P}_{b-1}$ **then**

        Pick $K_b = K_{b-1}$ with probability $\frac{\exp(-\eta_b W_b(K_{b-1}))}{\exp(-\eta_{b-1} W_{b-1}(K_{b-1}))}$. Sample $K_b$ from a distribution $p_b$ with

        probability $1 - \frac{\exp(-\eta_b W_b(K_{b-1}))}{\exp(-\eta_{b-1} W_{b-1}(K_{b-1}))}$.

    **else**

        Sample $K_b$ from a distribution $p_b$. Terminate the algorithm if $\mathcal{P}_b$ is empty.

    **end if**

---

$T \geq \frac{(o(1)exp(O(|\mathcal{U}|)))^{3/2}}{(N(|\mathcal{U}|+1))^{1/2}}$, we have

$$\mathbb{E}_{K_{B-1:0}} \sum_{b=1}^{B-1} \mathcal{I}_{(K_b \neq K_{b-1})} = O(|\mathcal{U}|) + O(\eta_0 N T).$$

*Proof.* For all $b = 1, \dots, B-1$ such that $s_b = s_{b-1}$, given $K_{b-1}, \dots, K_0$, we have

$$Pr(K_b \neq K_{b-1}) \leq 1 - \frac{\exp(-\eta_b W_b(K_{b-1}))}{\exp(-\eta_{b-1} W_{b-1}(K_{b-1}))} \leq 1 - \frac{\exp(-\eta_{b-1} W_b(K_{b-1}))}{\exp(-\eta_{b-1} W_{b-1}(K_{b-1}))}$$

$$= 1 - \exp(-\eta_{b-1} w'_{b-1}(K_{b-1})) \leq 1 - \exp(-\eta_0 w'_{b-1}(K_{b-1}))$$

$$\leq \eta_0 w'_{b-1}(K_{b-1}) = \eta_0 \frac{w_{b-1}(K_{b-1})}{p_{b-1}(K_{b-1})}, \tag{57}$$

where the second inequality is because $\eta_b = \eta_{b-1}$ when $s_b = s_{b-1}$, the third inequality uses $\eta_0 \geq \eta_b$ for all $b \geq 0$, and the last inequality uses $1 + x \leq e^x$ for all $x \in \mathbb{R}$. Now, given a set of controllers $i^b \in \mathcal{P}_b$ for $b = 0, \dots, B-1$, we can upper-bound $\sum_{b=0}^{B-2} w_b(i^b)$ by $t_{b+1} - t_b \leq \tau_b$ as follows:

$$\sum_{b=0}^{B-2} w_b(i^b) = \sum_{b=0}^{B-2} \sum_{t=t_b}^{t_{b+1}-1} c_t(x_t, u_t) \leq \sum_{b=0}^{B-2} \sum_{t=t_b}^{t_{b+1}-1} 2L_{c_1}(\|x_t\|^2 + \|u_t\|^2) + L_{c2}(\|x_t\| + \|u_t\|) + c_{0,\max}$$

$$\leq \sum_{b=0}^{B-2} \sum_{t=t_b}^{t_{b+1}-1} 2L_{c_1}((1 + 2L_\pi^2)\|x_t\|^2 + 2\pi_{0,\max}^2) + L_{c2}((1 + L_\pi)\|x_t\| + \pi_{0,\max}) + c_{0,\max}$$

$$\leq \sum_{b=0}^{B-2} 2L_{c_1}(1 + 2L_\pi^2)H(\tau_b)\|x_{t_b}\|^2 + L_{c2}(1 + L_\pi)H(\tau_b)\|x_{t_b}\| + \tau_b[4L_{c_1}\pi_{0,\max}^2 + L_{c2}\pi_{0,\max} + c_{0,\max}]$$

$$= O(\exp(O(|\mathcal{U}|))H(\tau_{b_U})) + O(\sum_{b=0}^{B-2} H(\tau_b)) + O(\sum_{b=0}^{B-2} \tau_b), \tag{58}$$

where the first inequality is due to (27), the second inequality is by (9) and (33), the third inequality is due to using $\beta(\cdot) \leq 1$ to derive $\sum_{t=0}^{t_{b+1}-t_b}[\beta(t)]^2 \leq \sum_{t=0}^{t_{b+1}-t_b}[\beta(t)] = H(t_{b+1} - t_b)$, and the last equality can be derived in the same fashion with (36). With $T \geq \frac{(o(1)exp(O(|\mathcal{U}|)))^{3/2}}{(N(M+1))^{1/2}}$, we obtain by (44) that

$$O(\exp(O(|\mathcal{U}|))H(\tau_{b_U})) + O(\sum_{b=0}^{B-2} H(\tau_b)) + O(\sum_{b=0}^{B-2} \tau_b) \leq O(T). \tag{59}$$

Thus, one can write

$$\mathbb{E}_{K_{B-1:0}} \sum_{b=1}^{B-1} \mathcal{I}_{(K_b \neq K_{b-1})} = \sum_{b=1}^{B-1} \mathbb{E}_{K_{b:0}} \mathcal{I}_{(K_b \neq K_{b-1})} = \sum_{b=1}^{B-1} \mathbb{E}_{K_{b-1:0}} \mathbb{E}_{K_b}[\mathcal{I}_{(K_b \neq K_{b-1})} \mid K_{b-1:0}]$$

$$= \sum_{b=1}^{B-1} \mathbb{E}_{K_{b-1:0}} Pr(K_b \neq K_{b-1} \mid K_{b-1:0})$$

$$= \sum_{b=1}^{B-1} \mathbb{E}_{K_{b-1:0}}[Pr(s_b = s_{b-1}, \mathcal{P}_b = \mathcal{P}_{b-1} \mid K_{b-1:0}) Pr(K_b \neq K_{b-1} \mid s_b = s_{b-1}, \mathcal{P}_b = \mathcal{P}_{b-1}, K_{b-1:0})$$
$$+ Pr(s_b \neq s_{b-1} \text{ or } \mathcal{P}_b \neq \mathcal{P}_{b-1} \mid K_{b-1:0}) Pr(K_b \neq K_{b-1} \mid s_b \neq s_{b-1} \text{ or } \mathcal{P}_b \neq \mathcal{P}_{b-1}, K_{b-1:0})]$$

$$= |\mathcal{L}| + U + \sum_{b=1}^{B-1} \mathbb{E}_{K_{b-1:0}}[Pr(s_b = s_{b-1}, \mathcal{P}_b = \mathcal{P}_{b-1} \mid K_{b-1:0}) Pr(K_b \neq K_{b-1} \mid s_b = s_{b-1}, \mathcal{P}_b = \mathcal{P}_{b-1}, K_{b-1:0})$$

$$\leq |\mathcal{L}| + U + \sum_{b=1}^{B-1} \mathbb{E}_{K_{b-1:0}} Pr(K_b \neq K_{b-1} \mid s_b = s_{b-1}, \mathcal{P}_b = \mathcal{P}_{b-1}, K_{b-1:0})$$

$$\leq |\mathcal{L}| + U + \sum_{b=1}^{B-1} \mathbb{E}_{K_{b-1:0}} \eta_0 \frac{w_{b-1}(K_{b-1})}{p_{b-1}(K_{b-1})}$$

$$= |\mathcal{L}| + U + \sum_{b=1}^{B-1} \mathbb{E}_{K_{b-2:0}} \mathbb{E}_{K_{b-1}} \left[ \eta_0 \frac{w_{b-1}(K_{b-1})}{p_{b-1}(K_{b-1})} \mid K_{b-2:0} \right]$$

$$= |\mathcal{L}| + U + \sum_{b=1}^{B-1} \eta_0 \mathbb{E}_{K_{b-2:0}} \sum_{K_{b-1} \in \mathcal{P}_{b-1}} p_{b-1}(K_{b-1}) \frac{w_{b-1}(K_{b-1})}{p_{b-1}(K_{b-1})}$$

$$\leq |\mathcal{L}| + U + \sum_{b=1}^{B-1} \eta_0 N \mathbb{E}_{K_{b-2:0}} w_{b-1}(i^{b-1}) \tag{60}$$

for the controller $i^{b-1} = \arg\max_{i \in \mathcal{P}_{b-1}} w_{b-1}(i)$. The first equality is because $K_{B-1}, \ldots, K_{b+1}$ does not affect on $\mathcal{I}_{(K_b \neq K_{b-1})}$ and the second inequality is by (57). Taking expectation of (58) with respect to $K_{b-1:0}$ and applying it to (60) yields

$$\mathbb{E}_{K_{B-1:0}} \sum_{b=1}^{B-1} \mathcal{I}_{(K_b \neq K_{b-1})} = |\mathcal{L}| + U + O(\eta_0 NT)$$

by (59). Using $|\mathcal{L}| = O(U)$ in Lemma D.3 and $U \leq |\mathcal{U}|$ completes the proof. $\square$

Algorithm 3 uses the same distribution with Algorithm 1 if $b = 0$ or $s_b \neq s_{b-1}$ or $\mathcal{P}_b \neq \mathcal{P}_{b-1}$. It turns out that even if $s_b = s_{b-1}$ and $\mathcal{P}_b = \mathcal{P}_{b-1}$, the distribution of policy from Algorithm 1 and 3 are indeed the same, which is motivated by Anava et al. (2015). For the sake of completeness, we state the lemma in this paper.

**Lemma F.2.** *Let $p_b$ and $\tilde{p}_b$ denote the distribution of policy at batch $b = 0, \ldots, B-1$ resulting from Algorithm 1 and 3, respectively. Then, $p$ and $\tilde{p}$ are the same distribution.*

*Proof.* For $b = 0$, $p_0(k) = \tilde{p}_0(k) = \frac{1}{N}$ for all $k \in \mathcal{P}_0$. For all $b = 1, \ldots, B-1$ such that $s_b \neq s_{b-1}$ or $\mathcal{P}_b \neq \mathcal{P}_{b-1}$, it holds that $p_b = \tilde{p}_b$. Thus, it suffices to prove the induction step for $b = 1, \ldots, B-1$ such that $s_b = s_{b-1}$ and $\mathcal{P}_b = \mathcal{P}_{b-1}$. Define $Y_b := \sum_{k \in \mathcal{P}_b} \exp(-\eta_b W_b(k))$ and suppose that $p_{b-1} = \tilde{p}_{b-1}$. Thus, we have

$$\tilde{p}_b(k) = \tilde{p}_{b-1}(k) \cdot \frac{\exp(-\eta_b W_b(k))}{\exp(-\eta_{b-1} W_{b-1}(k))} + p_b(k) \cdot \sum_{i \in \mathcal{P}_b} (1 - \frac{\exp(-\eta_b W_b(i))}{\exp(-\eta_{b-1} W_{b-1}(i))}) \cdot \tilde{p}_{b-1}(i)$$

$$= p_{b-1}(k) \cdot \frac{\exp(-\eta_b W_b(k))}{\exp(-\eta_{b-1} W_{b-1}(k))} + p_b(k) \cdot \sum_{i \in \mathcal{P}_b} (1 - \frac{\exp(-\eta_b W_b(i))}{\exp(-\eta_{b-1} W_{b-1}(i))}) \cdot p_{b-1}(i)$$

$$= \frac{\exp(-\eta_{b-1} W_{b-1}(k))}{Y_{b-1}} \cdot \frac{\exp(-\eta_b W_b(k))}{\exp(-\eta_{b-1} W_{b-1}(k))}$$
$$+ \frac{\exp(-\eta_b W_b(k))}{Y_b} \sum_{i \in \mathcal{P}_b} (1 - \frac{\exp(-\eta_b W_b(i))}{\exp(-\eta_{b-1} W_{b-1}(i))}) \frac{\exp(-\eta_{b-1} W_{b-1}(k))}{Y_{b-1}}$$

$$= \frac{\exp(-\eta_b W_b(k))}{Y_{b-1}} + \frac{\exp(-\eta_b W_b(k))}{Y_b} \sum_{i \in \mathcal{P}_b} \frac{\exp(-\eta_{b-1} W_{b-1}(i)) - \exp(-\eta_b W_b(i))}{Y_{b-1}}$$

$$= \frac{\exp(-\eta_b W_b(k))}{Y_{b-1}} + \frac{\exp(-\eta_b W_b(k))}{Y_b} \cdot \frac{Y_{b-1} - Y_b}{Y_{b-1}} = \frac{\exp(-\eta_b W_b(k)) \cdot Y_{b-1}}{Y_b \cdot Y_{b-1}} = p_b(k),$$

where the first equality is due to the law of total probability, the second equality is due to the induction hypothesis, and the fifth equality is by $\mathcal{P}_b = \mathcal{P}_{b-1}$. Notice that $s_b = s_{b-1}$ yields $\eta_b = \eta_{b-1}$ and $W_b(k) \geq W_{b-1}(k)$, and thus $0 \leq \frac{\exp(-\eta_b W_b(k))}{\exp(-\eta_{b-1} W_{b-1}(k))} \leq 1$; *i.e.*, the probability distribution is properly defined for every batch. This completes the proof. $\qquad\square$

**Theorem F.3** (Regret with switching costs bound with known $|\mathcal{U}|$). *In Algorithm 3, let* $\tau_0 = \lfloor(\frac{z}{N(|\mathcal{U}|+1)})^{1/2}\rfloor$ *and* $\tau_b = \lceil(\frac{(\nu b + z)}{N(|\mathcal{U}|+1)})^{1/2}\rceil$ *for every* $b \geq 1$ *with the constants* $z, \nu > 0$ *that satisfies* $\tau_0 > 0$ *and* $\frac{\tau_1}{\tau_0}(\beta(\tau_0))^2 < \frac{1}{2\sqrt{2}}$. *Also, let* $\eta_0 = O(\frac{(|\mathcal{U}|+1)^{2/3}}{T^{2/3}N^{1/3}d^{1/3}})$. *When* $T \geq \max\{\frac{(o(1)\exp(O(|\mathcal{U}|)))^{3/2}}{(N(|\mathcal{U}|+1))^{1/2}}, \frac{|\mathcal{U}|^{3/2}d}{(N(|\mathcal{U}|+1))^{1/2}}, N(|\mathcal{U}|+1)d\}$, *we have*

$$\mathbb{E}_{K_{B-1:0}}\left[\sum_{t=0}^{T}[c_t(x_t, u_t) - c_t(x_t', u_t')] + d\sum_{b=1}^{B-1}\mathcal{I}_{(K_b \neq K_{b-1})} - d\sum_{l=1}^{|\mathcal{L}|}\mathcal{I}_{(i_l' \neq i_{l-1}')}\right]$$
$$= \tilde{O}(T^{2/3}N^{1/3}(|\mathcal{U}|+1)^{1/3}d^{1/3}) + o(T),$$

*which implies that we achieve a sublinear regret bound. Moreover, when* $\lim_{t\to\infty} H(t) < \infty$ *and* $T \geq \max\{\frac{\exp(O(|\mathcal{U}|))}{d^{2/3}}, \frac{|\mathcal{U}|^{3/2}d}{(N(|\mathcal{U}|+1))^{1/2}}, N(|\mathcal{U}|+1)d\}$, *we have*

$$\mathbb{E}_{K_{B-1:0}}\left[\sum_{t=0}^{T}[c_t(x_t, u_t) - c_t(x_t', u_t')] + d\sum_{b=1}^{B-1}\mathcal{I}_{(K_b \neq K_{b-1})} - d\sum_{l=1}^{|\mathcal{L}|}\mathcal{I}_{(i_l' \neq i_{l-1}')}\right] = \tilde{O}(T^{2/3}N^{1/3}(|\mathcal{U}|+1)^{1/3}d^{1/3}).$$

*Proof.* The distribution of policy is the same for Algorithm 1 and 3 by Lemma F.2. Thus, we can use Theorem D.8 with Lemma F.1 to achieve

$$\mathbb{E}_{K_{B-1:0}}\left[\sum_{t=0}^{T}[c_t(x_t, u_t) - c_t(x_t', u_t')] + d\sum_{b=1}^{B-1}\mathcal{I}_{(K_b \neq K_{b-1})} - d\sum_{l=1}^{|\mathcal{L}|}\mathcal{I}_{(i_l' \neq i_{l-1}')}\right]$$
$$\leq \frac{\tilde{O}(|\mathcal{U}|+1)}{\eta_0} + \frac{\eta_0 N}{2}[\exp(O(|\mathcal{U}|))O(\tau_{B-1}H(\tau_{B-1})) + O(\sum_{b=0}^{B-1}(\tau_b)^2)]$$
$$+ O(\sum_{b=0}^{B-1}H(\tau_b)) + O(d|\mathcal{U}|) + O(d\eta_0 NT), \quad (61)$$

since $d \geq 1$ and $\sum_{l=1}^{L}\mathcal{I}_{(i_l' \neq i_{l-1}')} \geq 0$. Notice that $(\tau_b)_{b\geq 0}$ is the same for Algorithm 1 and 3. Accordingly, we still have $B = O(T^{2/3}N^{1/3}(|\mathcal{U}|+1)^{1/3})$ by (42) and $T \geq N(|\mathcal{U}|+1)d \geq N(|\mathcal{U}|+1)$. We also still have (43) and (44). Thus, with $T \geq \frac{(o(1)\exp(O(|\mathcal{U}|)))^{3/2}}{(N(|\mathcal{U}|+1))^{1/2}}$ and $T \geq \frac{|\mathcal{U}|^{3/2}d}{(N(|\mathcal{U}|+1))^{1/2}}$, we obtain that

$$\eta_0 N \exp(O(|\mathcal{U}|))O(\tau_{B-1}H(\tau_{B-1})) = o(d^{-1/3})\exp(O(|\mathcal{U}|)) = O(T^{2/3}N^{1/3}(|\mathcal{U}|+1)^{1/3}d^{-1/3}).$$
$$O(d|\mathcal{U}|) = O(T^{2/3}N^{1/3}(|\mathcal{U}|+1)^{1/3}d^{1/3}).$$

Also, with $T \geq N(|\mathcal{U}|+1)d$, we have

$$O(d\eta_0 NT) = O(T^{2/3}N^{1/3}(|\mathcal{U}|+1)^{1/3}d^{1/3}).$$

Combining all the above equalities with (61), one can write

$$\mathbb{E}_{K_{B-1:0}}\left[\sum_{t=0}^{T}[c_t(x_t, u_t) - c_t(x_t', u_t')] + d\sum_{b=1}^{B-1}\mathcal{I}_{(K_b \neq K_{b-1})} - d\sum_{l=1}^{|\mathcal{L}|}\mathcal{I}_{(i_l' \neq i_{l-1}')}\right]$$
$$= O(T^{2/3}N^{1/3}(|\mathcal{U}|+1)^{1/3}d^{1/3}) + O(\sum_{b=0}^{B-1}H(\tau_b)).$$

Using (14) shows a sublinear regret bound. When $\lim_{t\to\infty} H(t) < \infty$, (49) is modified to

$$\eta_0 N \exp(O(|\mathcal{U}|))O(\tau_{B-1}H(\tau_{B-1})) = O(T^{2/3}N^{1/3}(|\mathcal{U}|+1)^{1/3}d^{1/3}),$$

only with $T \geq \frac{\exp(O(|\mathcal{U}|))}{d^{2/3}}$. This completes the proof. $\qquad\square$

## G   NUMERICAL EXPERIMENT DETAILS

In the two following subsections, we will present experiment details on linear and nonlinear systems, respectively. Since our Algorithm 1 only hinges on the system state norm as a context, we can avoid computational burden; thus, Apple M1 Chip with 8-Core CPU is sufficient for the experiments. The error bars (shaded area) in all the figures in the paper report 95% confidence intervals based on the standard error. We calculate the standard error by randomly sampling 100 seeds to consider the variability of our experimental results. The first factor of variability is the randomness of selecting the policy determined by the probability calculated in Algorithm 1. The second factor is the randomness of adversarial disturbances stated in each experiment. For example, sinusoidal noise does not involve any randomness but the uniform random walk contains the randomness in the difference between two consecutive noises.

### G.1   EXPERIMENTS FOR THE LINEAR SYSTEM

In this subsection, we introduce the implementation details and present more experiments on the linear system (3) discussed in Example 1 of Section 5.

We consider three different noises for the experiments. To perform a fair comparison, the bounding constant $w_{\max}$ is set to 1.

(a) Sanity check: Gaussian noise with mean 0.3 and standard deviation 0.1, truncated to $[-0.4, 1]$

(b) Sinusoidal noise $w_t = \left[ \sin\left(\dfrac{t}{5\pi}\right), \sin\left(\dfrac{t}{11\pi}\right) \right]'$

(c) Uniform random walk, where $w_0 = \text{Uniform}\left[ \dfrac{1}{3} - \dfrac{2}{3T}, \dfrac{1}{3} + \dfrac{2}{3T} \right]^2$

$$\text{and } w_t - w_{t-1} \text{ follows Uniform}\left[ -\dfrac{2}{3T}, \dfrac{2}{3T} \right]^2,$$

where $T$ is time horizon. One can easily see that for uniform random walk, $|w_T| \leq 1$ for any $T$. Notice that we use statistical (Gaussian) noise for the sanity check, and the rest are the adversarial disturbances.

We perform the ablation study of Algorithm 1, which means that we consider four scenarios: (fixed, dynamic) batch length and (fixed, adaptive) learning rate. For all the experiments implementing the algorithm, we use $T = 3000, \eta_0 = 0.025, \gamma = 2.5, \alpha_0 = 1.01$, and $x_0 = [100, 200]'$. For the dynamic batch length, we consider $\tau_0 = 11$ and $\tau_b = \lceil \tau_0 \cdot (\frac{b+10}{10})^{0.5} \rceil$. It is well known that every (asymptotically) stabilizing controller in the linear system is indeed exponentially stabilizing controller (Khalil, 2015). Hence, we use $\beta(t) = 0.99^t$ without relaxing the assumptions on stabilizing controllers. Finally, we use $\delta = \frac{\gamma w_{\max}}{1 - \beta(\tau_0)}$. Since the sinusoidal noise case is already presented in Figure 2, we only present truncated Gaussian noise case and uniform random walk case here.

In Figures 2, 5, and 6, we observe that each component of DBAR, a dynamic batch length and an adaptive learning rate, *jointly* improves both the stability and the regret regardless of the noise form. For example, a dynamic batch length delays the time that large state norms occur during learning, but does not necessarily stabilize that state norm by itself (see Figures 6(a) and 6(b)). However, when applied together with an adaptive learning rate, a potential multiplicative exponential term is mitigated (see Remark 4.8) and the state norm is thus stabilized. This can be observed in Figures 2(d), 5(d), and 6(d) when comparing fixed and dynamic batch lengths under an adaptive learning rate. This results from using a non-decreasing batch length where the increasing ratio between two consecutive batch lengths is determined to converge to 1 (see Assumption 3.1). On the other hand, an adaptive learning rate effectively lowers the state norm at the time that large state norms occur without delay, since the learning rate adaptively decreases whenever the agent faces large state norm. This can be seen in 2(c), 5(c), and 6(c), the ablation study about the comparison between fixed and adaptive learning rates under a dynamic batch length. Thus, DBAR effectively stabilizes the state norm below $\gamma w_{\max}$ and minimizes the regret, where the two components support each other.

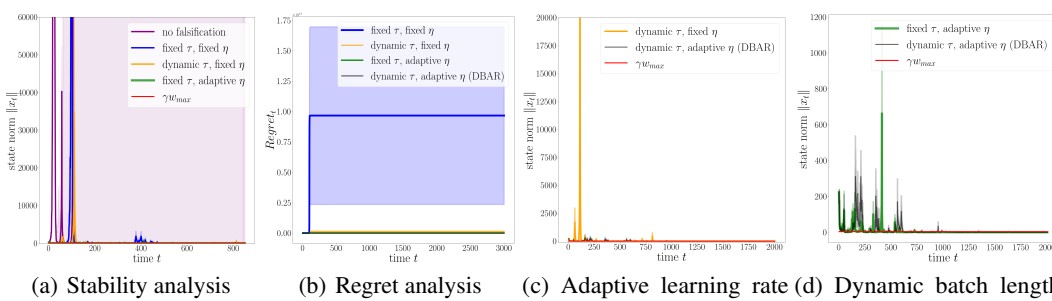

(a) Stability analysis  (b) Regret analysis  (c) Adaptive learning rate under dynamic batch length  (d) Dynamic batch length under adaptive learning rate

Figure 5: The stability and the regret in the linear system under truncated Gaussian noise. Ablation study of the algorithm is presented.

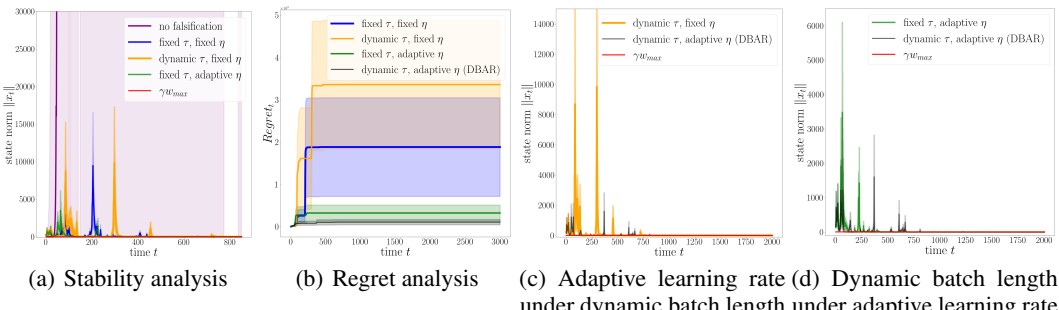

(a) Stability analysis  (b) Regret analysis  (c) Adaptive learning rate under dynamic batch length  (d) Dynamic batch length under adaptive learning rate

Figure 6: The stability and the regret in the linear system under Uniform random walk. Ablation study of the algorithm is presented.

## G.2 EXPERIMENTS FOR THE NONLINEAR SYSTEM

In this subsection, we introduce the implementation details and present more experiments on the nonlinear ball-beam system introduced in Example 2 of Section 5. To study this continuous-time nonlinear system, we first derive the first-order state representation of the leader system (4) with the states $(y_1, y_2, y_3, y_4) = (x, \dot{x}, -9.81B\theta, -9.81B\dot{\theta}) \in \mathbb{R}^4$ and the action $v = -9.81Bu_x$:

$$\dot{y_1} = y_2, \quad \dot{y_2} = 9.81B \sin\left(\frac{y_3}{9.81B}\right) + \frac{y_1 y_4^2}{B(9.81)^2} + 3w, \quad \dot{y_3} = y_4, \quad \dot{y_4} = v,$$

where $w_x$ is a sinusoidal noise $\sin\left(\frac{t}{7\pi}\right)$ and $w_{\max} = 1$. A nested saturating control policy is known to successfully stabilize the leader ball-beam system if the correct parameters are given, but it does not necessarily exponentially stabilize the system (Barbu et al., 1997). This necessitates our approach of extending the notion of stabilizing controllers beyond exponential assumptions. In this experiment, we aim to learn the parameters of the best stabilizing controller. We choose a nested saturating control policy $v'$ determined by three positive parameters $(p, k_1, k_2)$:

$$\epsilon = \frac{1}{\sqrt{1 + y_1^2 + y_2^2}}, \quad p_1 = p, \quad p_2 = \frac{p}{\epsilon}, \quad p_3 = \frac{p}{\epsilon^2}, \quad p_4 = \frac{p}{\epsilon^3},$$

$$z_1 = y_1 + k_1 y_2 + k_1 y_3 + y_4, \quad z_2 = y_2 + k_2 y_3 + y_4, \quad z_3 = y_3 + y_4, \quad z_4 = y_4,$$

$$v' = \sigma_{p_4}(z_4 + \sigma_{p_3}(z_3 + \sigma_{p_2}(z_2 + \sigma_{p_1}(z_1)))),$$

where $\sigma_p(z)$ is the saturating function defined as $p$ if $z > p$, $-p$ if $z < -p$, and $z$ if $|z| \leq p$. We consider the controller pool

$$V' = \{v' : p \in \{2, 16, 30, 44, 58, 72, 86, 100\}, \ k_1 \in \{2, 2.5, 3, 3.5, 4, 4.5, 5, 5.5, 6, 6.5\},$$
$$k_2 \in \{1, 1.5, 2, 2.5, 3, 3.5, 4, 4.5, 5, 5.5\}\},$$

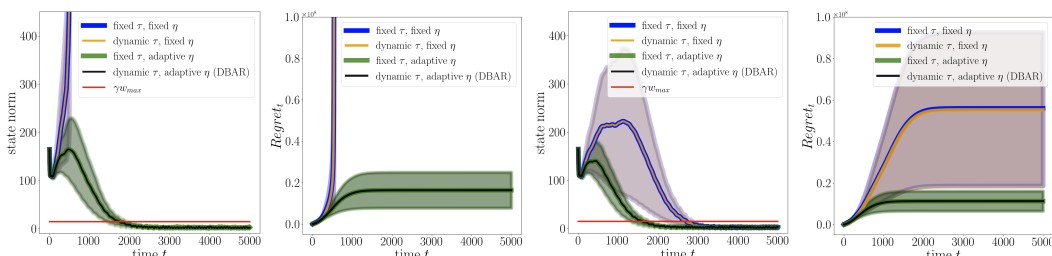

(a) Stability analysis: $\beta_1(t)$ (b) Regret analysis: $\beta_1(t)$ (c) Stability analysis: $\beta_2(t)$ (d) Regret analysis: $\beta_2(t)$

Figure 7: The stability and the regret in the noise-injected ball-beam system under sinusoidal noise and the choice of $\beta_1(t)$ or $\beta_2(t)$.

which has a total of 800 controllers. Moreover, the follower systems are formulated by generating $A, \tilde{A}$ in (5) as random matrices, where each entry is independently sampled from $\text{Unif}[0, 1]$. For the action $u_z$ of the follower systems, we consider a linear policy as in Example 1. Then, the action is parameterized by $u_z = K_z z$, where $K_z \in \mathbb{R}^{96 \times 96}$. We let $K_z$ be a multiple of identity matrix, and the diagonal entry is selected from the pool $\{-45, -47.5, -50, -52.5, -55, -60, -70, -80, -90, -100\}$. Thus, considering the actions of both leader and followers, the controller pool contains 8000 controllers. Among them, we do not know if each controller stabilizes the system.

For simplicity, we perform forward-Euler discretization on the system with a sampling time $0.01$. The resulting discrete-time states and actions are denoted by $[y^t, z^t]$ and $[v_y^t, v_z^t]$ at $t^{\text{th}}$ sampling time. We use the cost function $c_t(y^t, z^t, v_y^t, v_z^t) = \|y^t\|^2 + \|z^t\|^2 + \|v_y^t\|^2 + \|v_z^t\|^2$ to stabilize the ball position and the beam angle towards 0. We again perform the ablation study of Algorithm 1. For the experiments implementing the algorithm, we use $T = 5000$, $\eta_0 = 0.25$, $\gamma = 1.5$, $\alpha_0 = 1.01$, $y^0 = [-32, 24, 5.6, 24]$, and $z^0 = [10, 10, \dots] \in \mathbb{R}^{96}$. For the dynamic batch length, we consider $\tau_0 = 9$ and $\tau_b = \lceil \tau_0 \cdot (\frac{b+41}{40})^{0.5} \rceil$.

Unlike the choice of $\beta(t)$ in Section G.1, we select the stabilizing controller only to satisfy (asymptotic) ISS in Definition 2.3, instead of exponential ISS. To deeply study this notion, we consider different polynomially decreasing series (which is not exponentially decreasing) to be the candidates for $\beta(t)$:

$$\beta_1(t) = \min\left\{\frac{10}{t^{1.05}}, 1\right\}, \quad \beta_2(t) = \min\left\{\frac{10}{t^{1.08}}, 1\right\}.$$

Figures 3(b) and 3(c) show the stability and regret analysis of the system under $\beta_1(t)$. For the completeness, we present the same pictures in Figures 7(a) and 7(b).

In our experiment, there are 3400 controllers out of 8000 controllers that induces the system to explode, starting from the initial state. However, there exist far more destabilizing controllers within this pool, since most of 5600 controllers are only locally stabilizing controllers, meaning that the system is stabilized only at some initial states. With only few stabilizing controllers in the pool, Figure 7 illustrates that a dynamic batch length by itself still suffers from a multiplicative exponential term regarding a series of destabilizing controllers. However, for both $\beta_1(t)$ and $\beta_2(t)$, even though $H(t)$ and $O(\sum_{i=1}^{t} \frac{1}{i})$ are close, one can observe that the combination of the two components of DBAR effectively resolves this malignant term and the resulting closed-loop system enjoys both asymptotic system stability and the improved regret (see Table 1).

The behaviors of $\beta_1(t)$ and $\beta_2(t)$ are slightly different, in the sense that while DBAR still performs well, the system already appears stabilized even without some components of DBAR with $\beta_2(t)$. This stems from the amount of discarding the destabilizing controllers. $\beta_2(t)$ removes the controller with more strict criteria than $\beta_1(t)$ since $1.08 > 1.05$. This prevents the explosion of the nonlinear system by eliminating potential destabilizing controllers not yet seen in an unstable region in advance. However, in practice, if the given candidate controller set had not included any controller satisfying the strict assumptions, the algorithm would have *terminated*, failing to keep the system running. This finding again demonstrates why it is crucial to allow a broader class of controllers and still achieve a

tight regret bound. Moreover, the experimental results strongly support that our algorithm DBAR performs well for any choice of $\beta(t)$, which determines the scope of stabilizing controllers.

