# OpenReview forum: "Online Bandit Nonlinear Control with Dynamic Batch Length and Adaptive Learning Rate"
_ICLR.cc/2025/Conference — Submitted to ICLR 2025_

### Official Review · Reviewer_zi1A · 2024-10-17

**Soundness:** 3
**Presentation:** 2
**Contribution:** 2
**Rating:** 6
**Confidence:** 3

**Summary:**

This paper addresses the problem of online bandit nonlinear control, proposing a new algorithm called Dynamic Batch length and Adaptive learning Rate (DBAR). The algorithm in this paper is designed to handle a nonlinear dynamical system with a mix of stabilizing and destabilizing controllers, offering stability guarantees and regret bounds.

**Strengths:**

1. The paper provides rigorous theoretical analysis, including stability proofs and regret bounds, which demonstrate the effectiveness of the DBAR algorithm under weaker assumptions. The comparison with existing methods is clear and it strengthens the contribution.
2. The introduction of the DBAR algorithm relaxes the requirement for exponentially stabilizing controllers, allowing a broader range of controllers to stabilize the system. It also achieves better result for known and unknown $\mathcal{U}$.

**Weaknesses:**

1. While the theoretical analysis is thorough, the experiments are somewhat limited. The paper focuses on low-dimensional systems (a 2D linear system and a simple nonlinear system), and it would be beneficial to demonstrate the algorithm's performance in more complex, high-dimensional scenarios to validate the practical applicability of DBAR.
2. The writing of this paper is not very clear, especially in Section 2 and Section 3. I suggest using more references, formulas, and illustrations to explain the rationale behind the assumptions and the intuition behind the algorithm design, rather than relying solely on extensive text descriptions.

**Questions:**

1. Could you include more experimental results on complex systems?
2. Could you explain the necessity of Assumption 2.5? You may refer to previous literature to support your explanation.
3. Could you clarify the proof ideas and intuition behind Theorems 4.5 and 4.6? The current explanation is quite brief, and it's difficult to grasp the core proof strategy as well as how the earlier algorithm design relates to this proof. I suggest expanding this section and moving some of the lemma proofs, such as Lemma 4.7, to the appendix.

---

> ### Author Response · Authors · 2024-11-17
>
> We are very grateful to the reviewer for a thorough evaluation of our work and providing valuable feedback. We will improve the paper based on the provided comments.
>
> We have improved the paper and uploaded a new PDF file with two **important** revisions:
>
> 1. **Experiments**: We have replaced the experiments in Example $2$ with high-dimensional experiments (100 states) on leader-follower systems. The leader is represented by a previous ball-beam (nonlinear) system, and the followers leverage the leader's state to stabilize themselves. Please check the experimental setup in Example $2$ (see pages 9-10).
>
> 2. **Illustration**: In a new Figure $1$, we have illustrated a concept of the controller pool for the unknown nonlinear system, and how general an asymptotically stabilizing notion is. We hope that this figure helps the reader to understand the core concepts (see page 4, Figure $1$).
>
> Responses to the reviewer:
>
> **Weakness 1**: Thanks for the comment. We have now provided the experiments on high-dimensional systems in Example $2$.
>
> **Weakness 2**: Thank you for the comment. We will improve our presentation and strive to better illustrate our results. As a first step, we have presented Figure $1$ in Section $2$ to illustrate a concept of stabilizing notions and the controller pool for unknown system.
>
> **Question 1**: Yes, we have now provided 100-dimensional systems.
>
> **Question 2**: The assumptions such as Lipschitz continuous is quite mild. To understand Assumption $2.5$, we provide the following example (also see the pictorial illustration in Figure 1(b)). Given an unknown system, we can consider different regimes to model the possibilities of the unknown parameters of the system and then design a controller for each possibility. For example, consider a robot operating in an uncertain environment where there are different scenarios that could happen in the environment and we model it by a vector "$(a,b)$" for simplicity. We consider different intervals such as $[0,1)\times [0,1)$ or $[1,2)\times[1,2)$, etc. for "$(a,b)$" and then design a controller for each scenario. Now, we have a pool of controllers and we know at least one of them should work but do not know which one since we do not know the exact parameter of the system. If the intervals chosen for each "$a$" and "$b$" are too wide, a stabilizing controller may not exist and then Assumption $2.5$ will be violated. This assumption is always satisfied if we have a rich set of controllers in the given pool, as long as the system is stabilizable (if the system is not stabilizable, then its behavior cannot be controlled no matter what action we take). The core part of Assumption $2.5$ is the existence of a controller satisfying Definition $2.3$ and $2.4$, which means that we only require that at least one input-to-state stable (ISS) and incrementally stable (IS) controller "exists" in the candidate pool.
> The ISS and IS principles are a widely understood concept [1, 2] for handling disturbances in nonlinear systems, particularly in applications such as automotive and robotics systems, as it ensures that a vehicle or robot can perform safely despite variations in exogenous inputs like road conditions or obstacles.  Specifically, [3] assumed the existence of "exponentially" ISS and IS stabilizing controllers; the stabilizing behavior of such controllers should be exponentially fast, which may be quite restrictive for nonlinear systems to have such controllers. On the other hand, in our work, we only assume the existence of "asymptotically" ISS and IS stabilizing controllers whose stabilizing behavior can be arbitrarily slow.
> As mentioned in the paper (Lines 69-79), the difference between exponential stability and asymptotic stability is tantamount to the difference between convexity and strong convexity. As witnessed in many machine learning problems, strong convexity is a very restrictive assumption while convexity is more realistic.
> Therefore, the existence of an asymptotic ISS and IS controller is not a particularly restrictive assumption.
>
> Continued in the next rebuttal (references are also provided).

---

> ### Author Response · Authors · 2024-11-17
>
> **Question 3**: Thanks for the recommendation.
> We will defer Lemma $4.7$ to the appendix and add the following explanation.
>
> To clarify, our proof strategy starts from dividing the expected total cost into mix loss and mixability gap as in [4,5]. This technique is common in hedge setting, which is an instance of multi-armed bandit problems. Since the earlier algorithm relies on the "constant learning rate", they divide the expected cost into the $\text{mix loss}$ "$-\frac{1}{\eta_0}\log(\mathbb{E}\exp(-\eta_0 w_b'(k)))$" and the $\text{mixability gap}$ "$\mathbb{E} [w_b'(k)]-\text{mix loss}$" (notice that the denominator of the mix loss and the term inside the exponential function are both $\eta_0$, the constant learning rate).
> However, to deal with the absence of exponentially stabilizing controllers, we need to adopt an adaptive learning rate, thus the proof strategy should be modified to analyze $-\frac{1}{\eta_0}\log(\mathbb{E}_{k\sim p_b}\exp(-\eta_b w_b'(k)))$, where a new $\eta_b$ is an adaptively selected learning rate at batch $b$. Both the modified mix loss and the mixability gap lead to an additional term that is in $|L|$ and $|V|$. Both terms are on the order of $|U|$ (Lemma $4.7$), so they do not increase the regret compared to the earlier work [3].
>
> Note that our adaptive learning rate is determined by $\eta_b = \eta_0 / (\alpha_b)^{2s_b }$. Accordingly, while we obtain additional terms linear in $|U|$ due to adaptive learning rate, we benefit from having the term $\mathbb{E} \left[\sum_{b=0}^{B-1}\sum_{t=t_b}^{t_{b+1}-1} \bigr[\frac{c_t(x_t^K(i^*), u_t^K(i^*))}{(\alpha_b)^{2 s_b}} - c_t(x_t^*, u_t^*)\bigr]\right]$ instead of $\mathbb{E}\left[ \sum_{b=0}^{B-1}\sum_{t=t_b}^{t_{b+1}-1} \bigr[c_t(x_t^K(i^*), u_t^K(i^*)) - c_t(x_t^*, u_t^*)\bigr]\right]$ in a constant learning rate case.
> Since $(\alpha_b)^{2 s_b}$ increases (an adaptive learning rate decreases) when the state norm is in an unstable region, the term decreases to the extent that alleviates the multiplicative exponential term (see $o(T^{1/3})\cdot \exp(O(|U|))$ in Table $1$, "Dynamic Batching") that arises when using a dynamic batch length.
> In summary, our final regret has polynomial terms more in regret compared to the previous work, but instead alleviates multiplicative exponential term.
>
>
> [1] Sontag, "Input to state stability: Basic concepts and results", Nonlinear and Optimal Control Theory, 2008.
>
> [2] Khalil, "Nonlinear Systems", Pearson Education, 2015.
>
> [3] Li et al., "Online switching control with stability and regret guarantees", L4DC, 2023.
>
> [4] van Ervan et al., "Adaptive hedge", NeurIPS, 2011.
>
> [5] de Rooij et al., "Follow the leader if you can, hedge if you must", JMLR, 2014.
>
> Many thanks for reading our rebuttal.

---

> > ### Comment · Reviewer_zi1A · 2024-11-21
> > **Reply**
> >
> > Thank you for your response! I will maintain my positive score and recommend accepting this paper.

---

> > > ### Author Response · Authors · 2024-11-21
> > >
> > > Thanks for the feedback again! Please let us know if you have any further questions.

---

### Official Review · Reviewer_qZtK · 2024-10-24

**Soundness:** 3
**Presentation:** 3
**Contribution:** 3
**Rating:** 6
**Confidence:** 3

**Summary:**

In this paper, the authors study the online bandit nonlinear control, which aims to learn the best stabilizing controller from a pool of stabilizing and destabilizing controllers of unknown types for a given nonlinear dynamical system. They develop an algorithm, named Dynamic Batch length and Adaptive learning Rate (DBAR), and study its stability and regret. DBAR achieves better regret bounds than Exp3 under even weaker assumptions.

**Strengths:**

1. The problem of online bandit nonlinear control is well-motivated.

2. This paper weakens the assumptions in previous works and derive a stronger regret bound. The proof looks correct.

3. There are simulations to support the theoretical findings.

**Weaknesses:**

Main concerns:

1. It seems to me that the results heavily depend on the assumption that $H(t)$ has finite limitation (Thm 4.2) or $H(t)\leq O(\log t)$ (Thm 4.6), this assumption does not seem to be very mild ($H(t)$ has finite limitation is far from the conclusion in Lem 4.3). What will the result be like without such assumptions? Or is there any further justifications of such assumption?

2. In the experiments, the claim 'While the simulations are on low-dimensional systems for illustration purposes, similar observations can be made for high-dimensional systems.' is made. Is this a conjecture, or is there any experimental evidence for this? It seems not hard to do experiments with higher dimensionality.

Some questions and suggestions for writing:

1. The notation $|\mathcal{U}|$ first appears in the table, while it is not defined until Def 2.6.

2. The previous papers depend on 'exponential stability'. What is its definition and how is it stronger compared to 'asymptotic stability'?

3. I did not quite understand the motivation of Lemma 4.3. This is too weak to support the assumption $\lim_{t\rightarrow\infty}H(t)<\infty$.

4. Listing the steps to solve for $z,\nu$ in Theorem 4.6 could help justify the results.

**Questions:**

Please refer to the weakness part.

---

> ### Author Response · Authors · 2024-11-17
>
> We are very grateful to the reviewer for a thorough evaluation of our work and providing valuable feedback. We will improve the paper based on the provided comments.
>
> We have improved the paper and uploaded a new PDF file with two **important** revisions:
>
> 1. **Experiments**: We have replaced the experiments in Example $2$ with high-dimensional experiments (100 states) on leader-follower systems. The leader is represented by a previous ball-beam (nonlinear) system, and the followers leverage the leader's state to stabilize themselves. Please check the experimental setup in Example $2$ (see pages 9-10).
>
> 2. **Illustration**: In a new Figure $1$, we have illustrated a concept of the controller pool for the unknown nonlinear system, and how general an asymptotically stabilizing notion is. We hope that this figure helps the reader to understand the core concepts (see page 4, Figure $1$).
>
> Responses to the reviewer:
>
> **Main concern 1** ($H(t)$): Thank you for the comment. There may have been a confusion here since only some of our results depend on $H(t)$. In our paper, there are two types of theorems: those that depend on $H(t)$ and those that do not. The ones depending on $H(t)$ is Thm $4.2$ and the second part of Thm $4.6$. Thm $4.2$ implies finite-gain stability when $H(t) <\infty$, and the second part of Thm $4.6$ implies $\tilde{O}(T^{2/3})$ if $H(t) \leq \log(t)$. As the reviewer has noticed, these assumptions are restrictive but still extends beyond the exponentially stabilizing notions (geometrically series) and include some asymptotically stabilizing notions.
> Such theorems were given to fairly compare the existing works between ours. For example, in Table $1$, we presented the result when $H(t) \leq \log (t)$ to compare the regret between ours and the papers with exponentially stabilizing notions.
>
> On the other hand, Thm $4.1$ and the first part of Thm $4.6$ do not depend on $H(t)$. Whatever $H(t)$ comes in, we achieve an asymptotic system stability (Thm $4.1$), and concurrently, a sublinear regret bound is attained (see "we achieve a **sublinear** regret bound" in Thm $4.6$). In this case, we obtain a higher regret than $\tilde{O}(T^{2/3})$, but the regret bound is still sublinear. To the best of our knowledge, this is the first result in the literature that a sublinear regret is achieved even exponentially stabilizing controllers do not exist. To illustrate how challenging our setting is,
> let us further present a one-dimensional system, where the current system state is $1$ (newly presented in Figure $1(a)$). The goal is to achieve a state near $0$, and we would like to detect this stability by observing whether one arrives at a state less than $1-\epsilon$, where $\epsilon$ is an arbitrarily small positive number. Exponentially stabilizing controllers guarantee to detect the stability in $O(\log (1/\epsilon))$ time. However, with an asymptotically stabilizing controller, if the controller is designed to keep the system state unchanged for an arbitrarily long time $T$ and then collapse the state towards $0$ afterward, one cannot detect the stability before time $T$ regardless of how small $\epsilon$ is. In such a case, even though the controller ultimately achieves the goal, it may take a lot of time to learn whether a closed-loop system would be stable or not. Our algorithm achieves a **sublinear** regret even if only those troublesome controllers exist and simple (exponentially stabilizing) controllers do not exist at all. This achievement is due to our design of a *dynamic batch length* and an *adaptive learning rate*.
>
> To help reader understand which part needs $H(t)$ restriction, we will add the relevant explanations in the paper.
>
> **Main concern 2** (experiment): We have now provided the experiments on high-dimensional systems in Example $2$.
>
> Continued in the next rebuttal.

---

> ### Author Response · Authors · 2024-11-17
>
> **Question 1**: Thanks for the comment. In the new PDF, we added the explanation in the caption of Table 1: "$U$ is the set of destabilizing controllers and $|U|$ denotes its cardinality."
>
> **Question 2**: As mentioned in the response to Main concern 1, consider a single state $x(t)$ (also see Figure $1(a)$). Exponential stability means  there are positive constants $a$ and $b$ such that $|x(t)|\leq ae^{-bt}$, which means that the system should go to an equilibrium exponentially fast (the equilibrium is zero in this case). Asymptotic stability says $x(t)$ should eventually go to the equilibrium but there is no restriction on how fast it should go. It is just saying $\lim_{t\to\infty} x(t)=0$. In many systems in real world, say autonomous systems, we cannot make them go to a desired behavior exponentially fast. This is a very strong notion but is commonly used in control theory since it makes the mathematical analysis much simpler. Our paper adopts the realistic setting that only requires asymptotically stabilizing controllers.
>
> **Question 3**: Lemma $4.3$ is not to support the assumption $\lim_{t\to \infty} H(t) <\infty$. This lemma is to prove Thms $4.1$ and $4.2$ regardless of the assumption. Please notice that Thm $4.1$ does not require $\lim_{t\to \infty} H(t) <\infty$.
>
> **Question 4**: Technically, one can select any $z,v$ for the algorithm to work. It turns out that $z,v$ only affect the final regret only up to a constant. The simplest selection would be $z=v=1$. Then the polynomial batch length would start from $\tau_0 = \lfloor(\frac{1}{N(|U|+1)})^{1/2}\rfloor$ and subsequently have $\tau_b = \lceil(\frac{b+1}{N(|U|+1)})^{1/2}\rceil$ for $b\geq 1$, where $b$ is a batch number. To sum up, one can think of the batch length as growing proportionally to the square root of $b$. We will add some exposition to the paper to address this comment.
>
> Many thanks for reading our rebuttal.

---

> > ### Author Response · Authors · 2024-11-21
> > **Revisions addressing concerns about H(t)**
> >
> > To address the reviewer's first main concern about $H(t)$, we have slightly revised the introduction part, and added the relevant explanations in the paper. The revisions are blue-highlighted.
> >
> > Please note that
> >
> > 1. A *dynamic batch length* allows us to achieve both asymptotic system *stability* and a sublinear *regret*. However, we suffer from a multiplicative exponential term in return.
> >
> > 2. In Table 1, when using an *adaptive learning rate*, we can see that $o(T^{1/3})\cdot \exp(O(|U|))$ is improved to $\tilde{O}(T^{-1/3})\cdot \exp(O(|U|))$, and this is when $H(t) \leq O(\log {t})$ as we mentioned in the previous rebuttal. However, even when $H(t)$ is not restricted, $o(T^{1/3})\cdot \exp(O(|U|))$ is improved to $o(1)\cdot \exp(O(|U|))$ (see Corollary D.10), so it is true that multiplicative exponential term can be alleviated in all cases.
> >
> > Thus, our algorithm DBAR, the combination of these two components, effectively stabilizes the potential explosion of the system and enjoys the improved regret.
> >
> > Based on your constructive comments, we have added the following explanations in the updated manuscript:
> >
> > 1. In Lines 88-90 (in the introduction), we specified that a *dynamic batch length* (designed to grow unboundedly, but the growth amount eventually saturates) contributes to  both "asymptotic system stability" and "a sublinear regret".
> >
> > 2. In Lines 95-97 (in the introduction) and Lines 311-312, we specified that an *adaptive learning rate* contributes to alleviating a multiplicative exponential term in all cases. Moreover, we also indicated that the regret $\tilde{O}(T^{-1/3})\cdot \exp(O(|U|))$ is achievable for a specific class of stabilizing controllers.
> >
> > 3. In Lines 256-260, we again emphasized that a dynamic batch length contributes to  both "asymptotic system stability" and "a sublinear regret", and this was possible due to the design of $\lim_{b\to \infty} \frac{\tau_{b+1}}{\tau_b}=1$. For example, if we pick a geometric sequence (1,2,4,8, 16,...) as a dynamic batch length, both stability and a sublinear regret are not achievable.
> >
> > Many thanks.

---

> > > ### Comment · Reviewer_qZtK · 2024-11-23
> > >
> > > Thanks for your detailed response. My main concerns are addressed and I have increased my score.

---

> > > > ### Author Response · Authors · 2024-11-23
> > > >
> > > > Thank you very much! Please let us know if you have any further questions.

---

### Official Review · Reviewer_7GNK · 2024-10-30

**Soundness:** 3
**Presentation:** 2
**Contribution:** 2
**Rating:** 6
**Confidence:** 3

**Summary:**

This paper studies the problem of online bandit non-linear control where both transition dynamics and cost functions are non-linear and adversarial, and we only receive bandit feedback on either of them. Removing the assumption on exponentially stabilizing controllers, this paper ensured $\tilde{\mathcal O}(T^{2/3})$ regret with the help of only asymptotically stabilizing controllers.

Technically, this paper uses geometricly increasing batch lengthes to remove the requirement of exponentially stabilizing controllers, and further make the $\mathcal O(T^{1/3} e^{\lvert \mathcal U\rvert})$ dependency to $\mathcal O(T^{-1/3} e^{\lvert \mathcal U\rvert})$ via an adaptive learning rate tuning scheme.

**Strengths:**

1. The removal of exponentially stabilizing policies looks pretty good.
2. The resulting regret is satisfactory.
3. The algorithms are stated and motivated clearly.
4. Results are supplemented with numerical evaluations.

**Weaknesses:**

From the main text, it looks like the two main techniques, geometrically-increasing batch lengthes and geometrically-decaying learning rates, are the main technical contributions of this paper. But these two techniques seems both extensively studied in other online learning problems. So --
1. How are they different from those used in other online learning problems? Say, do you specifically adapt them to this control theory problem?
2. Do they bring special difficulties under this specific setup of nonlinear control? Say, do they break some previous analysis ideas in online nonlinear control?
3. Why previous works didn't use them?

[EDIT. After rebuttal, it turns out that batch lengths and learning rates are both not simply geometric, but follow some more meticulous designs.]

Minor:

1. To create a reference inside parentheses, please try using the command `\citep{}` instead of manually typing `(\citet{})`.
2. Section 2 is notation heavy. I suggest the authors to add subsections / paragraphs to group the definitions.
3. The results part (especially Section 4.2) contains a huge amount of formal statements with almost no informal descriptions of them; it is definitely not enjoyable to read. I suggest the authors to replace them with informal versions (and also add a paragraph of intuitive description before/after each of them) and defer the formal ones into the appendix.

**Questions:**

See Weaknesses.

---

> ### Author Response · Authors · 2024-11-17
>
> We are very grateful to the reviewer for a thorough evaluation of our work and providing valuable feedback. We will improve the paper based on the provided comments.
>
> We have improved the paper and uploaded a new PDF file with two **important** revisions:
>
> 1. **Experiments**: We have replaced the experiments in Example $2$ with high-dimensional experiments (100 states) on leader-follower systems. The leader is represented by a previous ball-beam (nonlinear) system, and the followers leverage the leader's state to stabilize themselves. Please check the experimental setup in Example $2$ (see pages 9-10).
>
> 2. **Illustration**: In a new Figure $1$, we have illustrated a concept of the controller pool for the unknown nonlinear system, and how general an asymptotically stabilizing notion is. We hope that this figure helps the reader to understand the core concepts (see page 4, Figure $1$).
>
> Responses to the reviewer:
>
> **Weakness** (Geometrical series and adaptation to control problem): Thanks for the comment. Our first technique is polynomially increasing batch length. We carefully designed the batch length to concurrently achieve asymptotic system stability and sublinear regret.
> It turns out that if we design the batch length to geometrically increase, we cannot achieve sublinear regret. Thus, the increase amount saturates as time goes by, which renders our algorithm more practical.
> The second technique is adaptive learning rate based on the state norm. It does not geometrically decay; we only decrease the learning rate if the state norm is large (state is unstable), and subsequently increase the learning rate if the state returns to a stable area.
>
> In online learning, the agent may progressively lengthen the batch length as the agent learns the system. Such a strategy is often used in incremental learning or curriculum learning, where the data availability or task complexity may increase over time.
> For example, [1] used geometrically increasing (doubling) batch length as they try to learn longer Atari games and escape local optima learned from short games.
> However, such a setting is different from ours, since we focus on a "single trajectory setting" in the control problem. Often, in a single trajectory setting, geometrically increasing batch length will lead to learning instabilty due to higher variance in learning a good policy. Thus, instead of using geometrically increasing length, we have carefully designed a polynomially increasing length to both guarantee stability and enough exploration.
>
> For the learning rate, in the context of online learning, it is well-known that reducing the learning rate will lead to early exploration and later stabilization. The work [2] adapts this advantage of decreasing learning rate to control problem and presents an algorithm based on non-increasing learning rate, but the paper only proves the results for constant learning rate.
> However, our claim is that decreasing the learning rate, regardless of the current state, does not significantly improve the control performance. We provide theoretical guarantees on our scheme based on the stability of state norm, and thus the rate is not necessarily non-increasing.
>
> **Weakness** (special difficulties and comparison with prior works): The previous work [2] assumed the existence of exponentially stabilizing controllers. The stabilizing behavior of such controllers should be exponentially fast, which may be quite restrictive for nonlinear systems to have such controllers. In that case, they do not need to use dynamic batch length since it does not take a lot of time to learn whether a closed-loop system would be stable or not. In our challenging setting where those simple controllers may not exist (stabilizng behavior can be arbitrarily slow in our case), a previously used "fixed batch length" fails to identify the system stability. Thus, we need a dynamic batch length to deal with a removal of exponentially stabilizing controllers.
> The use of a dynamic batch length in turn necessitates the use of an adaptive learning rate. For more details on the distinction between exponential and asymptotic notions, see Appendix A.
>
> **Minor 1**: Thanks for the recommendation. We updated the paper accordingly.
>
> **Minor 2**: To help readers understand Section 2 (assumptions and definitions), we provided the glossary on Appendix B. We will guide the readers to refer to Appendix B while reading Section $2$. Based on your comment, we have also presented Figure $1$ to illustrate the concept of an asymptotically stabilizing notion in the given controller pool.
>
> **Minor 3**: Thanks for the comment. We will soon add informal statements accordingly and defer some of formal theorems to the appendix.
>
> [1] Fuks et al., "An Evolution Strategy with Progressive Episode Lengths for Playing Games", IJCAI, 2019.
>
> [2] Li et al., "Online switching control with stability and regret guarantees", L4DC, 2023.
>
> Many thanks for reading our rebuttal.

---

> > ### Comment · Reviewer_7GNK · 2024-11-21
> >
> > Dear Authors,
> >
> > Thank you for your response! Yes, I misread your Assumption 3.1 as $\lim_{b\to \infty} \frac{\tau_{b+1}}{\tau_b}=\frac{\tau_1}{\tau_0}$. You mentioned that "It turns out that if we design the batch length to geometrically increase, we cannot achieve sublinear regret." -- can you kindly point me to the corresponding discussions in the text or give a quick justification here?
> >
> > I also admit that I interpreted your learning rate tuning too easily. Can you give a quick explanation on why "It indicates that the learning rate decreases in unstable states and increases back to the initial value when the state norm returns to a stable region" and also include it after Line 302 in your revision?
> >
> > I agree that both of them seem to be new. I'd be happy to increase my rating if you can let me understand what issues these two techniques are trying to tackle, and how they tackled these issues.
> >
> > Best,
> > Reviewer 7GNK

---

> ### Author Response · Authors · 2024-11-21
> **Reply**
>
> Thanks for the comments! We are really happy to hear back.
>
> **First of all**, for a dynamic batch length, Assumption 3.1 indicates that $\lim_{b \to \infty} \frac{\tau_{b+1}}{\tau_b} = 1$, which means the ratio of two consecutive batch lengths should approach 1 as time goes by (this includes polynomially increasing batch). The necessity of this assumption arises in the equation (14) (see Lines 993-999 in the Appendix), where we studied the asymptotic system stability. If we have $\lim_{b \to \infty} \frac{\tau_{b+1}}{\tau_b} > 1$, then the last part of (14) would be $0\cdot r^U$ instead of $0\cdot 1^U$ for some $r>1$, and this quantity is not guaranteed to converge to 0 since the number of Breaks $U$ goes to infinity as the number of batches $B$ goes to infinity. In other words, $0\cdot r^U = 0\cdot \infty \neq 0$ unless we have $r=1$ as our Assumption 3.1. Thus, the asymptotic system stability is violated. Not only that, the equation (14) is a core part of the sublinear regret proofs as in Line 1481 (Algorithm 1), Line 1637 (Algorithm 2), Line 1833 (Algorithm 3). Thus, if we have the ratio greater than 1, we cannot have a sublinear regret. Picking the ratio of increasing batch length to be 1 is perhaps the only way to obtain *both asymptotic system stability and a sublinear regret*, and we particularly picked polynomially increasing batch length among such batch length designs to achieve $\tilde{O}(T^{2/3})$ regret for the case $H(t) \leq O(\log(t))$.
>
> We really thank the reviewer's comment and we included a brief discussion with blue highlight (Lines 256-260) under Assumption 3.1 in the updated manuscript.
>
> **Second**, for an adaptive learning rate, Line 27 in Algorithm 1 says that $\eta_{b+1} = \frac{\eta_0}{ (\alpha_{b+1})^{s_{b+1}} }$, where $\eta_0$ is an initial learning rate. $\alpha_{b+1}$ and $s_{b+1}$ are determined in Lines 11-20 in the algorithm. To summarize, if the norm of the first state in the next batch is large ($||x_{t_{b+1}}||$ large), we adjust $(\alpha_{b+1})^{s_{b+1}}$ to be large ($s_{b+1}$ is increased by 1, and $\alpha_{b+1}$ is increased according to the norm). Subsequently, if $||x_{t_{b+1}}||$ is small enough (smaller than $\alpha_b ||x_0|| + \delta$), we let $s_{b+1}$ be $0$, meaning that we have exactly $\eta_{b+1} = \eta_0$. This implies that starting with an initial learning rate, if the state norm increases, a learning rate may decrease accordingly; however, if the state norm is stabilized, a learning rate increases back to an initial learning rate. We still can obtain a sublinear regret without this technique, but this allows us to alleviate the multiplicative exponential term in the regret bound ($o(T^{1/3})\cdot \exp(O(|U|))$ is improved to $\tilde{O}(T^{-1/3})\cdot \exp(O(|U|))$).
>
> As you recommended, we included the following brief sentence with blue highlight in Line 302 in the updated manuscript: "Since $(\alpha_{b+1})^{s_{b+1}}$ increases when the state norm $|| x_{t_{b+1}}||$ is large, and $s_{b+1}$ resets to zero for sufficiently small state norm, the corresponding learning rate decreases in unstable states and increases back to the initial value when the state norm returns to a stable region."
>
> **To conclude**, our dynamic batch length lets us achieve both asymptotic system stability and a sublinear regret, and an adaptive learning rate further improves this sublinear regret to $\tilde{O}(T^{2/3}) + \tilde{O}(T^{-1/3})\cdot \exp(O(|U|))$ in the case when $H(t) \leq O(\log(t))$. Thus, our algorithm DBAR, the combination of these two components, effectively stabilizes the potential explosion of the system and enjoys the improved regret.
>
> Many thanks!

---

> > ### Comment · Reviewer_7GNK · 2024-11-21
> >
> > Thank you for your explanations and revisions. I have updated my review accordingly.

---

> > > ### Author Response · Authors · 2024-11-21
> > >
> > > Thank you. Please let us know if you have any further questions!

---

### Official Review · Reviewer_BMap · 2024-11-04

**Soundness:** 3
**Presentation:** 3
**Contribution:** 2
**Rating:** 5
**Confidence:** 3

**Summary:**

The paper addresses the online bandit nonlinear control problem, where the objective is to learn an optimal controller for a nonlinear dynamical system amid unknown stabilizing and destabilizing controllers. The authors introduce the DBAR (Dynamic Batch length and Adaptive learning Rate) algorithm, which adapts batch length and learning rate to improve control stability and minimize regret without requiring exponentially stabilizing controllers. Unlike existing approaches that need stronger stability assumptions, DBAR works with a weaker, asymptotic stability notion. This flexibility allows DBAR to achieve stability and a regret bound even when the pool of stabilizing controllers is limited or includes only asymptotically stabilizing ones.

Theoretical contributions include proving asymptotic and finite-gain stability of DBAR and bounding its regret. The authors also compare DBAR's performance with existing algorithms, illustrating through simulations that DBAR provides improved stability and reduced regret in both linear and nonlinear systems under adversarial disturbances.

**Strengths:**

Originality:
    The paper introduces a new algorithm, DBAR, combining a dynamic batch length and an adaptive learning rate, which expands on existing Exp3 approaches for bandit learning. This design theoretically broadens the applicability of online control by relaxing the requirement for exponentially stabilizing controllers, a novel aspect in nonlinear control with adversarial disturbances.

Quality:
    The paper presents a well-structured algorithm and offers some theoretical analysis, including stability and regret bounds, which demonstrate rigor in method development. The authors provide proofs for their main results and discuss conditions under which stability is achieved, showcasing technical precision.

Clarity:
    The problem setup, assumptions, and main results are generally clear, with key terms defined (e.g., asymptotic stability and finite-gain stability) and a well-organized structure. The authors also make efforts to contextualize the algorithm in comparison to prior work, which aids readability for readers with a background in online control or bandit learning.

**Weaknesses:**

Originality:
    While DBAR's approach to adapting the learning rate and batch size is new, the actual contribution may be seen as incremental. The improvements over existing algorithms are mainly in modifying specific assumptions rather than introducing a breakthrough method, and it is unclear if these modifications substantively advance the field.

Quality:
    The theoretical guarantees, though valuable, rest on assumptions that may be difficult to satisfy in real-world applications, such as requiring knowledge of an ISS controller in advance. Furthermore, empirical validation is limited to relatively simple experiments, which may not fully demonstrate the algorithm’s effectiveness or scalability in more complex settings.

Significance:
    The practical impact of DBAR appears limited, as the algorithm’s assumptions—particularly on controller stability—may not align well with typical scenarios in nonlinear control. The significance for broader ML audiences is thus moderate, as DBAR seems primarily useful in niche settings where exact stability knowledge is available.

Significance:
    The paper's experimental evaluation does not robustly validate the theoretical claims, as it relies on lower-dimensional systems that may not capture the challenges posed by high-dimensional, complex environments. This limits the confidence in DBAR's utility beyond the examples shown, reducing its impact.

**Questions:**

1. How does the dynamically increasing batch length affect the algorithm's computational cost, particularly in real-time applications? It would be helpful to discuss any trade-offs between stability assurance and computational efficiency.

2. The current experimental setup involves relatively simple, low-dimensional systems. Can the authors provide results on more challenging benchmarks to better demonstrate DBAR’s performance and stability claims?

---

> ### Author Response · Authors · 2024-11-17
>
> We are very grateful to the reviewer for a thorough evaluation of our work and providing valuable feedback. We will improve the paper based on the provided comments.
>
> We have improved the paper and uploaded a new PDF file with two **important** revisions:
>
> 1. **Experiments**: We have replaced the experiments in Example $2$ with high-dimensional experiments (100 states) on leader-follower systems. The leader is represented by a previous ball-beam (nonlinear) system, and the followers leverage the leader's state to stabilize themselves. Please check the experimental setup in Example $2$ (see pages 9-10).
>
> 2. **Illustration**: *Based on your constructive comment*, in a *new* Figure $1$, we have illustrated a concept of the controller pool for the unknown nonlinear system, and how general an asymptotically stabilizing notion is. We hope that this figure helps ML audiences understand the core concepts of our setup in nonlinear control. (see page 4, Figure $1$).
>
> Responses to the reviewer:
>
> **Weakness 1** (Originality): We appreciate the reviewer's comment. As the reviewer stated in the strengths, our main contribution is on relaxing the requirement for exponentially stability controller. We carefully designed the polynomially increasing batch length and adaptive learning rate based on the state norm to achieve desirable properties even when only asymptotically stable controllers exist. We would like to emphasize that we did not use an existing method as is and our results all depend on our method of adjusting the batch length and learning rate in the learning process.
>
> The existing methods only deal with exponentially stable controllers while we focus on a much broader class of asymptotically stable controllers. To illustrate how significant the difference between the two types of controllers is, let us further present a one-dimensional system, where the current system state is $1$ (newly presented in Figure $1(a)$). The goal is to achieve a state near $0$, and we would like to detect this stability by observing whether one arrives at a state less than $1-\epsilon$, where $\epsilon$ is an arbitrarily small positive number. Exponentially stabilizing controllers guarantee to detect the stability in $O(\log (1/\epsilon))$ time. However, with an asymptotically stabilizing controller, if the controller is designed to keep the system state unchanged for an arbitrarily long time $T$ and then collapse the state towards $0$ afterward, one cannot detect the stability before time $T$ regardless of how small $\epsilon$ is. In such a case, even though the controller ultimately achieves the goal, it may take a lot of time to learn whether a closed-loop system would be stable or not. Our algorithm works even if only those troublesome controllers exist and simple (exponentially stabilizing) controllers do not exist at all. For more details, please see Appendix A.
> Furthermore, to provide a second perspective on this issue, as mentioned in the paper (Lines 69-79), the difference between exponential stability and asymptotic stability is tantamount to the difference between convexity and strong convexity. As witnessed in many machine learning problems, strong convexity is a very restrictive assumption while convexity is more realistic.
>
> Meanwhile, our numerical experiment also shows that the relaxed controller stability assumption is crucial. Notice that Li et al.'s work assumes at least one exponentially stabilizing controller in the pool (see Table 1). In Figure $2(a)$, Li et al.'s work ("fixed $\tau$, fixed $\eta$") stabilizes the linear system (the state norm reaches near zero), putting aside the time needed for the stabilization. This is because an exponentially stabilizing controller always exists for the linear system. However, in the nonlinear system case where an exponentially stabilizing controller may not exist (see Figure $3(b)$), Li et al.'s algorithm  incurs the system to explode, while our DBAR ("dynamic $\tau$, adaptive $\eta$") successfully stabilizes the system and achieves a good regret. By comparing Li et al's algorithm in Figures $2$ and $3$, we observe that allowing a broader class of controllers can greatly affect the algorithm's performance, and our DBAR works well even when exponentially stabilizing controllers do not exist.
>
> Continued in the next rebuttal.

---

> ### Author Response · Authors · 2024-11-17
>
> **Weakness 2** (Quality): There may have been a misunderstanding. Indeed, our assumption does not require prior knowledge of an ISS controller. We do not need to know which controller is ISS or not; we only require that at least one ISS controller "exists" in the candidate pool.
> The ISS principle is a widely understood concept for handling disturbances in nonlinear systems, particularly in applications such as automotive and robotics systems, as it ensures that a vehicle or robot can perform safely despite variations in exogenous inputs like road conditions or obstacles.
> Moreover, as mentioned above, we only assume the existence of "asymptotically" ISS stabilizing controllers whose stabilizing behavior can be arbitrarily slow.
> Therefore, the existence of an ISS controller is not a particularly restrictive assumption.
> To support the scalability, we have presented new experiments in Example $2$.
>
> **Weakness 3** (Significance): Controller stability and ISS stability is the same concept in our context. We are not sure about the reviewer's comment. Perhaps, some terminologies in the paper have caused a confusion. We believe that the typical scenario in nonlinear control is to design a controller that takes the states to an equilibrium, and this is about the design of stabilizing controllers. We have addressed the same problem in our paper. We do not assume stability knowledge. Given an unknown system, we can consider different regimes to model the possibilities of the unknown parameters of the system and then design a controller for each possibility. For example (see a *new* Figure $1(b)$ for the pictorial illustration for readers), consider a robot operating in an uncertain environment where there are different scenarios that could happen in the environment and we model it by a vector "$(a,b)$" for simplicity. We consider different intervals such as $[0,1)\times [0,1)$ or $[1,2)\times[1,2)$, etc. for "$(a,b)$" and then design a controller for each scenario. Now, we have a pool of controllers and we know at least one of them should work but do not know which one since we are not aware of the exact parameter of the system. Our method learns a correct controller from the pool. Our stability notion is the same as the one routinely considered in the control theory area as well as the new area of machine learning for control systems.
>
> **Weakness 4** (Significance): Thank you for your comment. We have a mathematical proof showing that the proposed idea works. We have provided new high-dimensional experiments in Example $2$.
>
> **Question 1**: The magnitude of learning rate and batch length does not affect the algorithm's complexity. For the learning rate, it is simply used to calculate Line 28 of Algorithm $1$ regardless of its magnitude. For the batch length, the only (potential) intensive part is Line 21 of Algorithm $1$. Given the batch $b$ and its length $\tau_b$, it simply adds all of the bandit feedback over $b$, so the complexity of the line is $O(\tau_b)$. Since the sum of all batch lengths is the algorithm time $T$, the total complexity would be $O(T)$ and this is certainly a linear-time algorithm.
>
> **Question 2**: We have now provided 100-dimensional systems in Example $2$.
>
> Many thanks for reading our rebuttal.

---

> > ### Author Response · Authors · 2024-11-25
> > **Hoping for Feedback**
> >
> > Dear Reviewer BMap,
> >
> > Since the discussion period will end in a couple of days, we would greatly appreciate it if you could check our responses and let us know whether there are further concerns or issues that we can address. We hope that the reviewer will kindly reconsider their score if our responses are satisfactory. Many thanks.
> >
> > Best regards,
> >
> > Submission1215Authors

---

### Author Response · Authors · 2024-11-20

Dear reviewers: Since the discussion period would end in a week, could you please read our rebuttal and let us know if you have any further concerns or comments? We provided extensive answers to your previous comments and hope to discuss them with you during this discussion period. We appreciate your time and effort.

Best regards,

Submission1215Authors

---

### Meta-Review · Area_Chair_8tCP · 2024-12-18

**Metareview:**

Summary:
This paper investigates online nonlinear control using bandit feedback. Its primary contribution lies in introducing dynamic batch length and learning rate techniques to relax the commonly used exponentially stabilizing controller assumption. Instead, it adopts a weaker assumption termed the asymptotic stabilizing assumption, achieving a better regret bound. Experimental results support the authors' claims.

Strengths:

- The use of dynamic batch size and adaptive step size to mitigate stability issues is a compelling approach.
- The motivation for this work is clear and well-articulated.
- The experiments, while conducted on simulations, seem sound and adequately support the proposed methodology.

Weaknesses:

- The paper’s presentation needs significant improvement and currently falls short of publication standards. Both my own reading and other reviewers' comments highlight the excessive reliance on dense notations, making the paper less accessible to readers unfamiliar with the field.
- The significance of the asymptotic stability assumption is insufficiently emphasized, leaving the contribution somewhat unclear in the context of existing literature. The draft lacks clarity in positioning this work within the broader research landscape.

Decision: I recommend rejecting this paper in its current form due to the weaknesses outlined above.

**Additional Comments On Reviewer Discussion:**

During the discussion phase, the authors expanded on the motivation behind this work and added additional simulation experiments. However, I believe that the current writing quality still falls short of the standard expected for an ICLR publication.

---

### Decision · Program_Chairs · 2025-01-22

Reject